# Kaleidoscope: An Efficient, Learnable Representation For All Structured Linear Maps

**Tri Dao** [1], **Nimit Sharad Sohoni**[\*2], **Albert Gu**[\*1], **Matthew Eichhorn** [3], **Amit Blonder** [4],
**Megan Leszczynski** [1], **Atri Rudra** [4], **Christopher Ré** [1]

[1] Department of Computer Science, Stanford University
[2] Institute for Computational and Mathematical Engineering, Stanford University
[3] Center for Applied Mathematics, Cornell University
[4] Department of Computer Science and Engineering, University at Buffalo, The State University of New York

`{trid,nims,albertgu}@stanford.edu, mae226@cornell.edu, amitblon@buffalo.edu,`
`mleszczy@stanford.edu, atri@buffalo.edu, chrismre@cs.stanford.edu`

## Abstract

Modern neural network architectures use structured linear transformations, such as low-rank matrices, sparse matrices, permutations, and the Fourier transform, to improve inference speed and reduce memory usage compared to general linear maps. However, choosing which of the myriad structured transformations to use (and its associated parameterization) is a laborious task that requires trading off speed, space, and accuracy. We consider a different approach: we introduce a family of matrices called *kaleidoscope matrices* (K-matrices) that provably capture *any* structured matrix with near-optimal space (parameter) and time (arithmetic operation) complexity. We empirically validate that K-matrices can be automatically learned within end-to-end pipelines to replace hand-crafted procedures, in order to improve model quality. For example, replacing channel shuffles in ShuffleNet improves classification accuracy on ImageNet by up to 5%. K-matrices can also simplify hand-engineered pipelines—we replace filter bank feature computation in speech data preprocessing with a learnable kaleidoscope layer, resulting in only 0.4% loss in accuracy on the TIMIT speech recognition task. In addition, K-matrices can capture latent structure in models: for a challenging permuted image classification task, a K-matrix based representation of permutations is able to learn the right latent structure and improves accuracy of a downstream convolutional model by over 9%. We provide a practically efficient implementation of our approach, and use K-matrices in a Transformer network to attain 36% faster end-to-end inference speed on a language translation task.

## 1 Introduction

Structured linear maps are fundamental and ubiquitous in modern machine learning. Their efficiency in speed (fast algorithms) and space (few parameters) can reduce computation and memory usage. The class of structured linear maps includes fixed specialized transforms such as the discrete Fourier transform (DFT) and Hadamard transform used in signal processing (Cooley et al., 1969), convolutions for image, language, and speech modeling (Gu et al., 2018), and low-rank and sparse matrices for efficient storage and inference on edge devices (Yu et al., 2017). Forms of structure such as sparsity have been at the forefront of recent advances in ML (Frankle & Carbin, 2019), and are critical for on-device and energy-efficient models, two application areas of tremendous recent interest (Tsidulko, 2019; Schwartz et al., 2019).

There are a plethora of classes of structured linear maps, each with a significantly different representation, algorithm, and implementation. They have different tradeoffs in terms of inference speed, training speed, and accuracy, and the conventional wisdom is that no one class works uniformly well across all applications. As a result, ML practitioners currently *hand-pick* specific classes of structured linear maps for each of their applications. This is a difficult and labor-intensive task.

---

[\*]These authors contributed equally.

Ideally, these problems should be addressed with a *universal* representation for structured linear maps: (i) Such a parameterization should be *expressive* enough to capture important classes of structure, with a nearly tight parameter count and runtime: the space required to represent the linear map should be close to optimal, and the resulting algorithm for matrix vector multiplication should be close to the fastest possible algorithm. (ii) The parameterization should be differentiable in order to be *learned* as a component of end-to-end ML pipelines, enabling it to easily be used as a drop-in replacement for manually engineered structured components. (iii) The parameterization should admit practically *efficient* algorithms for training and inference, in terms of both speed and memory.

Currently, no class of structured linear maps satisfies all of these criteria. Most existing classes of structured matrices—such as the class of low-rank matrices—fail to tightly capture other important types of structure. For example, the DFT has an efficient structured representation of size $O(n \log n)$, yet cannot be well-approximated by a low-rank transform of size $\ll n^2$. Another important type of structure is *sparsity*; lots of exciting recent work has focused on the design of sparse neural networks. For instance, sparse networks of comparable quality to their dense counterparts—yet an order of magnitude fewer parameters—may be created via pruning (Han et al., 2016) or by identifying "winning lottery tickets" (Frankle & Carbin, 2019). In parallel, recent theoretical results by De Sa et al. (2018) show that sparsity and the notion of structure in linear maps are fundamentally linked: any given matrix can be factored into a product of sparse matrices with total parameter count equal to the efficiency (i.e. minimum arithmetic circuit complexity) of the matrix. In other words, the representation of linear maps as products of sparse matrices tightly captures *all* forms of structure. Unfortunately, it is difficult to actually *learn* these sparse factorizations, because it requires finding the sparsity patterns of the factors—a discrete, nondifferentiable search problem. Thus, current methods for training sparse neural networks are either expensive (Frankle & Carbin, 2019) or rely on highly hand-tuned heuristics for evolving the sparsity patterns throughout training (Dettmers & Zettlemoyer, 2019).

By contrast, we propose a representation of linear maps as products of sparse matrices with specific *predefined* sparsity patterns (Section 2), and show that it *does* satisfy our desiderata: it retains the expressiveness of unstructured sparsity, while being differentiably learnable and efficient like other structured representations. Concretely, our representation is based on products of a particular building block known as a *butterfly* matrix (Parker, 1995; Dao et al., 2019); we term such products *kaleidoscope matrices* (K-matrices for short).[1] (i) Our main theoretical contribution (Section 2.3) concerns the *expressiveness* of this representation: we show that any structured linear map (i.e. one that can be applied using $s \ll n^2$ arithmetic operations) can be represented as a K-matrix, with a nearly tight number of parameters and algorithmic complexity (both on the order of $s$ up to logarithmic factors). (ii) The kaleidoscope representation is fully differentiable; thus, all the parameters of a K-matrix can be *learned* using standard optimization algorithms such as SGD. (iii) Because of their simple, regular structure, K-matrices are practical and easy to use. We provide memory- and runtime-*efficient* implementations of K-matrix multiplication on CPU and GPU for training and inference, with a simple PyTorch interface.

We empirically validate that, due to their expressiveness, learnability, and efficiency, we can use K-matrices as a drop-in replacement for linear components in deep learning models. In Section 3.1, we use K-matrices to replace hand-crafted structure in two different settings. We simplify the six steps of filter bank computation in speech preprocessing into a single learnable K-matrix step, with only an 0.4% accuracy drop on the TIMIT speech recognition task. We use K-matrices to replace channel shuffles in ShuffleNet, improving ImageNet classification accuracy by up to 5%. In Section 3.2, we show that K-matrices can successfully recover latent structure; a K-matrix is used to learn latent permutations in a permuted image dataset (Permuted CIFAR), resulting in 9 points higher accuracy in a downstream CNN model. In Section 3.3, we show that our efficient K-matrix multiplication implementation can be applied to speed up real-world tasks: we replace linear layers with K-matrices in a DynamicConv-Transformer network to attain 36% faster end-to-end inference speed with a 1.0 drop in BLEU score on the IWSLT14 German→English translation task.

---

[1] A group of butterflies is known as a kaleidoscope.

## 2 A NEARLY-TIGHT PARAMETERIZATION OF ALL STRUCTURED MATRICES

We first present some background on the characterization of all structured matrices (i.e. those with subquadratic multiplication algorithms) as products of sparse factors, along with the definition of butterfly matrices. We then propose a differentiable family of kaleidoscope matrices, composed of products of butterfly matrices, and prove their expressivity: all structured matrices can be represented in this form, with almost optimal parameter count and runtime.

### 2.1 BACKGROUND: SPARSE FACTORIZATION, BUTTERFLY MATRICES

**Sparse factorization**   One method of constructing matrices with theoretically fast matrix-vector multiplication algorithms is as a product of sparse matrices, so that multiplication by an arbitrary vector has cost proportional to the total number of nonzeros (NNZ) of the matrices in the product. Surprisingly, the converse is also true. De Sa et al. (2018) introduce the concept of *sparse product width* (SPW), which roughly corresponds to the total NNZ in a factorization of a matrix, and show that it is an asymptotically optimal descriptor of the algorithmic complexity of matrix-vector multiplication (Bürgisser et al., 2013). We use a similar argument in the proof of our main theorem (Section 2.3). However, attempting to *learn* such a factorization of a given matrix is difficult, as the sparsity constraint is not continuous. Moreover, because of the possibly irregular sparsity patterns, it is difficult to realize the theoretical speedups in practice (Gray et al., 2017; Gahvari et al., 2007).

**Butterfly matrices**   Butterfly matrices, encoding the recursive divide-and-conquer structure of the fast Fourier transform (FFT) algorithm, have long been used in numerical linear algebra (Parker, 1995; Li et al., 2015) and machine learning (Mathieu & LeCun, 2014; Jing et al., 2017; Munkhoeva et al., 2018; Dao et al., 2019; Choromanski et al., 2019). Here we define butterfly matrices, which we use as a building block for our hierarchy of kaleidoscope matrices.

**Definition 2.1.** *A **butterfly factor** of size $k \geq 2$ (denoted as $\mathbf{B}_k$) is a matrix of the form $\mathbf{B}_k = \begin{bmatrix} \mathbf{D}_1 & \mathbf{D}_2 \\ \mathbf{D}_3 & \mathbf{D}_4 \end{bmatrix}$ where each $\mathbf{D}_i$ is a $\frac{k}{2} \times \frac{k}{2}$ diagonal matrix. We restrict $k$ to be a power of 2.*

**Definition 2.2.** *A **butterfly factor matrix** of size $n$ with block size $k$ (denoted as $\mathbf{B}_k^{(n)}$) is a block diagonal matrix of $\frac{n}{k}$ (possibly different) butterfly factors of size $k$:*

$$\mathbf{B}_k^{(n)} = \mathrm{diag}\left([\mathbf{B}_k]_1, [\mathbf{B}_k]_2, \ldots, [\mathbf{B}_k]_{\frac{n}{k}}\right)$$

**Definition 2.3.** *A **butterfly matrix** of size $n$ (denoted as $\mathbf{B}^{(n)}$) is a matrix that can be expressed as a product of butterfly factor matrices: $\mathbf{B}^{(n)} = \mathbf{B}_n^{(n)} \mathbf{B}_{\frac{n}{2}}^{(n)} \ldots \mathbf{B}_2^{(n)}$. Equivalently, we may define $\mathbf{B}^{(n)}$ recursively as a matrix that can be expressed in the following form:*

$$\mathbf{B}^{(n)} = \mathbf{B}_n^{(n)} \begin{bmatrix} [\mathbf{B}^{(\frac{n}{2})}]_1 & 0 \\ 0 & [\mathbf{B}^{(\frac{n}{2})}]_2 \end{bmatrix}$$

*(Note that $[\mathbf{B}^{(\frac{n}{2})}]_1$ and $[\mathbf{B}^{(\frac{n}{2})}]_2$ may be different.)*

### 2.2 THE KALEIDOSCOPE HIERARCHY

Using the building block of butterfly matrices, we formally define the kaleidoscope ($\mathcal{BB}^*$) hierarchy and prove its expressiveness. This class of matrices serves as a fully differentiable alternative to products of sparse matrices (Section 2.1), with similar expressivity. In Appendix J, we show where various common structured matrix classes are located within this hierarchy.

The building block for this hierarchy is the product of a butterfly matrix and the (conjugate) transpose of another butterfly matrix (which is simply a product of butterfly factors taken in the opposite order). Figure 1 visualizes the sparsity patterns of the butterfly factors in $\mathcal{BB}^*$, where the red and blue dots represent the allowed locations of nonzero entries.

**Definition 2.4** (Kaleidoscope hierarchy, kaleidoscope matrices)**.**

- *Define $\mathcal{B}$ as the set of all matrices that can be expressed in the form $\mathbf{B}^{(n)}$ (for some $n$).*
- *Define $\mathcal{BB}^*$ as the set of matrices $\mathbf{M}$ of the form $\mathbf{M} = \mathbf{M}_1 \mathbf{M}_2^*$ for some $\mathbf{M}_1, \mathbf{M}_2 \in \mathcal{B}$.*

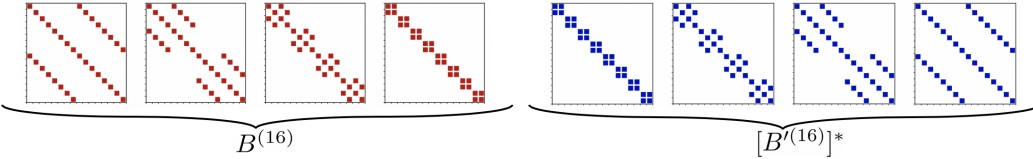

Figure 1: Visualization of the fixed sparsity pattern of the building blocks in $\mathcal{BB}^*$, in the case $n = 16$. The red and blue dots represent all the possible locations of the nonzero entries.

- *Define $(\mathcal{BB}^*)^w$ as the set of matrices $\mathbf{M}$ that can be expressed as $\mathbf{M} = \mathbf{M}_w \ldots \mathbf{M}_2 \mathbf{M}_1$, with each $\mathbf{M}_i \in \mathcal{BB}^*$ ($1 \le i \le w$). (The notation $w$ represents **width**.)*
- *Define $(\mathcal{BB}^*)^w_e$ as the set of $n \times n$ matrices $\mathbf{M}$ that can be expressed as $\mathbf{M} = \mathbf{S}\mathbf{E}\mathbf{S}^T$ for some $en \times en$ matrix $\mathbf{E} \in (\mathcal{BB}^*)^w$, where $\mathbf{S} \in \mathbb{F}^{n \times en} = [\mathbf{I}_n \quad 0 \quad \ldots \quad 0]$ (i.e. $\mathbf{M}$ is the upper-left corner of $\mathbf{E}$). (The notation $e$ represents **expansion** relative to $n$.)*
- *$\mathbf{M}$ is a **kaleidoscope matrix**, abbreviated as **K-matrix**, if $M \in (\mathcal{BB}^*)^w_e$ for some $w$ and $e$.*

The *kaleidoscope hierarchy*, or $(\mathcal{BB}^*)$ hierarchy, refers to the families of matrices $(\mathcal{BB}^*)^1_e \subseteq (\mathcal{BB}^*)^2_e \subseteq \ldots$, for a fixed expansion factor $e$. Each butterfly matrix can represent the identity matrix, so $(\mathcal{BB}^*)^w_e \subseteq (\mathcal{BB}^*)^{w+1}_e$. We show that the inclusion is proper in Appendix E. This hierarchy generalizes the $\mathcal{BP}$ hierarchy proposed by Dao et al. (2019), as shown in Appendix J.

**Efficiency in space and speed** Each matrix in $(\mathcal{BB}^*)^w_e$ is a product of $2w$ total butterfly matrices and transposes of butterfly matrices, each of which is in turn a product of $\log(ne)$ factors with $2ne$ nonzeros (NNZ) each. Therefore, each matrix in $(\mathcal{BB}^*)^w_e$ has $4wne\log(ne)$ parameters and a matrix-vector multiplication algorithm of complexity $O(wne\log ne)$ (by multiplying the vector with each sparse factor sequentially). We prove this more formally in Appendix E. For the applications in Section 3, $w$ and $e$ are small constants (up to 2), so those K-matrices have $O(n\log n)$ parameters and runtime.

## 2.3 ALL LOW-DEPTH STRUCTURED MATRICES ARE IN THE KALEIDOSCOPE HIERARCHY

We now present our main theoretical result: the fact that general linear transformations, expressed as low-depth linear arithmetic circuits, are captured in the $\mathcal{BB}^*$ hierarchy with low width. Arithmetic circuits are commonly used to formalize algebraic algorithmic complexity (Bürgisser et al., 2013); we include a primer on this in Appendix M. The quantities of interest are the total number of *gates* in the circuit, representing the total number of steps required to perform the algorithm for a serial processor, and the *depth*, representing the minimum number of steps required for a parallel processor.

**Theorem 1.** *Let $\mathbf{M}$ be an $n \times n$ matrix such that multiplication of $\mathbf{M}$ times an arbitrary vector $\mathbf{v}$ can be represented as a linear arithmetic circuit with $s$ total gates and depth $d$. Then, $\mathbf{M} \in (\mathcal{BB}^*)^{O(d)}_{O(\frac{s}{n})}$.*

The representation of such a matrix $\mathbf{M}$ in the $\mathcal{BB}^*$ hierarchy has $O(ds\log s)$ parameters and yields a $O(ds\log s)$ multiplication algorithm, compared to the $O(s)$ parameters and runtime of the circuit representation. To the best of our knowledge, the most general classes of efficient matrices that have been studied (De Sa et al., 2018) have depth $d$ on the order of $\log n$ or poly $\log n$. In these cases, the representation with K-matrices matches the best known bounds up to polylogarithmic factors.

The crux of the proof of Theorem 1 (shown in Appendix F) is the construction of an almost tight representation of any sparse matrix as a K-matrix (i.e. a product of butterfly matrices): specifically, we show that any $n \times n$ sparse matrix with $s$ nonzeros is in $(\mathcal{BB}^*)^{O(\lceil \frac{s}{n} \rceil)}_{O(1)}$ (Theorem 3, Appendix I). We then leverage the expressivity result of products of sparse matrices to represent all arithmetic circuits (similar to the sparse product width result of De Sa et al. (2018) referenced in Section 2.1) to complete the proof of Theorem 1.

This intermediate result is also a novel characterization of sparse matrices. For a matrix with $s$ NNZ, the kaleidoscope representation has $O(s\log n)$ parameters and runtime, instead of the optimal $O(s)$ parameters and runtime; so, we trade off an extra logarithmic factor in space and time for full differentiability (thanks to the fixed sparsity patterns in the representation). The intuition behind

the result is as follows: a sparse matrix with $s$ NNZ can be written as a sum of $\lceil s/n \rceil$ matrices each with at most $n$ NNZ. Any $n \times n$ matrix with at most $n$ NNZ, up to permuting the rows and columns, is a product of two butterfly matrices (Lemma I.1). Sorting networks (Knuth, 1997) imply that permutation matrices are in $(\mathcal{BB}^*)^{O(\log n)}$, but we tighten the result to show that they are in fact in $\mathcal{BB}^*$ (Theorem 2, Appendix G). We thus obtain a kaleidoscope representation for each summand matrix with $O(n \log n)$ parameters. By the addition closure property of the $\mathcal{BB}^*$ hierarchy (Lemma H.5), each sparse matrix with $s$ NNZ then has a kaleidoscope representation with $O(s \log n)$ parameters.

**Tight representation for structured linear maps common in ML**   Even though Theorem 1 suggests that the kaleidoscope representation can be loose by logarithmic factors, many structured linear maps common in ML can be represented in this hierarchy with an *optimal* number of parameters and runtime compared to the best known parameterizations, up to constant factors. Appendix J includes several examples such as discrete transforms (the DFT, discrete cosine transform (DCT), discrete sine transform (DST), and Hadamard transform), convolution (i.e. circulant matrices), Toeplitz matrices (Gray, 2006), structured matrices for kernel approximation $((HD)^3$ (Yu et al., 2016)) and compact neural network design (Fastfood (Le et al., 2013), ACDC (Moczulski et al., 2016)). There have been other large classes of structured matrices proposed in the machine learning literature, such as Toeplitz-like (Sindhwani et al., 2015) and low displacement rank (LDR) (Thomas et al., 2018), but they are not known to be able to capture these common structures as tightly as K-matrices can. More detailed discussions are in Appendix A.

## 2.4   EXTENSIONS

**ReLU networks with low-depth structured weight matrices**   In Appendix L, we prove that finding an efficient circuit for a ReLU network can be reduced to finding efficient circuits for each of its weight matrices, with at most a constant factor greater size and run-time (i.e. number of gates). We also show that ReLU networks with kaleidoscope weight matrices have near-linear VC dimension in the number of parameters, matching the bound for networks with unconstrained weight matrices (Bartlett et al., 1999; Harvey et al., 2017) and LDR (Thomas et al., 2018). This yields a corresponding sample complexity bound.

**Orthogonal kaleidoscope hierarchy**   *Orthogonal* butterfly matrices are one commonly used variant due to their improved stability (Parker, 1995), where each butterfly factor is constrained to be orthogonal: $\begin{bmatrix} \mathbf{C} & \mathbf{S} \\ -\mathbf{S} & \mathbf{C} \end{bmatrix}$ with $\mathbf{C}, \mathbf{S}$ being diagonal and $\mathbf{C}^2 + \mathbf{S}^2 = \mathbf{I}$. Similar to the $\mathcal{BB}^*$ hierarchy, in Appendix K, we define the $\mathcal{OBB}$ hierarchy consisting of products of orthogonal butterfly matrices and diagonal matrices, and show that this hierarchy has the same expressiveness as the $\mathcal{BB}^*$ hierarchy.

## 3   EMPIRICAL EVALUATION

We validate three claims that suggest that kaleidoscopes are a promising technique to learn different types of structure in modern architectures.

1. Section 3.1: for applications in speech and lightweight computer vision relying on highly hand-crafted structured transformations, we show that we can recover—and even improve—the quality of such architectures by simply replacing existing hand-structured components with K-matrices, with only a small overhead in memory and computation.
2. In Section 3.2, for a challenging task with latent structure (Permuted CIFAR-10), a K-matrix-based relaxation of permutations is able to learn the right latent permutation, yielding 9 points better accuracy in a downstream CNN compared to standard RNN and CNN baselines used on such permuted image classification tasks.
3. In Section 3.3, we show that, although not yet highly optimized, our current implementation of K-matrices can improve the inference throughput of DynamicConv Transformer, a state-of-the-art fast machine translation model, by 36%, with only a relatively small drop in translation quality.

In all of the above applications, as K-matrices are fully differentiable, we simply train them jointly with the rest of the model using standard learning algorithms (such as SGD). Full details for all of the experiments (precise architectures, hyperparameters, etc.) are in Appendix B [2].

## 3.1 REPLACING HAND-CRAFTED STRUCTURES

We validate that kaleidoscope matrices can recover or improve on the performance of hand-crafted structure in ML models. For example, a single learnable kaleidoscope layer can be used to replace the hand-engineered filter bank speech preprocessing pipeline with only 0.4% loss in accuracy on the TIMIT speech recognition task (Section 3.1.1). Replacing channel shuffles in ShuffleNet with learnable K-matrices improves classification accuracy on ImageNet by up to 5.0% (Section 3.1.2).

### 3.1.1 SPEECH PREPROCESSING

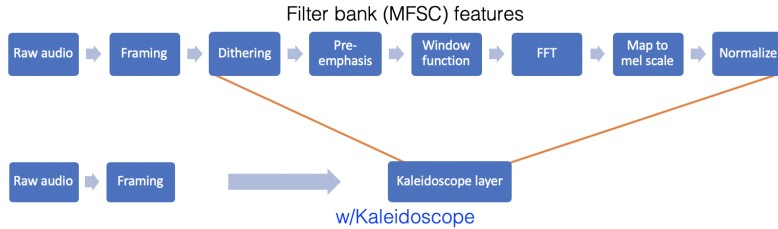

Figure 2: Comparison of the standard MFSC featurization pipeline with our "kaleidoscope" pipeline.

We show that K-matrices can remove the need for hand-tuning by significantly simplifying speech recognition data preprocessing pipelines. In particular, we can entirely replace the complex hand-crafted MFSC featurization commonly used in speech recognition tasks with a fully learnable kaleidoscope layer, with only 0.4% drop in accuracy on the TIMIT speech recognition benchmark. Results are presented in Table 1. Our approach is competitive with the accuracy of standard models that use hand-crafted features, and significantly outperforms current approaches for learning from raw audio input.

Table 1: TIMIT phoneme error rate (PER%) for different methods. Our kaleidoscope, raw-input version of the model (row 3) performs competitively with the original model trained on MFSC features (row 1), with only an 0.4% drop in PER. It significantly outperforms existing approaches that learn from raw audio, i.e. without handcrafted featurization (e.g. SincNet [row 2], which to our knowledge attains the previous state-of-the-art for learning from raw audio), and is only 0.8% less accurate than the overall state-of-the-art on TIMIT.[3] Additional comparisons are given in Appendix B.1.

| Method | Test set PER% | Raw audio input |
|---|---|---|
| MFSC features + LSTM | 14.2 | ✗ |
| SincNet (Ravanelli et al., 2019) | 17.2 | ✓ |
| *Kaleidoscope + LSTM* | 14.6 | ✓ |

Modern speech recognition models currently rely on carefully hand-crafted features extracted from the audio, which are then fed into an *acoustic model*. By contrast, learning directly from the raw audio—i.e. end-to-end learning from the audio waveform without any manual featurization—obviates the need for this complicated and often expensive preprocessing step. There have been recent attempts to learn directly from raw audio, such as SincNet (Ravanelli & Bengio, 2018); however, they often rely on specialized architectures designed by domain experts. Instead, we use a standard RNN speech recognition architecture, but use a learnable kaleidoscope layer to replace the featurization steps.

---

[2]Code that implements Kaleidoscope matrix multiplication is available at https://github.com/HazyResearch/learning-circuits

[3]The current state-of-the-art results from Ravanelli et al. (2018) use a concatenation of *three* different speech audio featurizations—MFSC, MFCC, and fMLLR—as the neural network input, along with a customized RNN architecture (LiGRU) specifically designed for speech recognition.

The baseline architecture takes as input *filter bank* (MFSC) features, which are a popular standard featurization for speech recognition (Paliwal, 1999) and involve several steps hand-crafted specifically for this domain. These features are extracted from the raw audio waveform, and fed as the input into a Bi-LSTM model. We significantly simplify this pipeline by replacing the featurization step with a trainable kaleidoscope layer that is trained end-to-end together with the Bi-LSTM. The original pipeline and our modified kaleidoscope version are depicted in Figure 2.

The computation of MFSC features involves a series of painstakingly hand-designed steps (further described in Appendix B.1), each involving their own hyperparameters: (i) the waveform is *framed* (split into chunks), (ii) the waveform is *dithered* (noise is added), (iii) pre-emphasis is applied, (iv) the Hamming window is applied, (v) the FFT is applied and the power spectrum is computed, (vi) the result is mapped to the *mel scale* (which involves applying a particular linear transformation and then taking the logarithm of the result), (vii) cepstral mean and variance normalization is applied. We replace the last six steps (ii-vii) of this featurization process with a learnable kaleidoscope layer; specifically, after windowing, we multiply the input by a K-matrix, and then compute the logarithm of the power spectrum; the output is fed into the Bi-LSTM model.

### 3.1.2 REPLACING CNN CHANNEL SHUFFLE

We evaluate how K-matrices can improve the quality of hand-crafted, lightweight architectures for computer vision tasks, without the need for hand-tuning. We select ShuffleNet (Zhang et al., 2018), which is a state-of-the-art lightweight CNN architecture that uses a manually designed "channel shuffle" permutation matrix to improve performance. By replacing this fixed permutation with a learnable K-matrix, we achieve up to 5% further improvement in classification accuracy, without hand-tuned components and with a modest space penalty of up to 10%. Results are given in Table 2.

Table 2: Top-1 classification accuracy of ShuffleNet on ImageNet validation set (parameter counts in parentheses). We compare our approach (col. 3) with our reimplementation of 'vanilla' ShuffleNet (col. 1) and a recent approach based on the Hadamard transform (col. 2).[4] We report results for different network width multipliers (# channels). The last column shows the differences in accuracy and parameter count between our approach and vanilla ShuffleNet; using a learnable K-matrix in place of each fixed permutation (shuffle) or Hadamard matrix improves accuracy by up to 5%.

|  | Shuffle | Hadamard | Kaleidoscope (K.) | K. vs. Shuffle |
|---|---|---|---|---|
| 0.25 ShuffleNet g8 | 44.1% (0.46M) | 43.9% (0.46M) | **49.2**% (0.51M) | +5.0% (+0.05M) |
| 0.5 ShuffleNet g8 | 57.1% (1.0M) | 56.2% (1.0M) | **59.5**% (1.1M) | +2.4% (+0.1M) |
| 1.0 ShuffleNet g8 | 65.3% (2.5M) | 65.0% (2.5M) | **66.5**% (2.8M) | +1.2% (+0.2M) |

Grouped convolution (Krizhevsky et al., 2012) is often used to reduce parameter count and speed up inference compared to standard convolution, but, by default, channels in different groups cannot exchange information. To remedy this, ShuffleNet uses a permutation matrix to shuffle the channels after each grouped convolution. Zhao et al. (2019) propose to instead use the Hadamard transform before and after each grouped convolution to mix the channels. In place of these hand-engineered solutions, we use a K-matrix before and after each grouped convolution, and learn these end-to-end together with the rest of the network. As shown in Table 2, across a range of sizes, replacing the channel shuffles with K-matrices results in improved performance at comparable parameter counts.

### 3.2 LEARNING A LATENT PERMUTATION

We show that K-matrices can be used in a challenging task for which existing classes of structured linear maps have not been found suitable. We investigate the problem of image classification on a *permuted* image dataset (Permuted CIFAR-10). This problem is challenging due to the discrete nature of learning the latent permutation of the dataset; we present a differentiable relaxation for this using a K-matrix as a key component. Results are presented in Table 3; compared to methods that do

---

[4]Despite our best effort, we were unable to reproduce the original accuracy reported by Zhang et al. (2018), a problem similarly faced by Zhao et al. (2019) and Lyu et al. (2019). Zhao et al. (2019) use block Hadamard transform and pre-activation ShuffleNet, so their results are not directly comparable with those reported here.

not have a permutation learning step, our approach gets 9 points higher accuracy (84.4% to 93.6%), coming within 2 points of the accuracy on the un-permuted dataset (94.9%).

Table 3: Permuted CIFAR-10 validation set classification accuracy (%). Our kaleidoscope layer is able to nearly perfectly recover the latent structure, allowing a downstream CNN to approach the accuracy of a standard ResNet18 on the unpermuted dataset (last column).

| Model | FC | RNN | CNN | Dense + CNN | K + CNN | Unpermuted |
|---|---|---|---|---|---|---|
| Accuracy | 61.2 | 57.8 | 73.7 | 84.4 | **93.6** | 94.9 |

In this task, we use a permuted image classification dataset (Permuted CIFAR-10), wherein a fixed global permutation is applied to the pixels of every image in the original input set. Typically, only fully-connected (FC) and recurrent models are applied to such datasets (Le et al., 2015), because the permutation destroys locality in the image, presenting a difficulty for CNNs. However, CNNs are much better-suited for standard image tasks. We thus expect that learning the permutation and then applying a standard CNN should outperform these baselines. As mentioned in Section 2, the kaleidoscope hierarchy provides a nearly tight parameterization of permutations; this makes them a natural fit for the permutation learning step.

Experimentally, we use a K-matrix to represent a distribution over permutations, which converges to a single permutation at the end of training. The correct latent structure is learned by applying samples from this distribution to the permuted training images, and minimizing an auxiliary smoothness-based loss that encourages the reconstructed images to be more "natural" (i.e. vary smoothly pixel-to-pixel). The learned permutation is evaluated by training a ResNet18 with the K-matrix permutation layer inserted at the beginning. Full details of our approach are provided in Appendix B.3.

In Table 3, we compare our approach to a ResNet18 without this extra K-matrix layer, a ResNet18 with an extra dense matrix at the beginning instead of a K-matrix, and other baselines. As generic representations such as unstructured matrices do not have the requisite properties to fit in the pipeline, these baselines fail to effectively learn the latent permutation. We emphasize that a K-matrix provides this ability to recover latent structure despite not being specialized for permutations. Figure 3 describes the pipeline and displays examples of permuted and unpermuted images.

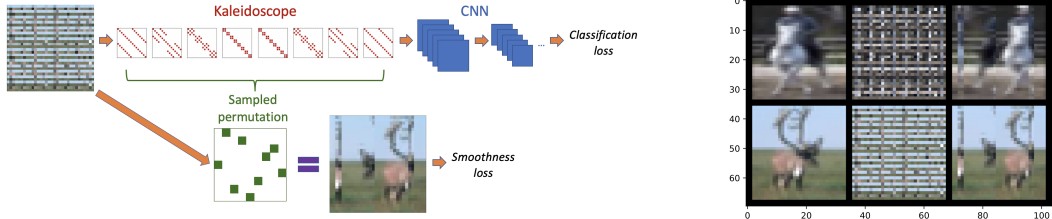

Figure 3: (a) (Left) Schematic describing permutation learning approach. The inputs are multiplied by a K-matrix and then fed into a CNN, from which the classification loss is computed. Separately, the input is permuted by a permutation matrix sampled from the distribution described by the K-matrix, and a "smoothness" loss (Rudin et al., 1992) is computed from the result, as described in Appendix B.3. (b) (Right) Left panel: original (unpermuted) example images. Center panel: the permuted versions. Right panel: these images after then applying the permutation recovered by the K-matrix. The K-matrix is able to nearly unscramble the images into their unpermuted versions.

## 3.3 SPEEDING UP INFERENCE

We evaluate the inference speed benefit of using K-matrices on a real language translation model. We choose the state-of-the-art DynamicConv Transformer translation model (Wu et al., 2019), which offers 20% inference speedup over the standard Transformer model, and replace dense matrices in the decoder's linear layers with K-matrices, which leads to a further 36% inference speedup (Table 4).

As outlined in Section 2.3, K-matrices admit a simple and fast $O(n \log n)$ matrix-vector multiplication algorithm. We provide fast implementations of this algorithm in C++ and CUDA, with an interface to PyTorch (Paszke et al., 2017), and use this implementation in our experiments.

Table 4: Inference speed on the IWSLT-14 German-English translation task (test set). Using K-matrices instead of dense matrices in the DynamicConv decoder linear layers results in 36% faster inference speed (measured on a single-threaded CPU with a batch size of 1 and beam size of 1).

| Model | # params | BLEU | Sentences/sec | Tokens/sec |
|---|---|---|---|---|
| Transformer (Vaswani et al., 2017) | 43M | 34.4 | 3.0 | 66.4 |
| DynamicConv Transformer (Wu et al., 2019) | 39M | **35.2** | 3.6 | 80.2 |
| DynamicConv Transformer w/ K-matrices (ours) | **30M** | 34.2 | **4.9** | **103.4** |

We use K-matrices to replace all the linear layers in the decoder of DynamicConv (since 90% of inference time is spent in the decoder). As shown in Table 4, on the IWSLT-14 German-English translation task, this yields a 25% smaller model with 36% faster inference time on CPU, at the cost of 1.0 drop in BLEU score.[5] (Our model also nearly matches the state-of-the-art BLEU performance of 2 years ago obtained by the Transformer model (Vaswani et al., 2017), despite being over 60% faster for inference than the Transformer.) The majority (55%) of inference time is spent in matrix-vector multiplication; our implementation of K-matrix-vector multiplication is about 2 times faster than the optimized implementation of dense matrix-vector multiplication in the Intel MKL library. Direct comparisons of K-matrix multiplication with this and other highly-optimized routines such as the FFT are further detailed in Appendix C.

## 4 CONCLUSION

We address the problem of having to manually choose among the numerous classes of structured linear maps by proposing the universal (expressive, efficient, and learnable) family of kaleidoscope matrices. We prove that K-matrices can represent any structured linear maps with near-optimal space and time complexity. Empirical validations suggest that K-matrices are a promising and flexible way to employ structure in modern ML; they can be used to reduce the need for hand-engineering, capture challenging latent structure, and improve efficiency in models. We are excited about future work on further hardware-optimized implementations of K-matrices, to fully realize the size and speed benefits of structured matrices on a broad array of real-world applications.

ACKNOWLEDGMENTS

We thank Avner May and Jian Zhang for their helpful feedback.

We gratefully acknowledge the support of DARPA under Nos. FA87501720095 (D3M), FA86501827865 (SDH), and FA86501827882 (ASED); NIH under No. U54EB020405 (Mobilize), NSF under Nos. CCF1763315 (Beyond Sparsity), CCF1563078 (Volume to Velocity), and 1937301 (RTML); ONR under No. N000141712266 (Unifying Weak Supervision); the Moore Foundation, NXP, Xilinx, LETI-CEA, Intel, IBM, Microsoft, NEC, Toshiba, TSMC, ARM, Hitachi, BASF, Accenture, Ericsson, Qualcomm, Analog Devices, the Okawa Foundation, American Family Insurance, Google Cloud, Swiss Re, and members of the Stanford DAWN project: Teradata, Facebook, Google, Ant Financial, NEC, VMWare, and Infosys. The U.S. Government is authorized to reproduce and distribute reprints for Governmental purposes notwithstanding any copyright notation thereon. Any opinions, findings, and conclusions or recommendations expressed in this material are those of the authors and do not necessarily reflect the views, policies, or endorsements, either expressed or implied, of DARPA, NIH, ONR, or the U.S. Government. Matthew Eichhorn and Atri Rudra's research is supported by NSF grant CCF-1763481.

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

# A  RELATED WORK

## A.1  STRUCTURED MATRICES IN MACHINE LEARNING

Structured linear maps such as the DFT, the Hadamard transform and convolution are a workhorse of machine learning, with diverse applications including data preprocessing, random projection, featurization, and model compression. For example, the DFT is a crucial step in the standard filter bank speech preprocessing pipeline (Jurafsky & Martin, 2014), and is commonly used when dealing with time series data in general (Panaretos & Tavakoli, 2013). Fast random projection and kernel approximation methods rely on the fast Hadamard transform (Le et al., 2013; Yu et al., 2016) and convolution (Yu et al., 2015), and convolution is a critical component of modern image processing architectures (Krizhevsky et al., 2012) as well as being useful in speech recognition (Zeghidour et al., 2018) and natural language processing (Wu et al., 2019). Large learnable classes of structured matrices such as Toeplitz-like matrices (Sindhwani et al., 2015) and low-displacement rank (LDR) matrices (Thomas et al., 2018) have been used for model compression. However, despite their theoretical speedup, these structured matrix classes lack efficient implementations, especially on GPUs. Therefore, their use has largely been confined to small models (e.g. single hidden layer neural nets) and small datasets (e.g. CIFAR-10).

Butterfly matrices encode the recursive divide-and-conquer structure of the fast Fourier transform (FFT) algorithm. They were first used in numerical linear algebra for fast preconditioning (Parker, 1995). The butterfly factorization is then generalized to encompass complementary low-rank matrices commonly encountered in solving differential and integral equations (Rokhlin & Tygert, 2006; Tygert, 2008; 2010b;a; Li et al., 2015; 2018). In machine learning, butterfly matrices have been use to approximate the Hessian for fast optimization (Mathieu & LeCun, 2014), and to perform fast random projection (Jing et al., 2017; Munkhoeva et al., 2018; Choromanski et al., 2019). Dao et al. (2019) show that butterfly matrices can be used to learn fast algorithms for discrete transforms such as the Fourier transform, cosine/sine transform, Hadamard transform, and convolution.

## A.2  SPARSE MATRICES

Several classes of structured linear transforms are ubiquitous in modern deep learning architectures; particularly widespread examples include convolution and multiheaded attention. Recently, attempts to impose *sparsity* on the neural network weights have been gaining traction. State-of-the art approaches of this type typically accomplish this by pruning small weights (either gradually during training (Zhu & Gupta, 2017), or post-training (Han et al., 2016)) or by training a dense network and then identifying "winning lottery tickets"—sparse subnetworks which may then be retrained from scratch with appropriate initialization (Frankle & Carbin, 2019). Importantly, these approaches start from a dense network, and therefore training is expensive. There is also a more nascent line of work that aims to train unstructured sparse neural networks *directly* (Mocanu et al., 2018; Mostafa & Wang, 2019; Dettmers & Zettlemoyer, 2019; Evci et al., 2019). These approaches maintain a constant network sparsity level throughout training, and use heuristics to evolve the sparsity pattern during training. One drawback is that the indices of the nonzero entries need to be stored in addition to the entry values themselves, which increases the memory required to store the sparse weight tensors. Another drawback is that these approaches to learn the sparsity pattern are based on intricate heuristics, which can be brittle. We note that these heuristic sparsification techniques could potentially be combined with our approach, to further sparsify the K-matrix factors.

## A.3  SPEECH RECOGNITION FROM RAW AUDIO

Numerous works focus on the problem of speech recognition from raw audio input, i.e. without manual featurization. SincNet (Ravanelli & Bengio, 2018) is a CNN-based architecture parameterized with sinc functions, designed so that the first convolutional layer imitates a band-pass filter. Zeghidour et al. (2018) formulate a learnable version of a filter bank featurization; their filters are initialized as an approximation of MFSC features and then fine-tuned jointly with the rest of the model. Sainath et al. (2015) proposed a powerful combined convolutional LSTM (CLDNN)-based model for learning from raw audio, using a large amount of training data. The WaveNet generative architecture (van den Oord et al., 2016), based on dilated convolutions, has been adapted to speech recognition and can be trained on raw audio. Other approaches that can learn from raw audio can be found in (Palaz et al.,

2013; Collobert et al., 2016; Ghahremani et al., 2016). To our knowledge, the 14.6% PER achieved by our kaleidoscope + LSTM model on the TIMIT test set is the lowest error rate obtained by a model trained directly on the raw audio.

### A.4 LEARNING PERMUTATIONS

Permutation matrices find use in tasks such as matching and sorting (among many others). Techniques to obtain posterior distributions over permutations have been developed, such as the exponential weights algorithm (Helmbold & Warmuth, 2009) and the Gumbel-Sinkhorn network (Mena et al., 2018).

Classifying images with permuted pixels is a standard task to benchmark the ability of RNNs to learn long range dependencies. Le et al. (2015) propose the Permuted MNIST task, in which the model has to classify digit images with all the pixels permuted. Many new RNN architectures, with unitary or orthogonal weight matrices to avoid gradient explosion or vanishing, have been proposed and tested on this task (Le et al., 2015; Arjovsky et al., 2016; Wisdom et al., 2016; Mhammedi et al., 2017; Trinh et al., 2018). Standard gated RNN architectures such as LSTM and GRU have also been found to be competitive with these new RNN architectures on this task (Bai et al., 2018).

## B ADDITIONAL EXPERIMENTAL DETAILS

### B.1 SPEECH PREPROCESSING

In this section, we fully describe our settings and procedures for the speech preprocessing experiments in Section 3.1.1, and present additional auxiliary baselines and results.

#### B.1.1 EXPERIMENTAL SETUP

We evaluate our speech recognition models on the TIMIT speech corpus (Garofolo et al., 1993), a standard benchmark for speech recognition. The input is audio (16-bit, 16 kHz .wav format), and the target is the transcription into a sequence of phonemes (units of spoken sound). Our evaluation metric is the phoneme error rate (PER) between the true phoneme sequence and the phoneme sequence predicted by our model. We use PyTorch (Paszke et al., 2017), the Kaldi speech recognition toolkit (Povey et al., 2011), and the PyTorch-Kaldi toolkit (Ravanelli et al., 2019) for developing PyTorch speech recognition models for all our experiments and evaluations.

#### B.1.2 MODEL AND EVALUATION

Our baseline Bi-LSTM architecture is taken from the PyTorch-Kaldi repository.[6] This is a strong baseline model that, to the best of our knowledge, matches state-of-the-art performance for models that use a *single* type of input featurization (Ravanelli et al., 2019). The original Bi-LSTM model takes as input filter bank features. These are computed as follows: (i) the waveform is framed (split into chunks of 25 ms each that overlap by 10 ms each), (ii) the waveform is dithered (zero-mean Gaussian random noise is added), (iii) pre-emphasis is applied to amplify high frequencies, (iv) the Hamming window function (Harris, 1978) is applied, (v) the FFT is applied, and the power spectrum of the resulting (complex-valued) output is computed, (vi) the power spectrum (which has dimension 512) is mapped to the "mel scale" (which is a scale intended to mimic human auditory perception (Stevens et al., 1937)) by multiplication with a specific banded matrix of dimension $512 \times 23$, and the entrywise logarithm of the output is taken (the 23 outputs are called the *filters*), and (vii) cepstral mean and variance normalization (Liu et al., 1993) is applied. Numerical hyperparameters of this procedure include the dither noise scale, the pre-emphasis coefficient, the Hamming window size, the number of mel filters, and more; we kept all these same as the Kaldi/PyTorch-Kaldi defaults.

In contrast, our "K-matrix version" of the model takes as input the raw waveform, split into chunks the same way as before but with no normalization, dithering, or other preprocessing, which is then fed into a complex-valued kaleidoscope $[(\mathcal{B}\mathcal{B}^*)^2]$ matrix. Similarly to the nonlinear steps in computing filter bank features, the logarithm of the power spectrum of the output (which has dimension 512)

---

[6]This open-source repository can be found at `https://github.com/mravanelli/pytorch-kaldi`.

is then computed. This output is fed into the Bi-LSTM; the Bi-LSTM and kaleidoscope layer are trained together in standard end-to-end fashion. The Bi-LSTM architecture is not modified aside from changing the input dimension from 23 to 512; this (along with the $\approx 75$K parameters in the kaleidoscope layer itself) results in approximately a 1.1M increase in the total number of parameters compared to the model that takes in MFSC features (a modest 8% relative increase). Total training time for our kaleidoscope-based architecture is 7% greater than that required for the model that uses MFSC features, not counting the time required to precompute the MFSC features; the FLOPs for inference-time are approximately 15% greater (mostly due to the larger dimension of the input to the Bi-LSTM; the kaleidoscope layer accounts for less than 0.5% of the total FLOPs).

As baselines, we also compare to inserting other types of linear transformations before the Bi-LSTM: fixed linear transformations (such as the fixed FFT, or no transform at all [i.e. the identity]), other trainable structured layers (low-rank, circulant, and sparse [using the sparse training algorithm of Dettmers & Zettlemoyer (2019)]), and a trainable *unstructured* (dense) linear layer. The kaleidoscope layer performs the best out of all such approaches. The fact that it outperforms even a dense linear layer with more parameters is particularly notable, as it suggests that the structural bias imposed by the K-matrix representation is beneficial for performance on this task. Full results are given in Table 5.

Table 5: TIMIT phoneme error rate (PER%, $\pm$ standard deviation across 5 random seeds).

| Model | Test set PER% | # Parameters |
|---|---|---|
| Low rank + LSTM | $23.6 \pm 0.9$ | 15.5M |
| Sparse + LSTM | $21.7 \pm 0.9$ | 15.5M |
| Circulant + LSTM | $23.9 \pm 0.9$ | 15.4M |
| Dense + LSTM | $15.4 \pm 0.6$ | 15.9M |
| FFT + LSTM | $15.7 \pm 0.1$ | 15.4M |
| Identity + LSTM | $20.7 \pm 0.3$ | 15.4M |
| *Kaleidoscope + LSTM* | $14.6 \pm 0.3$ | 15.4M |
| MFSC features + LSTM | $14.2 \pm 0.2$ | 14.3M |
| SincNet (Ravanelli et al., 2019) | 17.2 | 10.0M |
| LiGRU (Ravanelli et al., 2018) | 13.8 | 12.3M |

In our experiments, we grid search the initial learning rate for the "preprocessing layer" (if applicable) in {5e-5, 1e-4, 2e-4, 4e-4, 8e-4, 1.6e-3}, and fix all other hyperparameters (including the initial learning rates for the other parts of the network) to their default values in the PyTorch-Kaldi repository. The model and any preprocessing layers are trained end-to-end with the RMSProp optimizer for 24 epochs (as per the defaults in PyTorch-Kaldi). For each model, we use the validation set to select the best preprocessing learning rate, while the final error rates are reported on the separate held-out test set. For all structured matrix baselines except circulant (which always has $n$ parameters for an $n \times n$ matrix), the number of parameters in the structured matrices is set to equal the number of parameters in the butterfly layer, while the unconstrained matrix is simply a standard dense complex-valued square matrix. For all experiments with a trainable "preprocessing layer," we initialize the preprocessing matrix to represent the FFT (or approximate it as closely as possible [i.e. minimize the Frobenius error to the true FFT matrix], in the case of low-rank, sparse, and circulant), which we found to outperform random initialization.

### B.1.3 Extension: Combining MFSC and kaleidoscope

As an additional experiment, we sought to investigate whether *combining* the hand-engineered MFSC featurization pipeline and a learnable kaleidoscope layer (instead of replacing the former with the latter) could lead to accuracy gains. Specifically, in this experiment we first used the standard filter bank featurization pipeline described above, and trained end-to-end as usual. Then, we replaced the FFT step with a K-matrix initialized to the FFT, and made the weights of the Hamming window function and the mel filter bank matrix learnable as well (similarly to (Zeghidour et al., 2018)). We fine-tuned the resulting architecture for an additional 10 epochs. The final test PER% attained by this "hybrid" model is **14.0** $\pm$ 0.3; the model has 14.4M parameters—a negligible increase over the 14.3M in the original architecture. Thus, by combining the manually encoded domain knowledge in the filter bank featurization and allowing this structure to be *learnable* rather than fixed, we are able

to nearly match the state-of-the-art 13.8% accuracy on TIMIT. While this "hybrid" model certainly involves some hand-engineering, the state-of-the-art results use a concatenation of *three* different speech audio featurizations—MFSC, MFCC, and fMLLR—as the neural network input, along with a customized RNN architecture (LiGRU) specifically designed for speech recognition, and thus require a more complicated pipeline that is arguably even more hand-crafted.

### B.2 REPLACING CNN CHANNEL SHUFFLE

#### B.2.1 MODEL ARCHITECTURES

ShuffleNet is a convolutional neural network with residual (skip) connections that uses a permutation matrix to shuffle the channels after each grouped 1x1 convolution, sending the $i$-th channel to the $(i \bmod g)$-th group, where $g$ is the total number of groups. The architecture for each residual block in ShuffleNet is: 1x1 group conv → Batch norm, ReLU → Permutation → 3x3 depthwise conv → Batch norm → 1x1 group conv. The permutation is fixed.

Zhao et al. (2019) propose to instead use the Hadamard transform before and after each grouped 1x1 convolution to mix the channels. Note that the Hadamard transforms are placed *before* the batch normalization and ReLU layer (unlike the permutation matrix in the original ShuffleNet design). In particular, the architecture for each block is: Hadamard → 1x1 group conv → Hadamard → Batch norm, ReLU → 3x3 depthwise conv → Batch norm → 1x1 group conv. The Hadamard transform is fixed.

In our architecture, we use a kaleidoscope matrix in $\mathcal{OBB}$ (product of an orthogonal butterfly matrix, a diagonal matrix, and the transpose of another butterfly matrix) before and after each grouped 1x1 convolution. We place the second K-matrix after the batch norm and ReLU, to more closely mimic the original ShuffleNet design. The structure for each block is: K-matrix → 1x1 group conv → Batch norm, ReLU → K-matrix → 3x3 depthwise conv → Batch norm → 1x1 group conv. The K-matrices are trained along with the rest of the network, rather than being fixed.

#### B.2.2 EXPERIMENTAL SETUP

We evaluate the CNN architectures on the image classification task of the standard ImageNet dataset (Russakovsky et al., 2015). We use the standard data augmentation, training, and evaluation pipeline as in (Xie et al., 2017). We train with SGD on 8 GPUs for 90 epochs, with a total batch size of 2048 and initial learning rate 0.8. For the 1.0 ShuffleNet g8 architecture, we reduce the total batch size to 1792 to fit into GPU memory, and correspondingly linearly scale the initial learning rate to 0.7. Other hyperparameters (e.g. learning rate schedule, weight decay, etc.) are kept the same as in the ShuffleNet paper (Zhang et al., 2018). We use the training script from NVIDIA's deep learning examples repository.[7]

#### B.2.3 ADDITIONAL RESULTS

In Table 6, we report top-5 classification accuracy on ImageNet, to complement the top-1 accuracies in Table 2.

Table 6: Top-5 classification accuracy of ShuffleNet on ImageNet validation set. We report results for different network width multipliers (number of channels), and for different kinds of matrices used for channel mixing. Using a learnable K-matrix in place of each fixed permutation (shuffle) or Hadamard matrix improves top-5 accuracy by up to 4.8%. Parameter counts are the same as in Table 2.

|  | Shuffle | Hadamard | Kaleidoscope (K.) | K. vs. Shuffle |
|---|---|---|---|---|
| 0.25 ShuffleNet g8 | 68.6% | 68.4% | **73.4%** | +4.8% |
| 0.5 ShuffleNet g8 | 79.9% | 79.2% | **81.7%** | +1.8% |
| 1.0 ShuffleNet g8 | 86.0% | 85.8% | **86.8%** | +0.8% |

---

[7]`https://github.com/NVIDIA/DeepLearningExamples/tree/master/PyTorch/Classification/RN50v1.5`

In each setting, the total training time of our K-matrix approach is within 20% of the total training time of vanilla ShuffleNet.

In Figure 4, we plot the loss and accuracy on the training set and validation set when we train 1.0 ShuffleNet g8, with either a fixed permutation (Shuffle) or a K-matrix for channel shuffling. Even though each K-matrix is a product of multiple (sparse) matrices, the model with K-matrices takes about the same number of training steps to converge as the baseline model does. One possible reason is that we constrain the K-matrices to be orthogonal (Section 2.4), thus avoiding vanishing or exploding gradients.

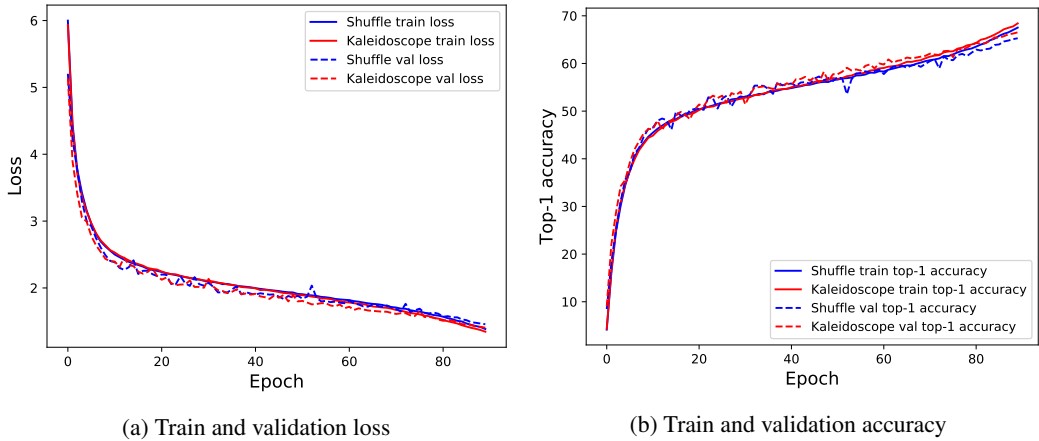

(a) Train and validation loss                     (b) Train and validation accuracy

Figure 4: Loss and top-1 accuracy of 1.0 ShuffleNet g8 with either a fixed permutation (Shuffle) or a K-matrix for channel shuffling. The K-matrix model takes about the same number of training steps to converge as does the baseline model.

### B.3    LEARNING PERMUTATIONS

#### B.3.1    DATASET

The permuted CIFAR-10 dataset is constructed by applying a fixed permutation to every input. We choose to use the 2-D bit-reversal permutation,[8] i.e., the bit reversal permutation on 32 elements is applied to the rows and to the columns. This permutation was chosen because it is locality-destroying: if two indices $i, j$ are close, they must differ in a lower-order bit, so that the bit-reversed indices $i', j'$ are far. This makes it a particularly challenging test case for architectures that rely on spatial locality such as "vanilla" CNNs.

#### B.3.2    MODEL AND TRAINING

We describe the model architectures used in Section 3.1 (those reported in Table 3).

**Our model (K + CNN)**    The model represents a fixed permutation $P$, parametrized as a K-matrix, to learn to recover the true permutation, followed by a standard ResNet18 architecture (He et al., 2016). Because of the simple decomposable nature of the butterfly factors (Section 2.1), our parameterization is easily extensible with additional techniques:

    (i) We constrain each butterfly factor matrix in the K-matrix to be doubly-stochastic. For example, each $2 \times 2$ block in the butterfly factor matrix of block size 2 has the form $\begin{bmatrix} a & 1-a \\ 1-a & a \end{bmatrix}$, where $a \in [0, 1]$. We treat this block as a *distribution* over permutations,

---

[8]The bit-reversal permutation reverses the order of the bits in the binary representation of the indices. For example, indices [0, 1, ..., 7] with binary representations [000, 001, ..., 111] are mapped to [000, 100, ..., 111], which corresponds to [0, 4, 2, 6, 1, 5, 3, 7]

generating the identity $\begin{bmatrix} 1 & 0 \\ 0 & 1 \end{bmatrix}$ with probability $a$ and the swap $\begin{bmatrix} 0 & 1 \\ 1 & 0 \end{bmatrix}$ with probability $1-a$. Butterfly factor matrices with larger block sizes are constrained to be doubly-stochastic in a similar manner. In this way, a permutation is sampled for each butterfly factor matrix, and these permutations are composed to get the final permutation that is applied to the image.

(ii) For each minibatch, the examples $Px$ by applying permutation samples on the (permuted) inputs are fed into an additional unsupervised reconstruction loss

$$\sum_{0 \leq i,j < n} \left\| \begin{bmatrix} (Px)[i+1,j] - (Px)[i,j] \\ (Px)[i,j+1] - (Px)[i,j] \end{bmatrix} \right\|_2 \tag{1}$$

measuring total variation smoothness of the de-noised inputs. Such loss functions are often used in image denoising (Rudin et al., 1992). A final regularization loss was placed on the entropy of $P$, which was annealed over time to encourage $P$ to converge toward a sharper doubly-stochastic matrix (in other words, a permutation).

The model is trained with just the reconstruction loss to convergence before the standard ResNet is trained on top.

These techniques are applicable to the K-matrix as well as specialized methods for representing permutations such as Gumbel-Sinkhorn (Mena et al., 2018) and are important for recovering the true permutation. However, they are not applicable to a general linear layer, which showcases the flexibility of K-matrices for representing generic structure despite not being specially tailored for this task. We also remark that other classes of structured linear maps such as low-rank, circulant, and so on, are even less suited to this task than dense matrices, as they are incapable of representing all permutations.

**Baseline architectures**

1. Fully connected (FC): This is a 3-layer MLP, with hidden size 1024 and ReLU nonlinearity in-between the fully connected layers.

2. Recurrent neural network (RNN): We use a gated recurrent unit (GRU) model (Cho et al., 2014), with hidden size 1024. Many RNN architectures have been proposed to capture long-range dependency on permuted image dataset such as Permuted MNIST (Arjovsky et al., 2016). Standard gated architectures such as LSTM and GRU have shown competitive performance on the Permuted MNIST dataset, and we choose GRU as a baseline since it has been reported to slightly outperform LSTM (Bai et al., 2018).

3. CNN: We use the standard ResNet18 architecture, adapted to smaller image size of the CIFAR-10 dataset (changing stride from 2 to 1 of the first convolutional layer, and removing max-pooling layer that follows).

4. Dense + CNN: We add an additional linear layer (i.e. a dense matrix) of size $1024 \times 1024$ before the ResNet18 architecture. This dense layer can in theory represent a permutation, but cannot benefit from the additional techniques described above.

5. Baseline CNN (unpermuted): We use the standard ResNet18 architecture applied to the unpermuted CIFAR-10 dataset.

All models are trained for 200 total epochs, with the Adam optimizer. We use the standard learning rate schedule and weight decay from Mostafa & Wang (2019). We use Hyperband (Li et al., 2017) to tune other hyperparameters such as the initial learning rate and annealing temperature.

### B.4 Speeding up DynamicConv's inference

#### B.4.1 Model architecture

We start with the DynamicConv Transformer architecture (Wu et al., 2019), which is a variant of the Transformer architecture (Vaswani et al., 2017) where the self-attention in each layer is replaced with a light-weight DynamicConv module. We use the implementation from the Fairseq library(Ott et al., 2019),[9] with PyTorch version 1.2.

---

[9]This library can be found at `https://github.com/pytorch/fairseq`

The architecture of each layer of the decoder is: Linear → DynamicConv → Linear → LayerNorm → Encoder-decoder attention → LayerNorm → Linear → ReLU → Linear → ReLU → LayerNorm. In every layer of the decoder, we replace the dense weight matrix in each of the four Linear layers with a K-matrix from the $\mathcal{B}$ class (i.e. a butterfly matrix).

### B.4.2 TRAINING AND EVALUATION

The models are trained from scratch using the training script from the Fairseq repository, with the same hyperparameters (optimizer, learning rate, number of updates, etc.) used in the DynamicConv paper (Wu et al., 2019). We note that the DynamicConv model with K-matrices in the decoder trains slightly faster than the default DynamicConv model (both models are trained for 50,000 updates, which requires approximately 7% less time for the K-matrix model than for the default model).

To evaluate inference speed, we run the decoding script on the IWSLT-14 De-En test set in single-threaded mode on a server Intel Xeon CPU E5-2690 v4 at 2.60GHz, and measure wall-clock time. The test set contains 6750 sentences, with 149241 tokens. Following Wu et al. (2019), we set the batch size to 1 and beam size to 1 for this evaluation.

### B.4.3 ADDITIONAL COMPARISON WITH OTHER STRUCTURED MATRICES

We additionally compare the speed-quality tradeoff of K-matrices with other classes of structured matrices, when used to replace the fully-connected layers of DynamicConv's decoder. We consider the following additional classes of structured matrices: low-rank, circulant, Toeplitz-like (Sindhwani et al., 2015), ACDC (Moczulski et al., 2016), Fastfood (Le et al., 2013), and sparse. For classes with a variable number of parameters (e.g. low-rank, sparse), we set the number of parameters to match that of K-matrices. For sparse matrices, besides the result for an ensemble of 10 models (the default setting in the Fairseq repository), we also report the result for a single model, as that could have faster inference time (since ensembling/averaging sparse matrices produces a less sparse matrix).

In Figure 5, we plot the tradeoff between translation quality (measured by BLEU score) and inference speed (sentences per second). Most classes of structured matrices produce similar translation quality (between 34.1 and 34.4 BLEU score). K-matrices have the second fastest inference time, only 7% slower than low-rank matrices. We note that low-rank matrices benefit from very well-tuned BLAS routines (matrix-matrix multiplication). Even though our implementation of K-matrix multiplication is not yet highly optimized, it is already quite close to the speed of low-rank matrix multiplication at an equivalent parameter count.

## C SPEED BENCHMARK AND IMPLEMENTATION DETAILS

Each K-matrix (for fixed width and expansion), has an $O(n \log n)$ matrix-vector multiplication algorithm: sequentially multiply the input vector with each of the sparse factors. Our implementation of this simple algorithm is surprisingly competitive with optimized subroutines, both on GPU (e.g. for training) and on CPU (e.g. for inference). In Figure 6, we compare the speed of multiplying by a K-matrix in class $\mathcal{B}$ (i.e. a butterfly matrix) against a specialized implementation of the FFT. We normalize the speed by the speed of dense matrix-matrix multiply (on GPU) or dense matrix-vector multiply (on CPU). On GPU, with input sizes $n = 1024$ and batch size 2048, the training time (forward and backward) of K-matrices matrix is about 3x faster than dense matrix multiply (GEMM from cuBLAS). For inference on CPU, the kaleidoscope fast multiplication can be one or two orders of magnitude faster than GEMV. Over a range of matrix sizes, our implementation is within a factor of 2-4x of specialized implementations of the FFT, a highly optimized kernel.

Our implementation is also memory-efficient. In the forward pass through the $O(\log n)$ sparse factors, we do not store the intermediate results, but recompute them during the backward pass. Therefore the activation memory required is $O(bn)$ for an input batch size of $b$.

## D SYNTHETIC MATRIX RECOVERY

We directly validate Theorem 1 on well-known types of structured matrices used in machine learning. Given a structured matrix $\mathbf{M}$, we attempt to represent $\mathbf{M}$ as closely as possible using K-matrices as

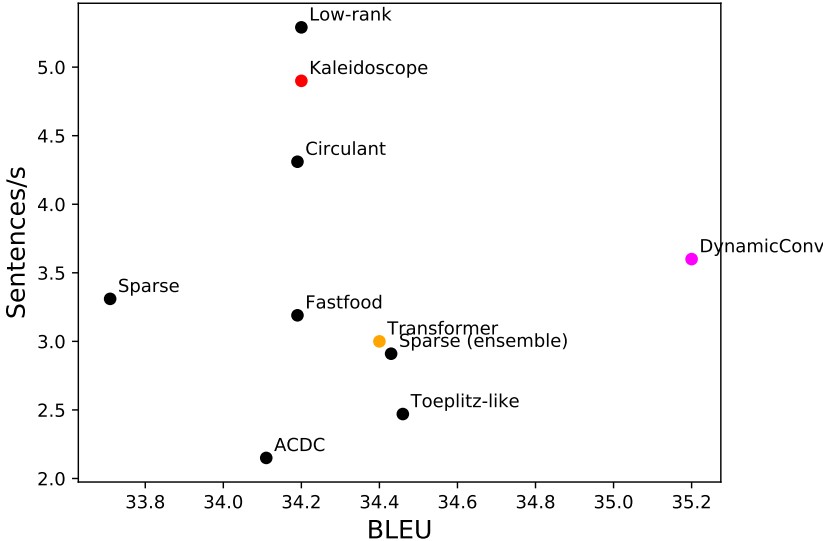

Figure 5: Tradeoff between translation quality (measured by BLEU score) and inference speed (sentences per second). K-matrices have the second fastest inference speed, only 7% slower than low-rank matrices.

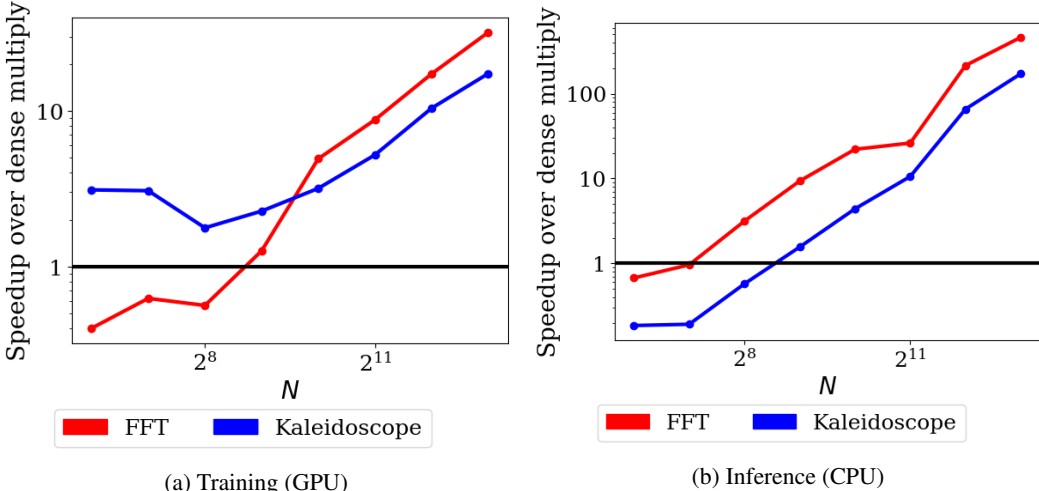

(a) Training (GPU)   (b) Inference (CPU)

Figure 6: Speedup of FFT and Kaleidoscope against dense matrix-matrix multiply (GEMM) for training, and against dense matrix-vector multiply (GEMV) for inference.

well as the standard classes of structured matrices: sparse and low-rank. In Table 7, we quantify the expressivity of each of these three methods, as measured by their ability to approximate a range of different structures. Results for "global minimum" of kaleidoscope matrices are obtained from the theoretical expressiveness results in Section I and Section J. Low-rank and sparse approximation have closed form solutions: truncating the SVD and keeping the largest-magnitude entries, respectively. We also report the results using SGD for kaleidoscope matrices to validate that good approximation with K-matrices can be obtained even from standard first-order optimization algorithms. Even with imperfect optimization, kaleidoscope matrices can still capture out-of-class target matrices better than low-rank and sparse matrices.

The target matrices are kaleidoscope, low-rank, sparse, convolution (i.e. circulant matrices), Fast-food (Le et al., 2013), and entrywise random IID Gaussian matrix (to show the typical magnitude of the error). All target matrices $\mathbf{M}$ were randomly initialized such that $\mathbb{E}[\mathbf{M}^T\mathbf{M}] = \mathbf{I}$.

Table 7: **Expressiveness of different classes of structured matrices**: Frobenius error of representing common structured matrices (columns) of dimension 256 using three structured representations of matrices with adjustable numbers of parameters. (Left group: Target matrices in the same class as the methods. Middle group: Target matrices with fixed number of parameters. Right: Random matrix to show typical scale of error.) Each method is allotted the same number of parameters, equal to a $\log n$ factor more than that of the target matrix. Low-rank and sparse matrices are unable to capture any structure outside their own class, while the minima for kaleidoscope matrices found via optimization better capture the actual structure for out-of-class targets better than the baselines.

| | Method $\diagdown$ Target | Kaleidoscope | Low-rank | Sparse | Convolution | Fastfood | Random |
|---|---|---|---|---|---|---|---|
| Global Min. | Kaleidoscope | 0.0 | 0.0 | 0.0 | 0.0 | 0.0 | |
| | Low-rank | 14.9 | 0.0 | 10.8 | 14.6 | 11.6 | 15.5 |
| | Sparse | 11.7 | 12.2 | 0.0 | 13.1 | 7.1 | 14.1 |
| With SGD | Kaleidoscope | 0.0 | 0.01 | 8.0 | **0.0** | **5.1** | 14.5 |

To find a kaleidoscope approximation with SGD, we used Hyperband to tune its learning rate (between 0.001 and 0.5).

# E  PROPERTIES OF THE $\mathcal{BB}^*$ HIERARCHY

Here, we justify why the definitions in Section 2.2 give rise to a hierarchy. We first make some basic observations about the parameterization.

**Observation E.1.** *An $n \times n$ matrix $\mathbf{M} \in \mathcal{BB}^*$ has $4n \log n$ parameters.*

*Proof.* $\mathbf{M}$ can be expressed as a product of $2 \log n$ butterfly factor matrices of size $n \times n$. Each of these factor matrices has 2 parameters per row, for a total of $2n$ parameters each. Hence, the total number of parameters is $4n \log n$. $\square$

**Observation E.2.** *Let $\mathbf{M}$ be an $n \times n$ matrix in $(\mathcal{BB}^*)_e^w$. Then, given an arbitrary vector $\mathbf{v}$ of length $n$, we can compute $\mathbf{Mv}$ with $O(wne \log(ne))$ field operations.*

*Proof.* Since $\mathbf{M} \in (\mathcal{BB}^*)_e^w$, we can decompose it as $\mathbf{SE}_1\mathbf{E}_2 \ldots \mathbf{E}_w\mathbf{S}^T$, where $\mathbf{S}$ is as given in Definition 2.4, and each $\mathbf{E}_i$ is an $en \times en$ matrix in $\mathcal{BB}^*$. Therefore, to compute $\mathbf{Mv}$, we can use associativity of matrix multiplication to multiply the vector by one of these matrices at a time.

Since all of these factors are sparse, we use the naïve sparse matrix-vector multiplication algorithm (begin with a 0-vector and perform the corresponding multiplication and addition for each nonzero matrix entry). $\mathbf{S}$ (and thus $\mathbf{S}^T$) have $n$ NNZ. Therefore, matrix-vector multiplication by $\mathbf{S}$ or $\mathbf{S}^T$ requires $O(n)$ operations, which is dominated by the butterfly matrix-vector multiplication. Each $\mathbf{E}_i$ can be further decomposed into $2 \log(ne)$ matrices with at most $2ne$ non-zero entries each (by Observation E.1). Therefore, matrix vector multiplication by each $\mathbf{E}_i$ requires $O(ne \log(ne))$. Since there are $w$ such $\mathbf{E}_i$, we require a total of $O(wne \log(ne))$ operations. $\square$

Now, we are ready to show that our definition of classes $(\mathcal{BB}^*)_e^w$ forms a natural hierarchy.

First, we must argue that all matrices are contained within the hierarchy.

**Lemma E.3.** *Let $\mathbf{M}$ be an arbitrary $n \times n$ matrix. Then $\mathbf{M} \in (\mathcal{BB}^*)^{(2n-2)}$.*

*Proof.* Corollary E.3 in Appendix K shows that any $n \times n$ matrix can be written in the form $\mathbf{M}_1\mathbf{M}_1'^* \ldots \mathbf{M}_{n-1}\mathbf{M}_{n-1}'^*\mathbf{MM}_n\mathbf{M}_n'^* \ldots \mathbf{M}_{2n-2}\mathbf{M}_{n-2}'^*$, where $\mathbf{M}_i, \mathbf{M}_i'$ are orthogonal butterfly matrices and $\mathbf{M}$ is a diagonal matrix. We can combine $D$ with $M_n$ to form another (possibly not orthogonal) butterfly matrix. This yields a decomposition of $\mathbf{M}$ as products of (possibly not orthogonal) butterfly matrices and their (conjugate) transposes, completing the proof. $\square$

Next, we argue that, up to a certain point, this hierarchy is strict.

**Lemma E.4.** *For every fixed $c \geq 1$, there is an $n \times n$ matrix $\mathbf{M}_n$ (with $n$ sufficiently large) such that $\mathbf{M}_n \in (\mathcal{BB}^*)^{c+1}$ but $\mathbf{M}_n \notin (\mathcal{BB}^*)^c$.*

*Proof.* Given $c$, fix $n$ to be a power of 2 such that $c < \frac{n}{4\log_2 n}$. For sake of contradiction, assume that every $n \times n$ matrix in $(\mathcal{BB}^*)^{c+1}$ is also in $(\mathcal{BB}^*)^c$. Let $\mathbf{A}$ be an arbitrary $n \times n$ matrix. From Lemma E.3, $\mathbf{A} \in (\mathcal{BB}^*)^{(2n-2)}$. From our assumption, we can replace the first $c + 1$ $\mathcal{BB}^*$ factors of $\mathbf{A}$ with $c$ (potentially different) $\mathcal{BB}^*$ factors and still recover $\mathbf{A}$. We can repeat this process until we are left with $c$ $\mathcal{BB}^*$ factors, implying that $\mathbf{A} \in (\mathcal{BB}^*)^c$. From Observation E.1, we require $4cn\log n < n^2$ (by our choice of $n$) parameters to completely describe $\mathbf{A}$. This is a contradiction since $\mathbf{A}$ is an arbitrary $n \times n$ matrix, and therefore has $n^2$ arbitrary parameters. Hence, there must be some $n \times n$ matrix in $(\mathcal{BB}^*)^{c+1}$ that is not in $(\mathcal{BB}^*)^c$. $\square$

# F   ARITHMETIC CIRCUITS IN $\mathcal{BB}^*$ HIERARCHY

In this appendix, we prove our main theoretical result, namely, our ability to capture general transformations, expressed as low-depth linear arithmetic circuits, in the $\mathcal{BB}^*$ hierarchy. This result is recorded in Theorem 1.

**Theorem 1.** *Let $\mathbf{M}$ be an $n \times n$ matrix such that matrix-vector multiplication of $\mathbf{M}$ times an arbitrary vector $\mathbf{v}$ can be represented as a be a linear arithmetic circuit $C$ comprised of $s$ gates (including inputs) and having depth $d$. Then, $\mathbf{M} \in (\mathcal{BB}^*)_{O(\frac{s}{n})}^{O(d)}$.*

To prove Theorem 1, we make use of the following two theorems.

**Theorem 2.** *Let $\mathbf{P}$ be an $n \times n$ permutation matrix (with $n$ a power of 2). Then $\mathbf{P} \in \mathcal{BB}^*$.*

**Theorem 3.** *Let $\mathbf{S}$ be an $n \times n$ matrix of $s$ NNZ. Then $\mathbf{S} \in (\mathcal{BB}^*)_4^{4\lceil \frac{s}{n} \rceil}$.*

Theorem 2 is proven in Appendix G, and Theorem 3 is proven in Appendix I.

*Proof of Theorem 1.* We will represent $C$ as a product of $d$ matrices, each of size $s' \times s'$, where $s'$ is the smallest power of 2 that is greater than or equal to $s$.

To introduce some notation, define $w_1, \ldots w_d$ such that $w_k$ represents the number of gates in the $k$'th layer of $C$ (note that $s = n + \sum_{k=1}^d w_k$). Also, define $z_1, \ldots z_d$ such that $z_1 = n$ and $z_k = w_{k-1} + z_{k-1}$ ($z_k$ is the number of gates that have already been used by the time we get to layer $k$).

Let $g_i$ denote the $i$'th gate (and its output) of $C$ ($0 \leq i < s$), defined such that:

$$g_i = \begin{cases} v_i & 0 \leq i < n \\ \alpha_j g_{i_1} + \beta_i g_{i_2} & n \leq i < s \end{cases}$$

where $i_1, i_2$ are indices of gates in earlier layers.

For the $k$'th layer of $C$, we define the $s' \times s'$ matrix $\mathbf{M}_k$ such that it performs the computations of the gates in that layer. Define the $i$'th row of $\mathbf{M}_k$ to be:

$$\mathbf{M}_k[i :] = \begin{cases} \mathbf{e}_i^T & 0 \leq i < z_k \\ \alpha_i \mathbf{e}_{i_1}^T + \beta_i \mathbf{e}_{i_2}^T & z_k \leq i < z_k + w_k \\ 0 & i \geq z_k + w_k \end{cases}$$

For any $0 \leq k \leq d$, let $\mathbf{v_k}$ be vector

$$\mathbf{v}_k = \mathbf{M}_k \ldots \mathbf{M}_2 \mathbf{M}_1 \begin{bmatrix} \mathbf{v} \\ 0 \end{bmatrix}.$$

We'd like to argue that $\mathbf{v}_d$ contains the outputs of all gates in $C$ (i.e, the $n$ values that make up $\mathbf{Mv}$). To do this we argue, by induction on $k$, that $\mathbf{v}_k$ is the vector whose first $z_{k+1}$ entries are $g_0, g_1, \ldots, g_{(z_k-1)}$, and whose remaining entries are 0. The base case, $k = 0$ is trivial. Assuming this holds for the case $k - 1$, and consider multiplying $\mathbf{v}_{k-1}$ by $\mathbf{M}_k$. The first $z_k$ rows of $\mathbf{M}_k$ duplicate

the first $z_k$ entries of $\mathbf{v}_{k-1}$ The next $w_k$ rows perform the computation of gates $g_{z_k}, \ldots, g_{(z_{k+1}-1)}$. Finally, the remaining rows pad the output vector with zeros. Therefore, $\mathbf{v}_k$ is exactly as desired.

The final matrix product will contain all $n$ elements of the output. By left multiplying by some permutation matrix $\mathbf{P}$, we can reorder this vector such that the first $n$ entries are exactly $\mathbf{Mv}$. Hence, we are left to argue the position of $\mathbf{PM}_d \ldots \mathbf{M}_2 \mathbf{M}_1$ within the $\mathcal{BB}^*$ hierarchy. Each $\mathbf{M}_k$ is a matrix with total $2w_k + z_k < 2s'$ NNZ. From Theorem 3, we can, therefore, represent $\mathbf{M}_k$ as a product of $O(1)$ matrices (of size $2s'$) in $\mathcal{BB}^*$. From Theorem 2, $\mathbf{P} \in \mathcal{BB}^*$. Note that $s \leq s' < 2s$, so $s' = \Theta(s)$.

Our final decomposition will have $O(d)$ $\mathcal{BB}^*$ factors, and requires an expansion from size $n$ to size $2s'$, or an expansion factor of $O(\frac{s}{n})$. Therefore, $\mathbf{M} \in (\mathcal{BB}^*)_{O(\frac{s}{n})}^{O(d)}$, as desired. $\qquad\square$

**Remark F.1.** *By applying Observation E.2, we see that Theorem 1 gives an $O(sd \log s)$ matrix vector multiplication algorithm for $\mathbf{M}$.*

# G   PERMUTATIONS IN $\mathcal{BB}^*$

In this appendix, we prove Theorem 2. In addition, we will also show that permutations are in $\mathcal{B}^*\mathcal{B}$, where the set $\mathcal{B}^*\mathcal{B}$ is defined analogously to $\mathcal{BB}^*$ (i.e. matrices of the form $\mathbf{M} = \mathbf{M}_1^*\mathbf{M}_2$ for some $\mathbf{M}_1, \mathbf{M}_2 \in \mathcal{B}$).

To prove Theorem 2, we decompose permutation matrix $\mathbf{P}$ into $\mathbf{P} = \mathbf{LR}$, with $\mathbf{L} \in \mathcal{B}$ and $\mathbf{R} \in \mathcal{B}^*$. Throughout the proof, we make use of the following definition.

**Definition G.1.** *Let $\mathbf{L}$ be an $n \times n$ permutation matrix ($n$ a power of 2). We say that $\mathbf{L}$ meets the $2^j$ **balance condition** if $\mathbf{L}$ can be divided into chunks of $2^j$ (with each chunk having all columns $i$ such that $\lfloor \frac{i}{2^j} \rfloor$ has the same value) such that for every $0 \leq m < 2^j$, each chunk has exactly one $\mathbf{L}[:, k] = \mathbf{e}_{\pi_k}$ with $\pi_k \equiv m \ (\mod 2^j)$. We say that $\mathbf{L}$ is **modular-balanced** if it meets the $2^j$ balance condition for each $2 \leq 2^j \leq n$.*

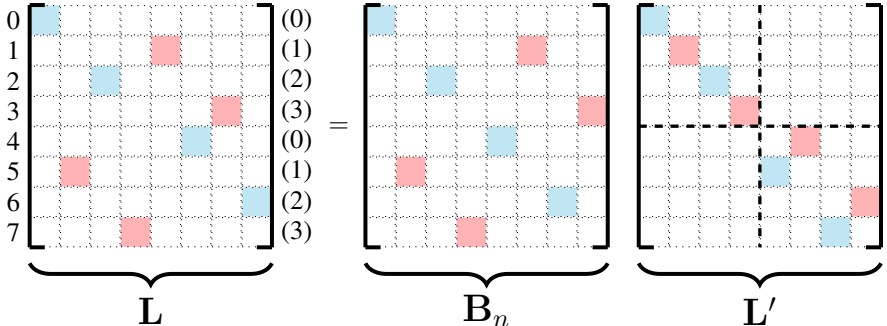

Figure 7: First step of decomposition of modular-balanced matrix $\mathbf{L}$. Here, the red entries must be permuted into the main diagonal blocks.

**Lemma G.1.** *Let $\mathbf{L}$ be an $n \times n$ modular-balanced matrix. Then $\mathbf{L} \in \mathcal{B}$.*

*Proof.* We proceed by induction on $n$. The base case $n = 2$ is trivial. As our inductive hypothesis, we assume that all modular-balanced matrices of size $\frac{n}{2} \times \frac{n}{2}$ are butterfly matrices of size $\frac{n}{2}$. From Definition 2.3, it is sufficient to show that $\mathbf{L}$ can be decomposed as:

$$\mathbf{L} = \mathbf{B}_n \underbrace{\begin{bmatrix} \mathbf{L}_1 & 0 \\ 0 & \mathbf{L}_2 \end{bmatrix}}_{\mathbf{L}'},$$

where $\mathbf{B}_n$ is a butterfly factor of size $n$ and each $\mathbf{L}_j$ is an $\frac{n}{2} \times \frac{n}{2}$ modular-balanced matrix.

Define $\mathbf{L}_1$ and $\mathbf{L}_2$ such that:

$$\mathbf{L}_1[i,j] = \mathbf{L}[i,j] + \mathbf{L}\left[i+\frac{n}{2},j\right] \qquad \mathbf{L}_2[i,j] = \mathbf{L}\left[i,j+\frac{n}{2}\right] + \mathbf{L}\left[i+\frac{n}{2},j+\frac{n}{2}\right].$$

Note that since $\mathbf{L}$ is a permutation matrix (and thus has exactly one non-zero entry per column), at most one term of each of these sums can be non-zero.

For sake of contradiction, assume $\mathbf{L}_1$ is not modular-balanced. Then, for some $2^j \leq \frac{n}{2}$, there are two columns $c_1, c_2$ such that $\left\lfloor\frac{c_1}{2^j}\right\rfloor = \left\lfloor\frac{c_2}{2^j}\right\rfloor$ and such that indices of the non-zero entries of $\mathbf{L}_1$ in columns $c_1$ and $c_2$ are the same modulo $2^j$. However, from the definition of $\mathbf{L}_1$, this implies that the indices of the non-zero entries of $\mathbf{L}$ in columns $c_1$ and $c_2$ are also the same modulo $2^j$, contradicting $\mathbf{L}$ being modular-balanced. Hence, $\mathbf{L}_1$ is modular-balanced. An analogous argument (that instead considers columns $c_1 + \frac{n}{2}, c_2 + \frac{n}{2}$ of $\mathbf{L}$) shows that $\mathbf{L}_2$ is also modular-balanced.

To complete the proof, we must argue that $\mathbf{B}_n$ is a butterfly factor of size $n$. Since each $\mathbf{L}_i$ is modular-balanced, it is a permutation matrix. Therefore, $\mathbf{L}'$ has exactly 1 non-zero entry in each of the first $\frac{n}{2}$ rows and columns from $\mathbf{L}_1$ and exactly 1 non-zero entry in each of the second $\frac{n}{2}$ rows and columns from $\mathbf{L}_2$. Hence, $\mathbf{L}'$ is a permutation matrix. Since both $\mathbf{L}$ and $\mathbf{L}'$ are permutation matrices, $\mathbf{B} = \mathbf{L}\left(\mathbf{L}'\right)^{-1}$ must also be a permutation matrix. Therefore, we can view $\mathbf{B}$ as performing a permutation of the rows of $\mathbf{L}'$ to get $\mathbf{L}$.

Consider the $i$'th row of $\mathbf{L}'$, with $0 \leq i < \frac{n}{2}$. There are two possible cases.

Case 1: $\mathbf{L}'[i,:] = \mathbf{L}[i,:]$

In this case, the column of $\mathbf{L}$ with a non-zero entry in row $i$ is in the left $\frac{n}{2}$ columns. The column of $\mathbf{L}$ with a non-zero entry in row $i + \frac{n}{2}$ must, therefore, be in the right $\frac{n}{2}$ columns, otherwise $\mathbf{L}$ would not satisfy the $\frac{n}{2}$ balance condition. Therefore, $\mathbf{L}'\left[i+\frac{n}{2},:\right] = \mathbf{L}\left[i+\frac{n}{2},:\right]$, so we set $\mathbf{B}[i,i] = \mathbf{B}\left[i+\frac{n}{2},i+\frac{n}{2}\right] = 1$.

Case 2: $\mathbf{L}'[i,:] \neq \mathbf{L}[i,:]$

By the definition of $\mathbf{L}'$, $\mathbf{L}'[i,:] = vL\left[i+\frac{n}{2},:\right]$. In this case, the column of $\mathbf{L}$ with a non-zero entry in row $i + \frac{n}{2}$ must be in the left $\frac{n}{2}$ columns. By the $\frac{n}{2}$ balance condition of $\mathbf{L}$, the column of $\mathbf{L}$ with a non-zero entry in row $i$ must be in the right $\frac{n}{2}$ columns. Therefore, $\mathbf{L}'\left[i+\frac{n}{2},:\right] = \mathbf{L}\left[i,:\right]$, so we set $\mathbf{B}\left[i,i+\frac{n}{2}\right] = \mathbf{B}\left[i+\frac{n}{2},i\right] = 1$.

In both cases, the non-zero entries of $\mathbf{B}$ fall into the correct diagonal bands (the main diagonal, and the bands $\frac{n}{2}$ away). Hence, $\mathbf{B}$ is a butterfly factor of size $n$. $\qquad\square$

Now, we consider the process of transforming $\mathbf{P}$ into a modular-balanced matrix. We make use of the following lemma.

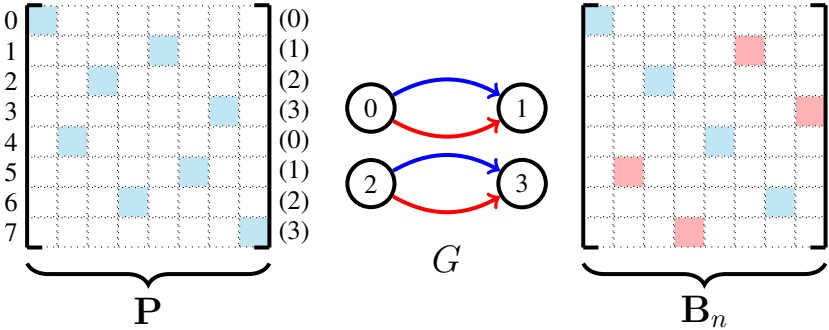

Figure 8: First step of balancing $8 \times 8$ bit reversal permutation (a component of the $8 \times 8$ DFT). Red signifies edges that must be flipped.

**Lemma G.2.** *Let $\mathbf{M}$ be a $k \times k$ matrix with 1 non-zero entry per column, such that for each $0 \leq m < \frac{k}{2}$, there are exactly 2 columns with non-zero entry in a row with index $\equiv m \left(\bmod \frac{k}{2}\right)$. Then, there is a butterfly factor $\mathbf{B}_k$ such that $\mathbf{M}\mathbf{B}_k = \mathbf{M}'$, where $\mathbf{M}'$ meets the $\frac{k}{2}$ balance condition.*

*Proof.* We construct a directed graph $G$ with nodes in $\left[\frac{k}{2}\right]$. For each $0 \leq i < \frac{k}{2}$ we add a directed edge from node $\left(s \mod \frac{k}{2}\right)$ to node $\left(t \mod \frac{k}{2}\right)$ if $\mathbf{M}[:, i] = \mathbf{e}_s$ and $\mathbf{M}\left[:, i + \frac{k}{2}\right] = \mathbf{e}_t$. Each node has (undirected) degree exactly 2 by the structure of $\mathbf{M}$. Hence, $G$ is a union of disjoint (undirected) cycles.

If $\mathbf{M}$ met the $\frac{k}{2}$ balance condition, then each node would additionally have in-degree exactly 1 and out-degree exactly 1. By reversing edges of $G$ such that each (undirected) cycle becomes a directed cycle, we can achieve this. However, reversing edges corresponds to swapping columns of $\mathbf{M}$ that are $\frac{k}{2}$ apart. Let $\mathbf{B}_k$ be the permutation matrix that performs all such swaps. $\mathbf{B}_k$ has non-zero entries only along the main diagonal and the diagonal bands $\frac{k}{2}$ away, and thus is a butterfly factor of size $k$. $\square$

We are ready to present the decomposition of $\mathbf{P}$.

**Lemma G.3.** *Let $\mathbf{P}$ be an $n \times n$ permutation matrix. Then we can decompose $\mathbf{P}$ into $\mathbf{P} = \mathbf{LR}$, where $\mathbf{L}$ is modular-balanced and $\mathbf{R} \in \mathcal{B}^*$.*

*Proof.* We repeatedly apply Lemma G.2. First, we conclude that there is a butterfly factor $\mathbf{B}_n$ such that

$$\mathbf{PB}_n = \mathbf{P}',$$

where $\mathbf{P}'$ meets the $\frac{n}{2}$ balance condition. Now, we consider the first and last $\frac{n}{2}$ columns of $\mathbf{P}'$ independently. We can again apply Lemma G.2 (twice) to conclude that there are butterfly factors $\left[\mathbf{B}_{\frac{n}{2}}\right]_1, \left[\mathbf{B}_{\frac{n}{2}}\right]_2$ such that

$$\mathbf{PB}_n \begin{bmatrix} \left[\mathbf{B}_{\frac{n}{2}}\right]_1 & 0 \\ 0 & \left[\mathbf{B}_{\frac{n}{2}}\right]_2 \end{bmatrix} = \mathbf{PB}_n^{(n)} \mathbf{B}_{\frac{n}{2}}^{(n)} = \mathbf{P}'',$$

where $\mathbf{P}''$ meets the $\frac{n}{2}$ and $\frac{n}{4}$ balance conditions.

We continue this process until we obtain a matrix that meets all of the balance conditions. Our final equation is of the form:

$$\mathbf{P} \cdot \mathbf{B}_n^{(n)} \mathbf{B}_{\frac{n}{2}}^{(n)} \dots \mathbf{B}_2^{(n)} = \mathbf{PB} = \mathbf{L},$$

where $\mathcal{B}$ is a butterfly matrix and $\mathbf{L}$ is a modular-balanced matrix. Let $\mathbf{R} = \mathbf{B}^{-1} = \mathbf{B}^*$ (since $\mathbf{B}$ is a permutation matrix, and thus is orthogonal) and hence $\mathbf{R} \in \mathcal{B}^*$. Then $\mathbf{P} = \mathbf{LR}$, as desired. $\square$

Theorem 2 follows immediately from Lemmas G.3 and G.1.

We now show that permutations are also in $\mathcal{B}^*\mathcal{B}$. We start with the relationship between butterfly matrices and the bit-reversal permutation.

**Lemma G.4.** *Let $\mathbf{P}_{\mathrm{br}}$ be the $n \times n$ bit-reversal permutation matrix where $n$ is some power of 2, and let $\mathbf{M}_1 \in \mathcal{B}$ be an $n \times n$ butterfly matrix. Then there is some butterfly matrix $\mathbf{M}_2 \in \mathcal{B}$ such that*

$$\mathbf{M}_1^* = \mathbf{P}_{\mathrm{br}} \mathbf{M}_2 \mathbf{P}_{\mathrm{br}}.$$

*Proof sketch.* For any input vector $\mathbf{x}$ of length $n$, to perform $\mathbf{M}_1^* \mathbf{x}$, we trace through $\log_2 n$ steps of the multiplication algorithm. At each step, we perform $2 \times 2$ matrix multiplication on elements of $\mathbf{x}$ whose indices are $n/2$ apart (e.g. indices 0 and $n/2$, 1 and $n/2 + 1$, etc.), then $n/4$ apart, and so on, till indices are that 1 apart. If we apply the bit-reversal permutation on $\mathbf{x}$, then indices that are $n/2$ apart will become 1 apart, indices that are $n/4$ apart will become 2 apart, and so on. So the multiplication algorithm $\mathbf{M}_1^* \mathbf{x}$ is equivalent to applying bit-reversal, then multiplying the permuted vector with another butterfly matrix (i.e. $2 \times 2$ matrix multiplication on indices that are 1 apart, then 2 apart, and so on, till indices that are $n/2$ apart). Finally we need to do another bit-reversal permutation to put all the indices back to the original order. If we call this other butterfly matrix $\mathbf{M}_2$, then we have shown that $\mathbf{M}_1^* \mathbf{x} = \mathbf{P}_{\mathrm{br}} \mathbf{M}_2 \mathbf{P}_{\mathrm{br}} \mathbf{x}$. This holds for all $\mathbf{x}$ (for the same matrix $\mathbf{M}_2$), so we have $\mathbf{M}_1^* = \mathbf{P}_{\mathrm{br}} \mathbf{M}_2 \mathbf{P}_{\mathrm{br}}$. $\square$

**Remark G.5.** *Lemma G.4 explains the connection between the two most common fast Fourier transform algorithm, decimation in time and decimation in frequency. Using the decimation-in-time*

*FFT, we can write the DFT matrix $\mathbf{F}$ as product of a butterfly matrix $\mathbf{M}_1$ and the bit-reversal permutation (see Section J):*

$$\mathbf{F} = \mathbf{M}_1 \mathbf{P}_{\mathrm{br}}.$$

*Taking conjugate transpose, we obtain $\mathbf{F}^* = \mathbf{P}_{\mathrm{br}} \mathbf{M}_1^*$ (recall that $\mathbf{P}_{\mathrm{br}}$ is its own transpose/inverse). On the other hand, $\mathbf{F}^*$ is just a scaled version of the inverse DFT matrix, so apply decimation-in-time FFT to the inverse DFT, we can write $\mathbf{F}^* = \mathbf{M}_2 \mathbf{P}_{\mathrm{br}}$ for some other butterfly matrix $\mathbf{M}_2$. Hence $\mathbf{P}_{\mathrm{br}} \mathbf{M}_1^* = \mathbf{M}_2 \mathbf{P}_{\mathrm{br}}$, and thus $\mathbf{P}_{\mathrm{br}} \mathbf{M}_1^* \mathbf{P}_{\mathrm{br}} = \mathbf{M}_2$ (for these particular butterfly matrices $\mathbf{M}_1$ and $\mathbf{M}_2$). Note that this yields another decomposition of the DFT matrix, $\mathbf{F} = \mathbf{P}_{\mathrm{br}} \mathbf{M}_2^*$, which is exactly the decimation-in-frequency FFT algorithm.*

We are ready to show that permutations are in $\mathcal{B}^* \mathcal{B}$.

**Lemma G.6.** *Let $\mathbf{P}$ be an $n \times n$ permutation matrix (with $n$ a power of 2). Then there are butterfly matrices $\mathbf{M}_1, \mathbf{M}_2 \in \mathcal{B}$ such that $\mathbf{P} = \mathbf{M}_1^* \mathbf{M}_2$.*

*Proof.* Consider the permutation $\tilde{\mathbf{P}} = \mathbf{P}_{\mathrm{br}} \mathbf{P} \mathbf{P}_{\mathrm{br}}$. By Theorem 2, there are some butterfly matrices $\tilde{\mathbf{M}}_1, \tilde{\mathbf{M}}_2 \in \mathcal{B}$ such that $\tilde{\mathbf{P}} = \tilde{\mathbf{M}}_1 \tilde{\mathbf{M}}_2^*$. Applying Lemma G.4, we can replace $\tilde{\mathbf{M}}_2^*$ with $\mathbf{P}_{\mathrm{br}} \mathbf{M}_2 \mathbf{P}_{\mathrm{br}}$ for some butterfly matrix $\mathbf{M}_2 \in \mathcal{B}$. We thus have:

$$\mathbf{P}_{\mathrm{br}} \mathbf{P} \mathbf{P}_{\mathrm{br}} = \tilde{\mathbf{M}}_1 \mathbf{P}_{\mathrm{br}} \mathbf{M}_2 \mathbf{P}_{\mathrm{br}}.$$

Pre- and post-multiply both sides by $\mathbf{P}_{\mathrm{br}}$ (which is its own inverse):

$$\mathbf{P} = \mathbf{P}_{\mathrm{br}} \tilde{\mathbf{M}}_1 \mathbf{P}_{\mathrm{br}} \mathbf{M}_2.$$

Applying Lemma G.4 again, we can replace $\mathbf{P}_{\mathrm{br}} \tilde{\mathbf{M}}_1 \mathbf{P}_{\mathrm{br}}$ with $\mathbf{M}_1^*$ for some butterfly matrix $\mathbf{M}_1 \in \mathcal{B}$. Thus:

$$\mathbf{P} = \mathbf{M}_1^* \mathbf{M}_2.$$

$\square$

## H $\quad \mathcal{B} \mathcal{B}^*$ Closure Lemmas

Here, we present some basic facts of the $\mathcal{B} \mathcal{B}^*$ hierarchy that will be useful for later constructions. For simplicity, we assume (WLOG via 0-padding) that all matrices are square matrices with size that is a power of 2.

**Lemma H.1.** *If $\mathbf{M} \in \mathcal{B}$ (or $\mathbf{M} \in \mathcal{B}^*$), then $\mathbf{D} \mathbf{M}, \mathbf{M} \mathbf{D} \in \mathcal{B}$ ($\mathcal{B}^*$ resp.) for any diagonal matrix $\mathbf{D}$.*

*Proof.* Left multiplication by a diagonal matrix scales the rows of $\mathbf{M}$ by the corresponding diagonal entries. The same can be achieved by scaling all entries the leftmost butterfly factor matrix. Similarly, right multiplication by a diagonal matrix scales the columns of $\mathbf{M}$, which can be achieved by scaling all entries in the columns of the rightmost butterfly factor matrix. $\square$

**Lemma H.2.** *Let $\mathbf{A}, \mathbf{B} \in \mathbb{F}^{n \times n}$. If $\mathbf{A} \in (\mathcal{B} \mathcal{B}^*)_e^{w_1}$ and $\mathbf{B} \in (\mathcal{B} \mathcal{B}^*)_e^{w_2}$ then $\mathbf{A} \mathbf{B} \in (\mathcal{B} \mathcal{B}^*)_e^{w_1 + w_2}$.*

*Proof.* Let $\mathbf{E}_{\mathbf{A}}, \mathbf{E}_{\mathbf{B}} \in \mathbb{F}^{en \times en}$ be defined such that $\mathbf{A} = \mathbf{S} \mathbf{E}_{\mathbf{A}} \mathbf{S}^T$, $\mathbf{B} = \mathbf{S} \mathbf{E}_{\mathbf{B}} \mathbf{S}^T$ (with $\mathbf{S}$ as in Definition 2.4). Then

$$\mathbf{A} \mathbf{B} = \mathbf{S} \underbrace{\begin{bmatrix} \mathbf{I}_n & 0 \\ 0 & 0 \end{bmatrix} \mathbf{E}_{\mathbf{A}}}_{en \times en} \underbrace{\begin{bmatrix} \mathbf{I}_n & 0 \\ 0 & 0 \end{bmatrix} \mathbf{E}_{\mathbf{B}}}_{en \times en} \mathbf{S}^T$$

$\begin{bmatrix} \mathbf{I}_n & 0 \\ 0 & 0 \end{bmatrix} \mathbf{E}_{\mathbf{A}} \in (\mathcal{B} \mathcal{B}^*)^{w_1}$, $\begin{bmatrix} \mathbf{I}_n & 0 \\ 0 & 0 \end{bmatrix} \mathbf{E}_{\mathbf{B}} \in (\mathcal{B} \mathcal{B}^*)^{w_2}$ by Lemma H.1. Hence, $\mathbf{A} \mathbf{B} \in (\mathcal{B} \mathcal{B}^*)_e^{w_1 + w_2}$ by Definition 2.4. $\square$

**Lemma H.3.** *Let $\mathbf{A}_1, \ldots, \mathbf{A}_m \in \mathbb{F}^{k \times k}$. If $\mathbf{A}_1, \ldots, \mathbf{A}_m \in (\mathcal{B} \mathcal{B}^*)_e^w$ then $\mathrm{Diag}(\mathbf{A}_1, \ldots, \mathbf{A}_m) \in (\mathcal{B} \mathcal{B}^*)_e^{w+2}$.*

*Proof.* For each $1 \leq i \leq m$, let $\mathbf{E}_{\mathbf{A}_i} \in \mathbb{F}^{ek \times ek}$ be defined such that $\mathbf{A}_i = \mathbf{S}\mathbf{E}_{\mathbf{A}_i}\mathbf{S}^T$ (with $\mathbf{S}$ as in Definition 2.4). Then

$$
\begin{bmatrix}
\mathbf{A}_1 & 0 & \dots & 0 \\
0 & \mathbf{A}_2 & \dots & 0 \\
\vdots & \vdots & \ddots & 0 \\
0 & 0 & \dots & \mathbf{A}_m
\end{bmatrix}
= \mathbf{SP}
\begin{bmatrix}
\mathbf{E}_{\mathbf{A}_1} & 0 & \dots & 0 \\
0 & \mathbf{E}_{\mathbf{A}_2} & \dots & 0 \\
\vdots & \vdots & \ddots & 0 \\
0 & 0 & \dots & \mathbf{E}_{\mathbf{A}_m}
\end{bmatrix}
\mathbf{P}^T \mathbf{S}^T
$$

where $\mathbf{P}$ is a permutation that that moves the first $k$ rows of each $\mathbf{E}_{\mathbf{A}_i}$ (in order) into the top $mk$ rows. From Theorem 2, $\mathbf{P} \in \mathcal{BB}^*$, (and so is $\mathbf{P}^T$, also a permutation). Within the RHS block matrix, the decompositions of each $\mathbf{E}_{\mathbf{A}_i}$ can be done in parallel, requiring total width $w$. Hence, $\mathrm{Diag}(\mathbf{A}_1, \dots, \mathbf{A}_m) \in (\mathcal{BB}^*)_e^{w+2}$, as desired. $\square$

**Remark H.4.** *If $e = 1$ in Lemma H.3, then $\mathbf{P}$ is unnecessary. Hence, $\mathrm{Diag}(\mathbf{A}_1, \dots, \mathbf{A}_m) \in (\mathcal{BB}^*)^w$.*

**Lemma H.5.** *Let $\mathbf{A}_1, \dots, \mathbf{A}_m$ be $k \times k$ matrices in $(\mathcal{BB}^*)_e^w$ then $\sum_{i=1}^m \mathbf{A}_i \in (\mathcal{BB}^*)_{4e}^{mw}$.*

*Proof.* For each $1 \leq i \leq m$, let $\mathbf{E}_{\mathbf{A}_i} \in \mathbb{F}^{ek \times ek}$ be defined such that $\mathbf{A}_i = \mathbf{S}\mathbf{E}_{\mathbf{A}_i}\mathbf{S}^T$ (with $\mathbf{S}$ as in Definition 2.4). Note that $\mathbf{E}_{\mathbf{A}_i} \in (\mathcal{BB}^*)^w$. Consider matrices of the form:

$$
\underbrace{\begin{bmatrix}
\mathbf{I}_{ek} & \mathbf{E}_{\mathbf{A}_i} & 0 & 0 \\
0 & \mathbf{I}_{ek} & 0 & 0 \\
0 & 0 & 0 & 0 \\
0 & 0 & 0 & 0
\end{bmatrix}}_{\mathbf{M}_i \in \mathbb{F}^{4ek \times 4ek}}
= \underbrace{\begin{bmatrix}
\mathbf{I}_{2ek} & \mathbf{I}_{2ek} \\
0 & 0
\end{bmatrix}}_{\mathbf{L}}
\underbrace{\begin{bmatrix}
\mathbf{I}_{ek} & 0 & 0 & 0 \\
0 & \mathbf{I}_{ek} & 0 & 0 \\
0 & 0 & \mathbf{E}_{\mathbf{A}_i} & 0 \\
0 & 0 & 0 & 0
\end{bmatrix}}_{\mathbf{S}}
\underbrace{\begin{bmatrix}
\mathbf{I}_{ek} & 0 & 0 & 0 \\
0 & \mathbf{I}_{ek} & 0 & 0 \\
0 & 0 & 0 & \mathbf{I}_{ek} \\
0 & 0 & \mathbf{I}_{ek} & 0
\end{bmatrix}}_{\mathbf{P}_1}
\underbrace{\begin{bmatrix}
\mathbf{I}_{2ek} & 0 \\
\mathbf{I}_{2ek} & 0
\end{bmatrix}}_{\mathbf{R}}.
$$

Here, $\mathbf{L}$ and $\mathbf{R}$ compute the sum of the $2ek \times 2ek$ matrices on the diagonal of $\mathbf{SP}_1$, where $\mathbf{P}_1$ is a permutation swapping $\mathbf{E}_{\mathbf{A}_i}$ to the $4^{th}$ $ek$-block column. Note that $\mathbf{S}$ is the diagonalization of four matrices in $(\mathcal{BB}^*)^w$, so $\mathbf{S} \in (\mathcal{BB}^*)^w$ by Remark H.4. In addition, since each block in $\mathbf{S}$ is a butterfly matrix of size $ek$, $\mathbf{S}$ only uses butterfly factors up to size $ek$, so the outer factor matrices of sizes $4ek$ and $2ek$ in $\mathbf{S}$ are unused. Also note that $\mathbf{L}$ and $\mathbf{R}$ are butterfly factor matrices of size $4ek$ (or $\mathbf{B}_{4ek}^{(4ek)}$), and $\mathbf{P}_1$ is a butterfly factor matrix of size $2ek$ (or $\mathbf{B}_{2ek}^{(4ek)}$). This allows us to fold the surrounding matrices $\mathbf{L}, \mathbf{P}_1, \mathbf{R}$ into $\mathbf{S}$, so $\mathbf{M}_i \in (\mathcal{BB}^*)^w$.

Through repeated application ($m$ times) of the identity

$$
\begin{bmatrix} \mathbf{I} & \mathbf{A} \\ 0 & \mathbf{I} \end{bmatrix}
\begin{bmatrix} \mathbf{I} & \mathbf{B} \\ 0 & \mathbf{I} \end{bmatrix}
= \begin{bmatrix} \mathbf{I} & \mathbf{A} + \mathbf{B} \\ 0 & \mathbf{I} \end{bmatrix},
$$

we see that

$$
\underbrace{\begin{bmatrix}
\mathbf{I}_{ek} & \sum_{i=1}^m \mathbf{E}_{\mathbf{A}_i} & 0 & 0 \\
0 & \mathbf{I}_{ek} & 0 & 0 \\
0 & 0 & 0 & 0 \\
0 & 0 & 0 & 0
\end{bmatrix}}_{\mathbf{M} \in \mathbb{F}^{4en \times 4en}}
= \prod_{i=1}^m \mathbf{M}_i. \tag{2}
$$

From Lemma H.2, $\mathbf{M} \in (\mathcal{BB}^*)^{mw}$. Finally, note that $\sum_{i=1}^m \mathbf{A}_i = \mathbf{S}\mathbf{M}\mathbf{P}_2\mathbf{S}^T$, where $\mathbf{P}_2$ is a permutation that moves the first $k$ columns of the second block-column of $\mathbf{M}$ to the left. $\mathbf{P}_2$ can be folded into the final summation factor $M_m$ as follows:

$$
\underbrace{\begin{bmatrix}
\mathbf{I}_{ek} & 0 & 0 & 0 \\
0 & \mathbf{I}_{ek} & 0 & 0 \\
0 & 0 & 0 & \mathbf{I}_{ek} \\
0 & 0 & \mathbf{I}_{ek} & 0
\end{bmatrix}}_{\mathbf{P}_1}
\underbrace{\begin{bmatrix}
\mathbf{I}_{2ek} & 0 \\
\mathbf{I}_{2ek} & 0
\end{bmatrix}}_{\mathbf{R}}
\underbrace{\begin{bmatrix}
0 & \mathbf{I}_{ek} & 0 & 0 \\
\mathbf{I}_{ek} & 0 & 0 & 0 \\
0 & 0 & \mathbf{I}_{ek} & 0 \\
0 & 0 & 0 & \mathbf{I}_{ek}
\end{bmatrix}}_{\mathbf{P}_2}
=
\underbrace{\begin{bmatrix}
0 & \mathbf{I}_{ek} & 0 & 0 \\
\mathbf{I}_{ek} & 0 & 0 & 0 \\
0 & 0 & \mathbf{I}_{ek} & 0 \\
0 & 0 & 0 & \mathbf{I}_{ek}
\end{bmatrix}}_{\mathbf{P}_1'}
\underbrace{\begin{bmatrix}
\mathbf{I}_{2ek} & 0 \\
\mathbf{I}_{2ek} & 0
\end{bmatrix}}_{\mathbf{R}} \tag{3}
$$

Hence, $\sum_{i=1}^m \mathbf{A}_i \in (\mathcal{BB}^*)_{4e}^{mw}$, as desired. $\square$

**Lemma H.6.** *Let* $\mathbf{M}$ *be an invertible* $n \times n$ *matrix such that* $M \in \mathcal{B}$. *Then* $\mathbf{M}^{-1} \in \mathcal{B}^*$.

*Proof.* We prove this in a series of steps.

First, let $\mathbf{B}_k$ be an invertible butterfly factor of size $k$. Consider the method of computing $\mathbf{B}_k^{-1}$ by performing Gaussian elimination on the matrix $[\mathbf{B}_k | \mathbf{I}_k]$ to obtain the matrix $\left[ \mathbf{I}_k | \mathbf{B}_k^{-1} \right]$. By the form of $\mathbf{B}$, non-zero entries within a row or column are always exactly $\frac{k}{2}$ positions apart. Therefore, the only row operations needed for this Gaussian elimination are:

- Scaling a row by a constant factor $c \neq 0$
- Addition of a row to another row exactly $\frac{k}{2}$ rows apart

Performing these operations on $\mathbf{I}_k$ will only allow non-zeros on the main diagonal and $\frac{k}{2}$ diagonals away from the main diagonal. Hence, $\mathbf{B}_k^{-1}$ is also a butterfly factor of size $k$.

Next, let $\mathbf{B}_k^{(n)}$ be an invertible butterfly factor matrix of size $n$ and block size $k$. Its inverse is the block diagonal matrix formed by the inverses of each of its constituent butterfly factors. From above, $\left( \mathbf{B}_k^{(n)} \right)^{-1}$ is also a butterfly factor matrix of size $n$ and block size $k$.

Finally, consider $\mathbf{M} \in \mathcal{B}$.

$$\mathbf{M}^{-1} = \left( \mathbf{B}_n^{(n)} \mathbf{B}_{\frac{n}{2}}^{(n)} \dots \mathbf{B}_2^{(n)} \right)^{-1} = \left( \mathbf{B}_2^{(n)} \right)^{-1} \left( \mathbf{B}_4^{(n)} \right)^{-1} \dots \left( \mathbf{B}_n^{(n)} \right)^{-1} = \mathbf{B}_2'^{(n)} \mathbf{B}_4'^{(n)} \dots \mathbf{B}_n'^{(n)} \in \mathcal{B}^*$$

$\square$

Finally, we include a closure result for the Kronecker product, another common matrix composition operation. Although Lemma H.7 is not directly used in the subsequent proofs, it allows for examples the results for the DFT to be lifted to higher-dimensional Fourier transforms. We also note that the closure bound in Lemma H.7 can be tightened in such cases (cf. Remark H.4).

**Lemma H.7.** *Let* $\mathbf{A}, \mathbf{B} \in \mathbb{F}^{n \times n}$. *If* $\mathbf{A} \in (\mathcal{B}\mathcal{B}^*)_e^{w_1}$ *and* $\mathbf{B} \in (\mathcal{B}\mathcal{B}^*)_e^{w_2}$ *then* $\mathbf{A} \otimes \mathbf{B} \in (\mathcal{B}\mathcal{B}^*)_e^{w_1 + w_2 + 6}$.

*Proof.* Note that
$$\mathbf{A} \otimes \mathbf{B} = (\mathbf{A} \otimes \mathbf{I})(\mathbf{I} \otimes \mathbf{B}) = \mathbf{P}^{-1}(\mathbf{I} \otimes \mathbf{A})\mathbf{P}(\mathbf{I} \otimes \mathbf{B}),$$

for some permutation $\mathbf{P}$. By Lemma H.3, $\mathbf{I} \otimes \mathbf{A}$ and $\mathbf{I} \otimes \mathbf{B}$ are in $(\mathcal{B}\mathcal{B}^*)_e^{w_1+2}, (\mathcal{B}\mathcal{B}^*)_e^{w_2+2}$ respectively. The result follows from combining with $\mathbf{P} \in \mathcal{B}\mathcal{B}^*$ and Lemma H.2. $\square$

# I   SPARSE MATRICES IN $\mathcal{B}\mathcal{B}^*$ HIERARCHY

In this appendix, we prove Theorem 3. First, we consider matrices with at most $n$ NNZ.

**Lemma I.1.** *let* $\mathbf{S}$ *be an* $n \times n$ *matrix with at most* $n$ *NNZ. Then,* $\mathbf{S} \in (\mathcal{B}\mathcal{B}^*)^4$.

We use this lemma and the addition closure lemma to prove Theorem 3.

*Proof of Theorem 3.* We note that any $s$ sparse matrix is the sum of $\left\lceil \frac{s}{n} \right\rceil$ matrices of at most $n$ NNZ, and we appeal to Lemma H.5. $\square$

In the rest of the section we will prove Lemma I.1. We begin by defining two classes of matrices that will be used in our decomposition.

**Definition I.1.** *An* $n \times n$ *matrix* $\mathbf{H}$ *is a **horizontal step matrix** if for every* $0 \leq i, i' < n$ *and* $0 \leq j \leq j' < n$, *if* $\mathbf{H}[i, j] \neq 0$ *and* $\mathbf{H}[i', j'] \neq 0$, *then* $j' - j \geq (i' - i) \mod n$.

*An* $n \times n$ *matrix* $\mathbf{V}$ *is a **vertical step matrix** if* $\mathbf{V}^*$ *is a horizontal step matrix.*

With this definition, the horizontal step matrix obeys a "Lipschitz-like" condition. Each column of a horizontal step matrix can have at most one non-zero entry, and given two non-zero columns $k$ apart, the non-zero entry in the right column must be between 0 and $k$ rows below the non-zero entry in the left column. Note that to show that a matrix is a horizontal step matrix, it is sufficient to argue that this condition holds for each pair of neighboring non-zero columns.

Similarly, each row of a vertical step matrix can have at most one non-zero entry, and given two non-zero rows $k$ apart, the non-zero entry in the lower row must be between 0 and $k$ columns to the right of the non-zero entry in the upper row.

**Lemma I.2.** *Let* $\mathbf{H}$ *be an* $n \times n$ *horizontal step matrix. Then* $\mathbf{H} \in \mathcal{B}$.

*Proof.* We proceed by induction on $n$. The base case $n = 2$ is trivial. As our inductive hypothesis, we assume that all horizontal step matrices of size $\frac{n}{2} \times \frac{n}{2}$ are butterfly matrices of size $\frac{n}{2}$. From Definition 2.3, it is sufficient to show that $\mathbf{H}$ can be decomposed as:

$$\mathbf{H} = \begin{bmatrix} \mathbf{D}_1 & \mathbf{D}_2 \\ \mathbf{D}_3 & \mathbf{D}_4 \end{bmatrix} \begin{bmatrix} \mathbf{H}_1 & 0 \\ 0 & \mathbf{H}_2 \end{bmatrix} = \begin{bmatrix} \mathbf{D}_1\mathbf{H}_1 & \mathbf{D}_2\mathbf{H}_2 \\ \mathbf{D}_3\mathbf{H}_1 & \mathbf{D}_4\mathbf{H}_2 \end{bmatrix}, \tag{4}$$

where $\mathbf{H}_1, \mathbf{H}_2$ are $\frac{n}{2} \times \frac{n}{2}$ horizontal step matrices and each $\mathbf{D}_k$ is a $\frac{n}{2} \times \frac{n}{2}$ diagonal matrix. Denote the four, $\frac{n}{2} \times \frac{n}{2}$ corner submatrices of $\mathbf{H}$ by:

$$\mathbf{H} = \begin{bmatrix} \mathbf{H}_{11} & \mathbf{H}_{12} \\ \mathbf{H}_{21} & \mathbf{H}_{22} \end{bmatrix}.$$

Then, define $\mathbf{H}_1$ and $\mathbf{H}_2$ by:

$$\mathbf{H}_1 = \mathbf{H}_{11} + \mathbf{H}_{21} \qquad \mathbf{H}_2 = \mathbf{H}_{12} + \mathbf{H}_{22}$$

For sake of contradiction, assume that $\mathbf{H}_1$ is not a horizontal step matrix. Then, there are $0 \le i, i' < \frac{n}{2}$, $0 \le j \le j' < \frac{n}{2}$ such that $\mathbf{H}_1[i, j] \ne 0$, $\mathbf{H}_1[i', j'] \ne 0$, and $j' - j < (i' - i) \mod \frac{n}{2}$. From our definition of $\mathbf{H}_1$, the non-zero entries in columns $j$ and $j'$ of $\mathbf{H}$ are either $\left((i' - i) \mod \frac{n}{2}\right)$ or $\left(\frac{n}{2} + (i' - i) \mod \frac{n}{2}\right)$, both of which are greater than $j' - j$, rows apart. This contradicts $\mathbf{H}$ being a horizontal step matrix. Hence, $\mathbf{H}_1$ must be a horizontal step matrix, as must $\mathbf{H}_2$ from an analogous argument.

Next, we define $\mathbf{D}_1, \mathbf{D}_2, \mathbf{D}_3, \mathbf{D}_4$ by:

$$\mathbf{D}_1[k, k] = \begin{cases} 1 & \mathbf{H}_{21}[k, :] = \mathbf{0} \\ 0 & \text{otherwise} \end{cases} \qquad \mathbf{D}_2[k, k] = \begin{cases} 1 & \mathbf{H}_{22}[k, :] = \mathbf{0} \\ 0 & \text{otherwise} \end{cases}$$

$$\mathbf{D}_3[k, k] = \begin{cases} 1 & \mathbf{H}_{11}[k, :] = \mathbf{0} \\ 0 & \text{otherwise.} \end{cases} \qquad \mathbf{D}_4[k, k] = \begin{cases} 1 & \mathbf{H}_{12}[k, :] = \mathbf{0} \\ 0 & \text{otherwise.} \end{cases}$$

To finish the proof, we argue the correctness of the decomposition by equating arbitrary entries of each of the 4 corner submatrices. We begin with the upper left submatrix.

$$
\begin{aligned}
\mathbf{D}_1\mathbf{H}_1[i, j] = \sum_{k=0}^{\frac{n}{2}} \mathbf{D}_1[i, k] \cdot \mathbf{H}_1[k, j] \qquad & \text{by definition of matrix multiplication} \\
= \mathbf{D}_1[i, i] \cdot \mathbf{H}_1[i, j] \qquad & \mathbf{D}_1 \text{ is a diagonal matrix} \\
= \mathbb{1}_{(\mathbf{H}_{21}[i,:]=\mathbf{0})} \cdot (\mathbf{H}_{11}[i, j] + \mathbf{H}_{21}[i, j]) \qquad & \text{by definition of } \mathbf{D}_1 \text{ and } \mathbf{H}_1
\end{aligned}
$$

Here, we consider two cases:

Case 1: $\mathbf{H}_{21}[i, j] \ne 0$

Since $\mathbf{H}$ is a horizontal step matrix (and hence may have at most one non-zero entry per column), it follows that $\mathbf{H}_{11}[i, j] = 0$. In this case, the indicator function evaluates to 0, so $\mathbf{D}_1\mathbf{H}_1[i, j] = 0 = \mathbf{H}_{11}[i, j]$, as desired.

Case 2: $\mathbf{H}_{21}[i, j] = 0$

If $\mathbf{H}_{11}[i,j] = 0$, then $\mathbf{D}_1\mathbf{H}_1[i,j] = 0 = \mathbf{H}_{11}[i,j]$. Otherwise, for sake of contradiction, suppose that $\mathbf{H}_{21}[i,:] \neq \mathbf{0}$. Then, two of the first $\frac{n}{2}$ columns of $\mathbf{H}$ would have non-zero entries $\frac{n}{2}$ rows apart, contradicting $\mathbf{H}$ being a horizontal step matrix. Hence, $\mathbf{H}_{21}[i,:] = \mathbf{0}$, so $\mathbf{D}_1\mathbf{H}_1[i,j] = \mathbf{H}_{11}[i,j]$, as desired.

In all cases, $\mathbf{D}_1\mathbf{H}_1[i,j] = \mathbf{H}_{11}[i,j]$, so our decomposition correctly recovers the upper left corner of $\mathbf{H}$. Analogous arguments show that the other three corners are also correctly recovered. Hence, our decomposition is correct, and by induction, $\mathbf{H} \in \mathcal{B}$. $\qquad\square$

**Corollary I.3.** *Let $\mathbf{V}$ be a vertical step matrix. Then $\mathbf{V} \in \mathcal{B}^*$.*

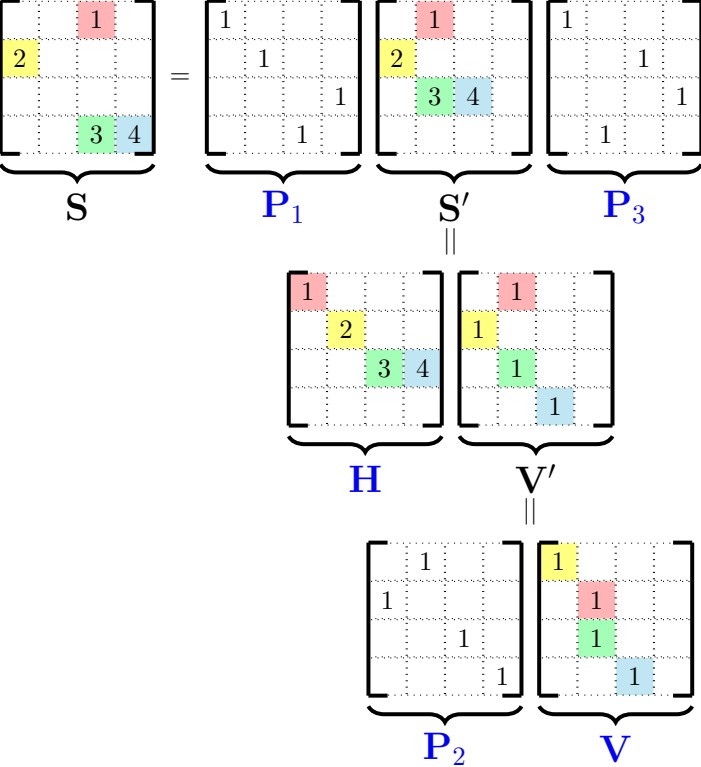

Figure 9: Decomposition of $4 \times 4$ sparse matrix $\mathbf{S}$ into $\mathbf{P}_1\mathbf{H}\mathbf{P}_2\mathbf{V}\mathbf{P}_3$

Now, we use step matrices to prove Lemma I.1.

*Proof of Lemma I.1.* Given $\mathbf{S}$, we decompose it as $\mathbf{S} = \mathbf{P}_1\mathbf{H}\mathbf{P}_2\mathbf{V}\mathbf{P}_3$, where each $P_\ell$ is a permutation matrix, $\mathbf{H}$ is a horizontal step matrix, and $\mathbf{V}$ is a vertical step matrix. For an example of this, see Figure 9.

We first decompose $\mathbf{S}$ as $\mathbf{S} = \mathbf{P}_1\mathbf{S}'\mathbf{P}_3$, where $\mathbf{P}_1$ is the permutation that moves all 0 rows of $\mathbf{S}$ to the bottom and $\mathbf{P}_3$ is the permutation that moves all 0 columns of $\mathbf{S}$ to the right.

Next, we further decompose $\mathbf{S}'$ into $\mathbf{S}' = \mathbf{H}\mathbf{V}'$ as follows. Since $\mathbf{S}'$ has $s \leq n$ NNZ, we can parameterize $\mathbf{S}'$ by $\theta = \{(c_k, i_k, j_k) : 0 \leq k < s\}$ such that $\mathbf{S}'[i_k, j_k] = c_k$, with the non-zero entries indexed in row-major order. Define matrix $\mathbf{H}$ by:

$$\mathbf{H}[:,k] = \begin{cases} c_k \cdot \mathbf{e}_{i_k} & 0 \leq k < s \\ \mathbf{0} & \text{otherwise.} \end{cases}$$

Define matrix $\mathbf{V}'$ by:

$$\mathbf{V}'[k,:] = \begin{cases} \mathbf{e}_{j_k}^T & 0 \leq k < s \\ \mathbf{0} & \text{otherwise.} \end{cases}$$

To show that $\mathbf{S}' = \mathbf{H}\mathbf{V}'$, we consider an arbitrary entry:

$$\mathbf{H}\mathbf{V}'[i,j] = \sum_{k=0}^{n} \mathbf{H}[i,k] \cdot \mathbf{V}'[k,j] \qquad \text{by definition of matrix multiplication}$$

$$= \sum_{k=0}^{s} \mathbf{H}[i,k] \cdot \mathbf{V}'[k,j] \qquad \mathbf{H} \text{ is 0 in all but first } s \text{ columns}$$

$$= \sum_{k=0}^{s} c_k \cdot \mathbb{1}_{i=i_k} \cdot \mathbb{1}_{j=j_k} \qquad \text{by definition of } \mathbf{H} \text{ and } \mathbf{V}'$$

Here, we note that $(i,j)$ can equal $(i_k, j_k)$ for at most one value of $k$ since the locations in $\theta$ are unique. Hence, $\mathbf{H}\mathbf{V}'[i,j] = c_k$ only if $(i,j) = (i_k, j_k)$ for some $k$, which is exactly the definition of $\mathbf{S}'$. Hence, $\mathbf{S}' = \mathbf{H}\mathbf{V}'$.

We argue that $\mathbf{H}$ is a horizontal step matrix through a series of assertions. First, note that $\mathbf{H}$ has exactly one non-zero entry in each of its first $s$ columns. Also, note that since $\theta$ is in row-major order, these non-zero entries are sorted (any column to the right cannot have a non-zero entry in a higher row). Hence, to show that $\mathbf{H}$ is a horizontal step matrix, it is sufficient to argue that adjacent columns of $\mathbf{H}$ have non-zero entries at most one row apart. This is equivalent to $\mathbf{S}'$ having no zero rows between two non-zero rows, which is guaranteed by $\mathbf{P}_1$. Hence, $\mathbf{H}$ is a horizontal step matrix.

Since $\mathbf{V}'$ has at most one non-zero entry per row, we may permute the rows of $\mathbf{V}'$ to obtain a matrix $\mathbf{V}$, where the non-zero entries of $\mathbf{V}$ are sorted (any lower row below cannot have a non-zero entry in an earlier column). Hence, for some permutation matrix $(\mathbf{P}_2)^{-1}$, $\mathbf{V} = (\mathbf{P}_2)^{-1}\mathbf{V}'$, which implies that $\mathbf{V}' = \mathbf{P}_2\mathbf{V}$. It has exactly one non-zero entry in each of its first $s$ columns. From the action of $\mathbf{P}_2$, these non-zero entries are sorted. Therefore, by the same argument as for $\mathbf{H}$ above, $\mathbf{V}^T$ is a horizontal step matrix. Hence, $\mathbf{V}$ is a vertical step matrix.

In all, we have found a decomposition $\mathbf{S} = \mathbf{P}_1\mathbf{H}\mathbf{P}_2\mathbf{V}\mathbf{P}_3$, where each $P_\ell$ is a permutation matrix ($\in \mathcal{B}\mathcal{B}^*$ by Theorem 2), $\mathbf{H}$ is a horizontal step matrix ($\in \mathcal{B}$ by Lemma I.2), and $\mathbf{V}$ is a vertical step matrix ($\in \mathcal{B}^*$ by Corollary I.3). Moreover, by Lemma G.6, $\mathbf{P}_2 \in \mathcal{B}^*\mathcal{B}$, so $\mathbf{H}, \mathbf{P}_2, \mathbf{V}$ can be combined to obtain $\mathbf{H}\mathbf{P}_2\mathbf{V} \in (\mathcal{B}\mathcal{B}^*)^2$. By Lemma H.2, $\mathbf{S} \in (\mathcal{B}\mathcal{B}^*)^4$. $\square$

**Corollary I.4.** *Let $\mathbf{R}$ be an $n \times n$ matrix of rank $r$. Then $\mathbf{R} \in (\mathcal{B}\mathcal{B}^*)_4^{8r}$.*

*Proof.* We can decompose $\mathbf{R}$ as $\mathbf{R} = \mathbf{G}\mathbf{H}^*$ where $\mathbf{G}, \mathbf{H}$ are $n \times r$ matrices. With appropriate zero-padding, both of these can be made into $n \times n$ matrices with at most $rn$ NNZ. The proof follows immediately from Theorem 3 and Lemma H.2. $\square$

# J  EXAMPLE OF K-MATRIX REPRESENTATION OF STRUCTURED MATRICES AND COMPARISON TO $\mathcal{B}\mathcal{P}$ HIERARCHY

In this appendix, we show explicitly how some common structured matrices (e.g. originating from fast transforms) can be represented as K-matrices. We also draw comparisons between the $\mathcal{B}\mathcal{B}^*$ hierarchy and the $\mathcal{B}\mathcal{P}$ hierarchy introduced by Dao et al. (2019).

**Lemma J.1.** *Let $\mathbf{F}_n$ be the Discrete Fourier Transform of size $n$. Then $\mathbf{F}_n \in (\mathcal{B}\mathcal{B}^*)^2$.*

*Proof.* From Parker (1995), we can express $\mathbf{F}_n$ as $\mathbf{F}_n = \mathbf{B}\,\mathbf{P}$, where $\mathbf{B} \in \mathcal{B}$ and $\mathbf{P}$ is a permutation (the bit reversal permutation). From Theorem 2, $\mathbf{P} \in \mathcal{B}\mathcal{B}^*$. Hence, by Lemma H.2, $\mathbf{F}_n \in (\mathcal{B}\mathcal{B}^*)^2$. $\square$

**Lemma J.2.** *Let $\mathbf{H}_n$ be the Hadamard Transform of size $n$. Then $\mathbf{H}_n \in \mathcal{B}\mathcal{B}^*$.*

*Proof.* $\mathbf{H}_n \in \mathcal{B}$, so trivially $\mathbf{H}_n \in \mathcal{B}\mathcal{B}^*$. $\square$

**Lemma J.3.** *Let $\mathbf{S}_n$ be the Discrete Sine Transform of size $n$. Then $\mathbf{S}_n \in (\mathcal{B}\mathcal{B}^*)^2$.*

*Proof.* As described in Makhoul (1980), $\mathbf{S}_n$ can be performed as a scaled permutation (separating the even and odd indices of the input, and reversing and negating the odd indices) composed with $\mathbb{F}_n$. Therefore, we may decompose $\mathbf{S}_n$ as $\mathbf{S}_n = \mathbf{B}\,\mathbf{P}_2\,\mathbf{D}\,\mathbf{P}_1$, where $\mathbf{P}_1, \mathbf{P}_2$ are permutations, $\mathbf{B} \in \mathcal{B}$, and $\mathbf{D}$ is a diagonal matrix. $\mathbf{P}_2\,\mathbf{D}\,\mathbf{P}_1$ is simply a permutation matrix with scaled entries, which can be equivalently expressed as $\mathbf{D}'\,\mathbf{P}'$ for some diagonal matrix $\mathbf{D}'$ and permutation $\mathbf{P}'$. By Lemma H.1, $\mathbf{B}\,\mathbf{D}' \in \mathcal{B}\mathcal{B}^*$. By Theorem 2, $\mathbf{P}' \in \mathcal{B}\mathcal{B}^*$. Hence, by Lemma H.2, $\mathbf{S}_n \in (\mathcal{B}\mathcal{B}^*)^2$. $\qquad\square$

**Remark J.4.** *An analogous argument shows that the Discrete Cosine Transform is also in $(\mathcal{B}\mathcal{B}^*)^2$.*

**Lemma J.5.** *Let $\mathbf{C}_n$ be an $n \times n$ circulant (convolution) matrix. Then $\mathbf{C}_n \in \mathcal{B}\mathcal{B}^*$.*

*Proof.* Using Theorem 2.6.4 of Pan (2001), we can express $\mathbf{C}_n$ as $\mathbf{C}_n = (\mathbf{F}_n)^{-1}\mathbf{D}\mathbf{F}_n$ where $\mathbf{F}_n$ is the Discrete Fourier Transform and $\mathbf{D}$ is a diagonal matrix. $(\mathbf{F}_n)^{-1} = \mathbf{B}\,\mathbf{P}$ (with $\mathbf{B} \in \mathcal{B}$, $\mathbf{P}$ a permutation), which implies that $\mathbf{F}_n = (\mathbf{P})^{-1}(\mathbf{B})^{-1}$. Therefore

$$\mathbf{C}_n = \mathbf{B}\,\mathbf{P}\,\mathbf{D}\,(\mathbf{P})^{-1}(\mathbf{B})^{-1}.$$

The middle three factors have the effect of performing a permutation, scaling each element, and undoing the permutation, which is equivalent to simply scaling by some diagonal matrix $\mathbf{D}'$. Hence, we are left with

$$\mathbf{C}_n = \mathbf{B}\,\mathbf{D}'(\mathbf{B})^{-1}.$$

By Lemma H.1, $\mathbf{B}\,\mathbf{D}' \in \mathcal{B}$. By Lemma H.6, $(\mathbf{B})^{-1} \in \mathcal{B}^*$. Hence, $\mathbf{C}_n \in \mathcal{B}\mathcal{B}^*$. $\qquad\square$

**Remark J.6.** *We can expand any $n \times n$ Toeplitz matrix $\mathbf{T}_n$ into a $2n \times 2n$ circulant matrix (with upper left $n \times n$ submatrix equal to $\mathbf{T}_n$). Hence, $\mathbf{T}_n \in (\mathcal{B}\mathcal{B}^*)^1_2$ by Lemma J.5.*

The Fastfood matrix class (Le et al., 2013) can be tightly captured in the $\mathcal{B}\mathcal{B}^*$ hierarchy:

**Lemma J.7.** *The product $\mathbf{SHDPHB}$ where $\mathbf{S}, \mathbf{D}, \mathbf{B}$ are diagonal matrices, $\mathbf{H}$ is the Hadamard transform, and $\mathbf{P}$ is a permutation matrix, is in $(\mathcal{B}\mathcal{B}^*)^3$.*

*Proof.* We have shown in Lemma J.2 that $\mathbf{H} \in \mathcal{B}\mathcal{B}^*$, and in Theorem 2 that $\mathbf{P} \in \mathcal{B}\mathcal{B}^*$. Since $\mathcal{B}\mathcal{B}^*$ is closed under diagonal multiplication (Lemma H.1), we conclude that $\mathbf{SHDPHB} \in (\mathcal{B}\mathcal{B}^*)^3$. $\qquad\square$

The two classes of matrices introduced in Moczulski et al. (2016), called AFDF and ACDC, are also tightly captured in the $\mathcal{B}\mathcal{B}^*$ hierarchy:

**Lemma J.8.** *Let $\mathbf{AF}^{-1}\mathbf{DF}$ be a product of a diagonal matrix $\mathbf{A}$, the inverse Fourier transform $\mathbf{F}^{-1}$, another diagonal matrix $\mathbf{D}$, and the Fourier transform $\mathbf{F}$. Then $\mathbf{AF}^{-1}\mathbf{DF} \in \mathcal{B}\mathcal{B}^*$.*

*Let $\mathbf{AC}^{-1}\mathbf{DC}$ be a product of a diagonal matrix $\mathbf{A}$, the inverse cosine transform $\mathbf{C}^{-1}$, another diagonal matrix $\mathbf{D}$, and the cosine transform $\mathbf{C}$. Then $\mathbf{AC}^{-1}\mathbf{DC} \in (\mathcal{B}\mathcal{B}^*)^4$.*

*Proof.* We have argued in Lemma J.5 that $\mathbf{F}^{-1}\mathbf{DF} \in \mathcal{B}\mathcal{B}^*$. Since $\mathcal{B}\mathcal{B}^*$ is closed under diagonal multiplication (Lemma H.1), we conclude that $\mathbf{AF}^{-1}\mathbf{DF} \in \mathcal{B}\mathcal{B}^*$.

We have shown that $\mathbf{C} \in (\mathcal{B}\mathcal{B}^*)^2$, so $\mathbf{C}^{-1} \in (\mathcal{B}\mathcal{B}^*)^2$ as well. Since $\mathcal{B}\mathcal{B}^*$ is closed under diagonal multiplication (Lemma H.1), we conclude that $\mathbf{AC}^{-1}\mathbf{DC} \in (\mathcal{B}\mathcal{B}^*)^4$. $\qquad\square$

**Remark J.9.** *Within each butterfly factor matrix of the DFT (excluding the bit reversal permutation) and the Hadamard transform, the columns are pairwise orthogonal and have norm 2. Hence, we can divide all factors by $\sqrt{2}$ to make orthogonal factor matrices. To counteract this scaling, we can add a diagonal matrix with $\sqrt{2}^{\log_2(n)} = \sqrt{n}$ in all entries to the factorization. By doing this we can place all of the above transforms in the $\mathcal{O}\mathcal{B}\mathcal{B}$ hierarchy (defined in Appendix K) with the same width and expansion factor.*

### J.1 MULTI-DIMENSIONAL TRANSFORMS

Here, we show that, using larger matrices, we are able to similarly capture multi-dimensional versions of the above transforms.

**Lemma J.10.** *Let $\mathbf{F}_n^2$ be the 2-dimensional Discrete Fourier Transform (represented as an $n^2 \times n^2$ matrix). Then $\mathbf{F}_n^2 \in (\mathcal{B}\mathcal{B}^*)^2$.*

*Proof.* The separation property of the 2-D DFT allows us to express its action on an $n \times n$ matrix as the composition of a 1-D DFT on each of its rows and a 1-D DFT on each of its columns. If we view the 2-D DFT as an $n^2 \times n^2$ matrix, its input and outputs will both be column vectors of size $n^2$. As our convention, we list the entries of the input vector in the row-major order corresponding to the $n \times n$ input matrix. Then, we consider the 2-D DFT in four steps, where the first two steps perform the 1-D DFT row-wise, and the second two steps perform the 1-D DFT column-wise:

Step 1: Permute the columns:

We permute the columns (with a bit reversal permutation), which performs a bit reversal permutation on each row. Viewing the input as a vector, this step corresponds to left multiplication by a permutation matrix $\mathbf{P}_c$ that permutes the entries of each chunk of size $n$ of the input vector. Step 2: Multiply each row by a butterfly matrix

Since the entries of the input were listed in row major order, this step is achieved through multiplication by a block diagonal matrix of $n$ butterfly matrices of size $n$, which can be viewed as a product of butterfly factor matrices $\mathbf{B}_n^{(n^2)} \ldots \mathbf{B}_{\frac{n}{2}}^{(n^2)} \mathbf{B}_2^{(n^2)}$.

Step 3: Permute the rows:

We permute the rows (with a bit reversal permutation), which performs a bit reversal permutation on each column. This corresponds to left multiplication by a permutation matrix $\mathbf{P}_r$. Since we are permuting the rows, $\mathbf{P}_r$ permutes the entries at the granularity of each $n$-chunk. Since Steps 1 and 2 each performed an identical computation to each $n$-chunk we can move this row permutation before Step 2, combining $\mathbf{P}_c$ and $\mathbf{P}_r$ into a single permutation $\mathbf{P}$.

Step 4: Multiply each column by a butterfly matrix

Consider multiplication by the first factor matrix. In each row, this matrix is taking linear combinations of adjacent column entries. In our length-$n^2$ vector, these entries will be exactly $n$ indices apart. Therefore this multiplication can be handled by a butterfly factor matrix $\mathbf{B}_{2n}^{(n^2)}$. Similarly, we find that this butterfly multiplication can be expressed as multiplication by a product of butterfly factor matrices $\mathbf{B}_{n^2}^{(n^2)} \ldots \mathbf{B}_{\frac{n^2}{2}}^{(n^2)} \mathbf{B}_{2n}^{(n^2)}$. Combined with the factor matrices from Step 2, these form a butterfly matrix $\mathbf{B}$ of size $n^2$.

In all, we see that the 2-D DFT may be realized as multiplication by a permutation matrix $\mathbf{P}$ followed by multiplication by a butterfly matrix $\mathbf{B}$. The same argument as Lemma J.1 shows that $\mathbf{F}_n^2 \in (\mathcal{B}\mathcal{B}^*)^2$. □

**Remark J.11.** *An analogous argument (using the separation property of the respective transforms) can be used to argue that 2-D Discrete Sine and Discrete Cosine transforms are in $(\mathcal{B}\mathcal{B}^*)^2$, and that 2-D Hadamard Transforms are in $\mathcal{B}\mathcal{B}^*$.*

**Lemma J.12.** *Let $\mathbf{C}_n^2$ be a 2-dimensional convolution matrix. Then $\mathbf{C}_n^2 \in \mathcal{B}\mathcal{B}^*$.*

*Proof.* We can express a 2-D convolution matrix as $\mathbf{C}_n^2 = (\mathbf{F}_n^2)^{-1} \mathbf{D} \mathbf{F}_n^2$, where $\mathbf{D}$ is diagonal, $\mathbf{F}_n^2$ is the 2-D Fourier transform and $(\mathbf{F}_n^2)^{-1}$ is the inverse 2-D Fourier transform. From the proof of Lemma J.10, we see that that we can express $\mathbf{F}_n^2$ (and similarly $(\mathbf{F}_n^2)^{-1}$) as the product of a butterfly matrix and a permutation matrix. The rest of the argument is analogous to the proof of Lemma J.5. □

**Remark J.13.** *Using an inductive argument, we can show that all k-dimensional ($k \in \mathbb{Z}$) variants of the above transforms, expressed as $n^k \times n^k$ matrices are contained in $\mathcal{B}\mathcal{B}^*$ or $(\mathcal{B}\mathcal{B}^*)^2$. To do this, we use the separation property of the transforms to break them into a $k - 1$-dimensional transform (the inductive hypothesis) followed by a 1-dimensional transform.*

# K    THE ORTHOGONAL KALEIDOSCOPE HIERARCHY

Through practical application of the butterfly matrices, it has been found useful to constrain them in orthogonality. In Section K.1 we will modify the existing kaleidoscope hierarchy to create the *orthogonal kaleidoscope hierarchy $\mathcal{OBB}$*. Then, in Section K.2, we will argue that all orthogonal matrices, and as a result all matrices, can also be expressed in this hierarchy in $O(n)$ width. Lastly, in Section K.3, we will argue that permutation matrices and sparse matrices also exist in this hierarchy in $O(1)$ width, which in turn implies a corresponding result for matrices with low-depth arithmetic circuits.

## K.1    DEFINITION

The definition of the orthogonal butterfly is identical to the original butterfly, with the constraint that all butterfly factors are orthogonal. We specify this definition below:

**Definition K.1** (Analog of Definition 2.1). *An **orthogonal butterfly factor** of size $k \geq 2$ (denoted as $\widetilde{\mathbf{B}}_k$) is a butterfly factor that is also orthogonal.*

**Definition K.2** (Analog of Definition 2.3). *An **orthogonal butterfly matrix** of size $n$ (denoted as $\widetilde{\mathbf{B}}^{(n)}$) is a butterfly matrix with all butterfly factor matrices being orthogonal.*

Note that the above definition implies that an orthogonal butterfly matrix, as well as its conjugate transpose, is orthogonal.

The orthogonal hierarchy definition nearly mimics the original hierarchy Definition 2.4, as follows:

**Definition K.3.**

- *We say that an $n \times n$ matrix $\mathbf{M} \in \widetilde{\mathcal{B}}$ if we can express $\mathbf{M} = \widetilde{\mathbf{B}}^{(n)}$.*

- *We say that an $n \times n$ matrix $\mathbf{M} \in \widetilde{\mathcal{B}}^*$ if we can express $\mathbf{M} = \left[\widetilde{\mathbf{B}}^{(n)}\right]^*$.*

- *We say that an $n \times n$ matrix $\mathbf{M} \in \mathcal{OBB}$ if we can express $\mathbf{M} = \mathbf{M}_1 \mathbf{D} \mathbf{M}_2$ for some $\mathbf{M}_1 \in \widetilde{\mathcal{B}}, \mathbf{M}_2 \in \widetilde{\mathcal{B}}^*$, and diagonal matrix $\mathbf{D}$. Note that $\mathbf{D}$ need not be full rank.*

- *Width $w$ and expansion $e$ in $(\mathcal{OBB})_e^w$ mimic the same definition as in the original hierarchy, using $\mathcal{OBB}$ instead of $\mathcal{BB}^*$, such that $\mathbf{E} \in (\mathcal{OBB})^w$.*

By padding if necessary, we will assume that $n$ is a power of 2.

## K.2    EXPRESSIVITY

In this subsection we prove that all orthogonal (resp. unitary) matrices are contained in $\mathcal{OBB}^n$. To do this, we consider the class of *Householder reflections*, given by $\mathbf{I} - 2\mathbf{u}\mathbf{u}^*$ for any unit vector $\mathbf{u}$ (Householder, 1958):

**Lemma K.1.** *All Householder reflections are in $\mathcal{OBB}$ with inner diagonal matrix $\mathbf{I}$.*

We will prove this lemma shortly. First, we use this lemma to present a decomposition for all orthogonal (resp. unitary) matrices.

**Lemma K.2.** *Let $\mathbf{M}$ be an $n \times n$ orthogonal/unitary matrix. Then $\mathbf{M} \in (\mathcal{OBB})^{n-1}$.*

*Proof.* We consider the $\mathbf{QR}$ decomposition of $\mathbf{M}$. It is known that we can compose $\mathbf{M}$ into a product of $n - 1$ Householder reflections and an orthogonal/unitary diagonal matrix (Householder, 1958).[10] From Lemma K.1, each Householder reflection is in $\mathcal{OBB}$.

To complete the proof, we argue that $\mathbf{R}$ can be folded into the rightmost butterfly matrix. Let $\mathbf{Q}_1$ be the rightmost butterfly factor matrix in $\mathbf{Q}$ ($\in \widetilde{\mathbf{B}}_n^{(n)}$). Right multiplication of $\mathbf{Q}_1$ by $\mathbf{R}$ scales each columns of $\mathbf{Q}_1$ by some $c \in \mathbb{C}$ with $||c|| = 1$ ($\mathbf{R}$ is unitary diagonal). This preserves both the sparsity

---

[10]$\mathbf{Q}$ is the (orthogonal/unitary) product of $n - 1$ Householder reflections. $\mathbf{R}$, the remaining upper triangular matrix after performing these reflections, is itself orthogonal/unitary, and therefore diagonal.

pattern of $\mathbf{Q}_1$ and the orthogonality of its columns. Moreover, the norm of each column of $\mathbf{Q}_1\mathbf{R}$ is 1. Therefore, $\mathbf{Q}_1\mathbf{R}$ is an orthogonal butterfly factor matrix, so $\mathbf{M} = \mathbf{QR} \in (\mathcal{OBB})^{n-1}$, as desired. $\square$

We now return to the proof of Lemma K.1

*Proof of Lemma K.1.* Given $\mathbf{u} \in \mathbb{C}^n$ ($n$ a power of 2), let $\mathbf{u}_0 = \mathbf{u}[: n/2] \in \mathbb{C}^{n/2}, \mathbf{u}_1 = \mathbf{u}[n/2 :] \in \mathbb{C}^{n/2}$ denote the first and second halves of $\mathbf{u}$.

To show that $\mathbf{H} \in \mathcal{OBB}$ with inner diagonal matrix $\mathbf{I}$, we proceed by induction. The base case for $n = 2$ is trivial. It suffices to show that there exist unitary butterfly factors $\mathbf{L}, \mathbf{R}$ such that $\mathbf{LHR}$ has the form $\begin{bmatrix} \mathbf{I}_{n/2} - 2\mathbf{v}_0\mathbf{v}_0^* & \mathbf{0} \\ \mathbf{0} & \mathbf{I}_{n/2} - 2\mathbf{v}_1\mathbf{v}_1^* \end{bmatrix}$ for some unit vectors $\mathbf{v}_0, \mathbf{v}_1 \in \mathbb{C}^{n/2}$.

Define

$$(\mathbf{v}_0[i], \mathbf{v}_1[i]) = \begin{cases} \left( \frac{\mathbf{u}_0[i]}{\sqrt{|\mathbf{u}_0[i]|^2 + |\mathbf{u}_1[i]|^2}}, \frac{\mathbf{u}_1[i]}{\sqrt{|\mathbf{u}_0[i]|^2 + |\mathbf{u}_1[i]|^2}} \right) & \text{if } |\mathbf{u}_0[i]|^2 + |\mathbf{u}_1[i]|^2 \neq 0 \\ (1, 0) & \text{otherwise} \end{cases}. \quad (5)$$

It is easily checked that

$$\begin{aligned} \mathbf{v}_0[i]^*\mathbf{v}_0[i] + \mathbf{v}_1[i]^*\mathbf{v}_1[i] &= 1 \\ \mathbf{v}_0[i]^*\mathbf{u}_0[i] + \mathbf{v}_1[i]^*\mathbf{u}_1[i] &= \sqrt{|\mathbf{u}_0[i]|^2 + |\mathbf{u}_1[i]|^2}. \\ \mathbf{v}_1[i]\mathbf{u}_0[i] - \mathbf{v}_0[i]^*\mathbf{u}_1[i] &= 0 \end{aligned} \quad (6)$$

We choose

$$\mathbf{L} = \begin{bmatrix} \text{Diag}(\mathbf{v}_0^*) & \text{Diag}(\mathbf{v}_1^*) \\ \text{Diag}(\mathbf{v}_1) & \text{Diag}(-\mathbf{v}_0) \end{bmatrix}$$

and $\mathbf{R} = \mathbf{L}^*$. $\mathbf{L}, \mathbf{R}$ are (permuted) direct sums of blocks of the form $\begin{bmatrix} \mathbf{v}_0[i]^* & \mathbf{v}_1[i]^* \\ \mathbf{v}_1[i] & -\mathbf{v}_0[i] \end{bmatrix}$, which are orthogonal by construction (via (5)). Hence, $\mathbf{L} \in \widetilde{\mathbf{B}}_n^{(n)}$ and $\mathbf{R} \in (\widetilde{\mathbf{B}}^*)_n^{(n)}$. Further,

$$\begin{aligned} \mathbf{LHR} &= \begin{bmatrix} \text{Diag}(\mathbf{v}_0^*) & \text{Diag}(\mathbf{v}_1^*) \\ \text{Diag}(\mathbf{v}_1) & \text{Diag}(-\mathbf{v}_0) \end{bmatrix} \left( \mathbf{I} - 2\begin{bmatrix} \mathbf{u}_0 \\ \mathbf{u}_1 \end{bmatrix} \begin{bmatrix} \mathbf{u}_0 \\ \mathbf{u}_1 \end{bmatrix}^* \right) \begin{bmatrix} \text{Diag}(\mathbf{v}_0^*) & \text{Diag}(\mathbf{v}_1^*) \\ \text{Diag}(\mathbf{v}_1) & \text{Diag}(-\mathbf{v}_0) \end{bmatrix}^* \\ &= \mathbf{I} - 2\begin{bmatrix} \text{Diag}(\mathbf{v}_0^*) & \text{Diag}(\mathbf{v}_1^*) \\ \text{Diag}(\mathbf{v}_1) & \text{Diag}(-\mathbf{v}_0) \end{bmatrix} \begin{bmatrix} \mathbf{u}_0 \\ \mathbf{u}_1 \end{bmatrix} \begin{bmatrix} \mathbf{u}_0 \\ \mathbf{u}_1 \end{bmatrix}^* \begin{bmatrix} \text{Diag}(\mathbf{v}_0^*) & \text{Diag}(\mathbf{v}_1^*) \\ \text{Diag}(\mathbf{v}_1) & \text{Diag}(-\mathbf{v}_0) \end{bmatrix}^* \\ &= \mathbf{I} - 2\underbrace{\begin{bmatrix} \mathbf{v}_0^* \circ \mathbf{u}_0 + \mathbf{v}_1^* \circ \mathbf{u}_1 \\ \mathbf{v}_1 \circ \mathbf{u}_0 - \mathbf{v}_0 \circ \mathbf{u}_1 \end{bmatrix}}_{\mathbf{w}} \underbrace{\begin{bmatrix} \mathbf{v}_0^* \circ \mathbf{u}_0 + \mathbf{v}_1^* \circ \mathbf{u}_1 \\ \mathbf{v}_1 \circ \mathbf{u}_0 - \mathbf{v}_0 \circ \mathbf{u}_1 \end{bmatrix}^*}_{\mathbf{w}}, \end{aligned}$$

where $\circ$ denotes the Hadamard product. From (6)

$$\mathbf{w}[i] = \begin{cases} \sqrt{|\mathbf{u}_0[i]|^2 + |\mathbf{u}_1[i]|^2} & i \in [n/2] \\ 0 & i \in [n/2 : n] \end{cases}$$

Denoting the first half of this vector by $\mathbf{w}_0 \in \mathbb{C}^{n/2}$, we have

$$\mathbf{LHR} = \begin{bmatrix} \mathbf{I} - 2\mathbf{w}_0\mathbf{w}_0^* & \mathbf{0} \\ \mathbf{0} & \mathbf{I} \end{bmatrix},$$

where $\|\mathbf{w}_0\|_2 = \|\mathbf{u}\|_2 = 1$. The result follows inductively. $\square$

As an immediate corollary, we can use Singular Value Decomposition to obtain a factorization for an arbitrary $n \times n$ matrix.

**Corollary K.3.** *Let $\mathbf{M}$ be an arbitrary $n \times n$ matrix. Then, $\mathbf{M} \in (\mathcal{OBB})^{2n-1}$, where all but one matrix in the decomposition is orthogonal (unitary).*

*Proof.* By employing Singular Value Decomposition, we can decompose $\mathbf{M}$ as $\mathbf{M} = \mathbf{U\Sigma V}^*$, where $\mathbf{U}, \mathbf{V}^*$ are orthogonal and $\mathbf{\Sigma}$ is diagonal. By Lemma K.2, $\mathbf{U}, \mathbf{V}^* \in (\mathcal{OBB})^{n-1}$, and trivially $\mathbf{\Sigma} \in \mathcal{OBB}$. Hence, $\mathbf{M} \in (\mathcal{OBB})^{2n-1}$. Note that $\mathbf{\Sigma}$ is the only matrix in the decomposition that is not orthogonal (unitary). $\square$

### K.3 CONSTRUCTIONS

We show that we can construct $s$-sparse matrices in the $\mathcal{OBB}$ hierarchy with the same width as the $\mathcal{BB}^*$ hierarchy. The proof follows a structure to that of Theorem 3. We begin by arguing about permutation and step matrices, then using the same factorization to argue that matrices with at most $n$ NNZ are contained in $(\mathcal{BB}^*)^4$. Then, we will appeal to a modified sum closure lemma to extend the argument to matrices of general $s$ NNZ. Similar to Appendix F, we can use these results to place all matrices with low-depth circuits for matrix vector multiplication in the $\mathcal{OBB}$ hierarchy.

#### K.3.1 PERMUTATIONS

We begin by presenting the argument that permutations are included in $\mathcal{OBB}$ as a corollary to Theorem 2.

**Corollary K.4.** *Let* $\mathbf{P}$ *be a permutation matrix. Then* $\mathbf{P} \in \widetilde{\mathcal{B}}\widetilde{\mathcal{B}}^*$.

*Proof.* We appeal to the decomposition from Theorem 2, noting that all butterfly factor matrices constructed in the proofs of Lemmas G.3 and G.1 are permutation matrices, and thus are orthogonal. Hence, $\mathbf{P} \in \mathcal{OBB}$ where the inner diagonal matrix is $\mathbf{I}$. $\square$

Similarly, the construction of Lemma G.6 also show that permutations are included in $\widetilde{\mathcal{B}}^*\widetilde{\mathcal{B}}$.

**Corollary K.5.** *Let* $\mathbf{P}$ *be a permutation matrix. Then* $\mathbf{P} \in \widetilde{\mathcal{B}}^*\widetilde{\mathcal{B}}$.

To prove the containment of sparse matrices within the $\mathcal{OBB}$ hierarchy, we make use of the following lemma.

**Lemma K.6.** *Let* $\mathbf{P}$ *be a permutation matrix and* $\mathbf{D}$ *a diagonal matrix. Then there exist diagonal matrices* $\mathbf{D}'$ *and* $\mathbf{D}''$ *such that:*

$$\mathbf{PD} = \mathbf{D}'\mathbf{P} \qquad \mathbf{DP} = \mathbf{PD}''.$$

*Proof.* Let $\sigma$ be the permutation such that $\mathbf{P}[i,j] = \delta_{i,\sigma(j)}$.

Define $\mathbf{D}'$ such that $\mathbf{D}'[\sigma(j), \sigma(j)] = \mathbf{D}[j,j]$. Then, if $i = \sigma(j)$:

$$(\mathbf{PD})[i,j] = \mathbf{P}[i,j]\mathbf{D}[j,j] = \mathbf{D}'[\sigma(j),\sigma(j)]\mathbf{P}[\sigma(j),j] = (\mathbf{D}'\mathbf{P})[\sigma(j),j] = (\mathbf{D}'\mathbf{P})[i,j].$$

Otherwise, if $i \neq \sigma(j)$, then $(\mathbf{PD})[i,j] = 0 = (\mathbf{D}'\mathbf{P})[i,j]$. Hence, $\mathbf{PD} = \mathbf{D}'\mathbf{P}$.

Define $\mathbf{D}''$ such that $\mathbf{D}''[j,j] = \mathbf{D}[\sigma(j), \sigma(j)]$. An analogous argument to above shows that $\mathbf{DP} = \mathbf{PD}''$. $\square$

#### K.3.2 STEP MATRICES

In the $\mathcal{BB}^*$ hierarchy (Lemma I.2), we were able to show that horizontal step matrices are butterfly matrices. Here, we present a similar result for the $\mathcal{OBB}$ hierarchy.

**Lemma K.7.** *Let* $\mathbf{H}$ *be an* $n \times n$ *horizontal step matrix. Then we can decompose* $\mathbf{H} = \mathbf{DO}$*, where* $\mathbf{D}$ *is a diagonal matrix and* $\mathbf{O} \in \widetilde{\mathcal{B}}$.

*Proof.* Throughout the proof, we make reference to the original horizontal step matrix construction given in Lemma I.2 and its proof.

To begin, we show that an arbitrary $2^k \times 2^k$ butterfly factor $\mathbf{H}_{2^k}$ in the decomposition of $\mathbf{H}$ can be expressed as the product of a diagonal matrix and an orthogonal butterfly factor. Since a butterfly factor is direct sum of $2 \times 2$ matrices, there is a permutation matrix $\mathbf{P}_{2^k}$ such that conjugation of $\mathbf{H}_{2^k}$ by $\mathbf{P}_{2^k}$ gives a block diagonal matrix $\mathbf{H}'_{2^k}$ of $\frac{n}{2}$ $2 \times 2$ matrices, i.e.

$$\mathbf{P}_{2^k}\mathbf{H}_{2^k}\mathbf{P}_{2^k}^* = \mathbf{H}'_{2^k}.$$

(See Figure 10 for an illustration.) Specifically, $\mathbf{P}_{2^k}$ is the permutation where:

$$\mathbf{P}_s[2i, :] = \mathbf{e}_i^T \qquad \mathbf{P}_s[2i+1, :] = \mathbf{e}_{i+\frac{n}{2}}^T.$$

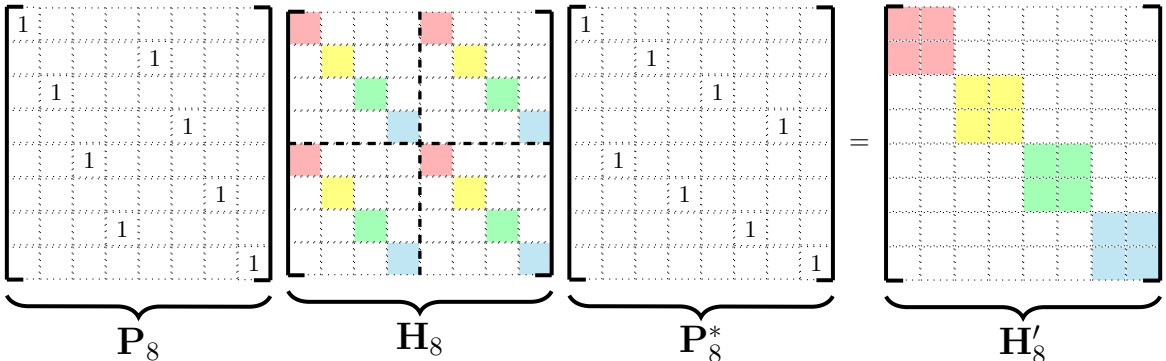

Figure 10: Block diagonalization of $\mathbf{H}_8$

We argue that each of these $2 \times 2$ blocks can be decomposed into a diagonal matrix times an orthogonal matrix. Note that the butterfly factor matrices constructed in the proof of Lemma I.2 each have at most one non-zero entry per column. Hence, there are 4 cases to consider. Note that matrices with at most one non-zero entry are exhausted by Cases 1 and 2.

Case 1: $\begin{bmatrix} a & 0 \\ 0 & b \end{bmatrix} = \underbrace{\begin{bmatrix} a & 0 \\ 0 & b \end{bmatrix}}_{\mathbf{D}} \underbrace{\begin{bmatrix} 1 & 0 \\ 0 & 1 \end{bmatrix}}_{\mathbf{O}}$

Case 2: $\begin{bmatrix} 0 & a \\ b & 0 \end{bmatrix} = \underbrace{\begin{bmatrix} a & 0 \\ 0 & b \end{bmatrix}}_{\mathbf{D}} \underbrace{\begin{bmatrix} 0 & 1 \\ 1 & 0 \end{bmatrix}}_{\mathbf{O}}$

Case 3: $\begin{bmatrix} a & b \\ 0 & 0 \end{bmatrix} = \underbrace{\begin{bmatrix} \sqrt{a^2 + b^2} & 0 \\ 0 & 0 \end{bmatrix}}_{\mathbf{D}} \underbrace{\begin{bmatrix} \frac{a}{\sqrt{a^2+b^2}} & \frac{b}{\sqrt{a^2+b^2}} \\ \frac{b}{\sqrt{a^2+b^2}} & \frac{-a}{\sqrt{a^2+b^2}} \end{bmatrix}}_{\mathbf{O}}, \qquad a, b \neq 0$

Case 4: $\begin{bmatrix} 0 & 0 \\ a & b \end{bmatrix} = \underbrace{\begin{bmatrix} 0 & 0 \\ 0 & \sqrt{a^2 + b^2} \end{bmatrix}}_{\mathbf{D}} \underbrace{\begin{bmatrix} \frac{b}{\sqrt{a^2+b^2}} & \frac{-a}{\sqrt{a^2+b^2}} \\ \frac{a}{\sqrt{a^2+b^2}} & \frac{b}{\sqrt{a^2+b^2}} \end{bmatrix}}_{\mathbf{O}}, \qquad a, b \neq 0$

In the last two cases, $\mathbf{O}$ is a $2 \times 2$ rotation matrix, which is commonly known to be orthogonal. Assume that we perform the above decomposition on all of the blocks of $\mathbf{H}'_{2^k}$ in parallel, therefore expressing $\mathbf{H}'_{2^k} = \mathbf{D}'\mathbf{O}'$. We now have

$$\mathbf{H}_{2^k} = \mathbf{P}^*_{2^k}\mathbf{D}'\mathbf{O}'\mathbf{P}_{2^k}.$$

By Lemma K.6, we can rewrite this as

$$\mathbf{H}_{2^k} = \mathbf{D}''\mathbf{P}^*_{2^k}\mathbf{O}'\mathbf{P}_{2^k}.$$

Note that $\mathbf{P}^*_{2^k}\mathbf{O}'\mathbf{P}_{2^k}$ is the product of three orthogonal matrices, and thus orthogonal. Additionally, the construction of $\mathbf{P}_{2^k}$ ensures that $\mathbf{P}^*_{2^k}\mathbf{O}'\mathbf{P}_{2^k}$ is butterfly factor.[11] Hence, $\mathbf{H}_{2^k}$ can be expressed as the product of a diagonal matrix and an orthogonal butterfly factor, as desired.

Now, we show that this decomposition of butterfly factors implies Lemma K.7. By performing this decomposition in parallel on each butterfly factor, we conclude that any butterfly factor matrix $\mathbf{H}^{(n)}_{2^k}$ of $\mathbf{H}$ can be decomposed as $\mathbf{H}^{(n)}_{2^k} = \mathbf{D}_{2^k}\mathbf{O}^{(n)}_{2^k}$.[12]

---

[11]Conjugation by $\mathbf{P}_{2^k}$ is an isomorphism from $2^k \times 2^k$ butterfly factors onto block diagonal matrices with $2^{k-1}$, $2 \times 2$ blocks. Therefore, conjugation by $\mathbf{P}^{-1}_{2^k} = \mathbf{P}^*_{2^k}$ maps a block diagonal matrix to a butterfly factor.

[12]Note that a block diagonal matrix composed of orthogonal matrices is, itself, orthogonal.

We complete the argument by induction on $n$. The base case $n = 2$ holds by the observation about butterfly factor matrices above. Assume that any horizontal step matrix of size $\frac{n}{2} \times \frac{n}{2}$ can be expressed as a diagonal matrix times an orthogonal butterfly matrix. Now, consider the $n \times n$ horizontal step matrix $\mathbf{H}$. From Lemma I.2, $\mathbf{H}$ can be expressed as

$$\mathbf{H} = \mathbf{B}_n^{(n)} \begin{bmatrix} \mathbf{H}_1 & 0 \\ 0 & \mathbf{H}_2 \end{bmatrix},$$

where $\mathbf{H}_1, \mathbf{H}_2$ are $\frac{n}{2} \times \frac{n}{2}$ horizontal step matrices. By our inductive hypothesis,

$$\mathbf{H} = \mathbf{B}_n^{(n)} \mathbf{D}_1 \begin{bmatrix} \mathbf{O}_1 & 0 \\ 0 & \mathbf{O}_2 \end{bmatrix},$$

where $\mathbf{D}_1$ is diagonal and $\mathbf{O}_1, \mathbf{O}_2$ are $\frac{n}{2} \times \frac{n}{2}$ matrices in $\widetilde{\mathcal{B}}$. However, $\mathbf{B}_n^{(n)} \mathbf{D}_1$ is a butterfly factor, and therefore can be expressed as $\mathbf{D}_n \mathbf{O}_n^{(n)}$. Therefore,

$$\mathbf{H} = \mathbf{D}_n \mathbf{O}_n^{(n)} \begin{bmatrix} \mathbf{O}_1 & 0 \\ 0 & \mathbf{O}_2 \end{bmatrix} = \mathbf{D}_n \mathbf{O},$$

with $\mathbf{O} \in \widetilde{\mathcal{B}}$, as desired. $\qquad \square$

Just as with the $\mathcal{B}\mathcal{B}^*$ hierarchy, the decomposition of vertical step matrices falls out as an immediate corollary to the horizontal step matrix proof.

**Corollary K.8.** *Let $\mathbf{V}$ be a vertical step matrix. Then we can decompose $\mathbf{V} = \mathbf{O}^* \mathbf{D}$, where $\mathbf{D}$ is a diagonal matrix and $\mathbf{O}^* \in \widetilde{\mathcal{B}}^*$.*

### K.3.3   SPARSE MATRICES

Now that we have argued about the decomposition of permutation and step matrices in the $\mathcal{O}\mathcal{B}\mathcal{B}$ hierarchy, we can leverage the construction from Lemma I.1 to argue about matrices with at most $n$ NNZ.

**Corollary K.9.** *Let $\mathbf{S}$ be an $n \times n$ matrix with at most $n$ NNZ. Then, $\mathbf{S} \in (\mathcal{O}\mathcal{B}\mathcal{B})^4$.*

*Proof.* We use the construction from Lemma I.1, along with Lemma K.7 and Corollary K.8, to express $\mathbf{S}$ as:

$$\mathbf{S} = \underbrace{\mathbf{O}_1 \mathbf{O}_1'}_{\mathbf{P}_1} \underbrace{\mathbf{D}_2 \mathbf{O}_2}_{\mathbf{H}} \underbrace{\mathbf{O}_3 \mathbf{O}_3'}_{\mathbf{P}_2} \underbrace{\mathbf{O}_4' \mathbf{D}_4}_{\mathbf{V}} \underbrace{\mathbf{O}_5 \mathbf{O}_5'}_{\mathbf{P}_3},$$

with each $\mathbf{O}_i \in \widetilde{\mathcal{B}}$, each $\mathbf{O}_j' \in \widetilde{\mathcal{B}}^*$, and each $\mathbf{D}_k$ diagonal. Since $\mathbf{P}_2$ is a permutation, by Corollary K.5, we can write it as $\tilde{\mathbf{O}}_3{}' \tilde{\mathbf{O}}_3$ for some $\tilde{\mathbf{O}}_3{}' \in \widetilde{\mathcal{B}}^*$ and $\tilde{\mathbf{O}}_3 \in \widetilde{\mathcal{B}}$. Moreover, noting that $\mathbf{O}_1'$ and $\mathbf{O}_5$ are permutations, we make use of Lemma K.6 to re-express $\mathbf{S}$ as:

$$\mathbf{S} = \underbrace{\mathbf{O}_1 \mathbf{D}_2' \mathbf{O}_1'}_{\mathbf{M}_1} \underbrace{\mathbf{O}_2 \tilde{\mathbf{O}}_3{}'}_{\mathbf{M}_2} \underbrace{\tilde{\mathbf{O}}_3 \mathbf{O}_4'}_{\mathbf{M}_3} \underbrace{\mathbf{O}_5 \mathbf{D}_4' \mathbf{O}_5'}_{\mathbf{M}_4}.$$

Note that each $\mathbf{M}_\ell \in \mathcal{O}\mathcal{B}\mathcal{B}$. Hence, $\mathbf{S} \in (\mathcal{O}\mathcal{B}\mathcal{B})^4$, as desired. $\qquad \square$

Just as in Appendix I, we would like to extend this orthogonal-based construction to capture matrices of general sparsity. To accomplish this, we introduce an addition closure lemma analogous to Lemma K.10 for the $\mathcal{O}\mathcal{B}\mathcal{B}$ hierarchy.

**Lemma K.10.** *Let $\mathbf{A}_1, \ldots, \mathbf{A}_m$ be $k \times k$ matrices in $(\mathcal{O}\mathcal{B}\mathcal{B})_e^w$ then $\sum_{i=1}^m \mathbf{A}_i \in (\mathcal{O}\mathcal{B}\mathcal{B})_{4e}^{mw}$.*

With Lemma K.10, we arrive at the following Corollary on general orthogonal sparsity.

**Corollary K.11.** *Let $\mathbf{S}$ be an $n \times n$ matrix with $s$ NNZ. Then, $\mathbf{S} \in (\mathcal{O}\mathcal{B}\mathcal{B})_4^{4\lceil \frac{s}{n} \rceil}$.*

*Proof.* Just as in the proof of Theorem 3, we accomplish this using a sum of $\left\lceil \frac{s}{n} \right\rceil$ matrices of at most $n$ NNZ. For handling the sum of matrices, we need to appeal to Lemma K.10. □

To conclude the argument, we give the proof of Lemma K.10.

*Proof of Lemma K.10.* For each $1 \leq i \leq m$, let $\mathbf{E_{A}}_i \in \mathbb{F}^{ek \times ek}$ be defined such that $\mathbf{A}_i = \mathbf{SE_{A}}_i \mathbf{S}^*$ (with $\mathbf{S}$ as in Definition 2.4). Note that $\mathbf{E_{A}}_i \in (\mathcal{OBB})^w$. Consider matrices of the form:

$$
\underbrace{\begin{bmatrix} \mathbf{I}_{ek} & 0 & 0 & \mathbf{E_{A}}_i \\ 0 & \mathbf{I}_{ek} & 0 & 0 \\ \mathbf{I}_{ek} & 0 & 0 & \text{-}\mathbf{E_{A}}_i \\ 0 & \mathbf{I}_{ek} & 0 & 0 \end{bmatrix}}_{\mathbf{M}_i \in \mathbb{F}^{4ek \times 4ek}} = \sqrt{2} \underbrace{\begin{bmatrix} \frac{1}{\sqrt{2}}\mathbf{I}_{2ek} & \frac{1}{\sqrt{2}}\mathbf{I}_{2ek} \\ \frac{1}{\sqrt{2}}\mathbf{I}_{2ek} & \text{-}\frac{1}{\sqrt{2}}\mathbf{I}_{2ek} \end{bmatrix}}_{\mathbf{O} \in \widetilde{\mathbf{B}}_{4ek}^{(4ek)}} \underbrace{\begin{bmatrix} \mathbf{I}_{ek} & 0 & 0 & 0 \\ 0 & \mathbf{I}_{ek} & 0 & 0 \\ 0 & 0 & \mathbf{E_{A}}_i & 0 \\ 0 & 0 & 0 & 0 \end{bmatrix}}_{\mathbf{K}} \underbrace{\begin{bmatrix} \mathbf{I}_{ek} & 0 & 0 & 0 \\ 0 & \mathbf{I}_{ek} & 0 & 0 \\ 0 & 0 & 0 & \mathbf{I}_{ek} \\ 0 & 0 & \mathbf{I}_{ek} & 0 \end{bmatrix}}_{\mathbf{P} \in \widetilde{\mathbf{B}}_{2ek}^{(4ek)}}
$$

Note that $\mathbf{K}$, a block diagonal matrix composed of matrices in $(\mathcal{OBB})^w$, is itself in $(\mathcal{OBB})^w$ since

$$
\mathbf{K} = \prod_{j=1}^{w} \underbrace{\begin{bmatrix} \mathbf{I}_{ek} & 0 & 0 & 0 \\ 0 & \mathbf{I}_{ek} & 0 & 0 \\ 0 & 0 & \mathbf{O}_j & 0 \\ 0 & 0 & 0 & \mathbf{I}_{ek} \end{bmatrix}}_{\mathbf{L}_j \in \widetilde{\mathcal{B}}} \underbrace{\begin{bmatrix} \mathbf{I}_{ek} & 0 & 0 & 0 \\ 0 & \mathbf{I}_{ek} & 0 & 0 \\ 0 & 0 & \mathbf{D}_j & 0 \\ 0 & 0 & 0 & 0 \end{bmatrix}}_{\text{Diagonal}} \underbrace{\begin{bmatrix} \mathbf{I}_{ek} & 0 & 0 & 0 \\ 0 & \mathbf{I}_{ek} & 0 & 0 \\ 0 & 0 & \mathbf{O}'_j & 0 \\ 0 & 0 & 0 & \mathbf{I}_{ek} \end{bmatrix}}_{\mathbf{R}_j \in \widetilde{\mathcal{B}}^*},
$$

where each $\mathbf{O}_j$ is a $ek \times ek$ matrix in $\widetilde{\mathcal{B}}$, and each $\mathbf{O}'_j$ is a $ek \times ek$ matrix in $\widetilde{\mathcal{B}}^*$. $\mathbf{L}_w$ (the leftmost factor) is a block diagonal matrix composed of 4 $ek \times ek$ matrices in $\widetilde{\mathcal{B}}$. Therefore, we can fold $\mathbf{O}$ into this factor (since a butterfly factor in $\widetilde{\mathbf{B}}_{4ek}^{(4ek)}$ was not yet used in $\mathbf{L}_w$) to conclude that $\mathbf{OL}_w \in \widetilde{\mathcal{B}}$. Similarly, since no btterfly factor from $\widetilde{\mathbf{B}}_{2ek}^{(4ek)}$ has been used in $\mathbf{R}_1$, we may fold $\mathbf{P}$ into $\mathbf{R}_1$ to conclude that $\mathbf{R}_1\mathbf{P} \in \widetilde{\mathcal{B}}^*$. Finally, we address the scalar multiple of $\sqrt{2}$ by multiplying all entries of any diagonal matrix in the decomposition of $\mathbf{K}$ by $\sqrt{2}$. Hence, we may conclude that $\mathbf{M}_i \in (\mathcal{OBB})^w$.

Through repeated application ($m$ times) of the identity

$$
\begin{bmatrix} \mathbf{I} & \mathbf{A}_1 & 0 & \mathbf{B}_1 \\ 0 & \mathbf{I} & 0 & 0 \\ \mathbf{I} & \mathbf{A}_2 & 0 & \mathbf{B}_2 \\ 0 & \mathbf{I} & 0 & 0 \end{bmatrix} \begin{bmatrix} \mathbf{I} & 0 & 0 & \mathbf{C}_1 \\ 0 & \mathbf{I} & 0 & 0 \\ \mathbf{I} & 0 & 0 & \mathbf{C}_2 \\ 0 & \mathbf{I} & 0 & 0 \end{bmatrix} = \begin{bmatrix} \mathbf{I} & \mathbf{A}_1 + \mathbf{B}_1 & 0 & \mathbf{C}_1 \\ 0 & \mathbf{I} & 0 & 0 \\ \mathbf{I} & \mathbf{A}_2 + \mathbf{B}_2 & 0 & \mathbf{C}_1 \\ 0 & \mathbf{I} & 0 & 0 \end{bmatrix}, \tag{7}
$$

we see that

$$
\prod_{i=1}^{m} \mathbf{M}_i = \underbrace{\begin{bmatrix} \mathbf{I}_{ek} & \sum_{i=2}^{m} \mathbf{E_{A}}_i & 0 & \mathbf{E_{A}}_1 \\ 0 & \mathbf{I}_{ek} & 0 & 0 \\ \mathbf{I}_{ek} & \text{-}\mathbf{E_{A}}_m + \sum_{i=2}^{m-1} \mathbf{E_{A}}_i & 0 & \mathbf{E_{A}}_1 \\ 0 & \mathbf{I}_{ek} & 0 & 0 \end{bmatrix}}_{\mathbf{M} \in \mathbb{F}^{4en \times 4en}}.
$$

Therefore, $\mathbf{M} \in (\mathcal{OBB})^{mw}$. Next, we note that

$$
\sum_{i=1}^{m} \mathbf{A}_i = \mathbf{SM} \underbrace{\begin{bmatrix} 0 & \mathbf{I}_{ek} & 0 & \mathbf{I}_{ek} \\ \mathbf{I}_{ek} & 0 & \mathbf{I}_{ek} & 0 \\ 0 & \mathbf{I}_{ek} & 0 & \text{-}\mathbf{I}_{ek} \\ \mathbf{I}_{ek} & 0 & \text{-}\mathbf{I}_{ek} & 0 \end{bmatrix}}_{\mathbf{Q}} \mathbf{S}^T.
$$

We would like to show that we can fold $\mathbf{Q}$ into the rightmost $\mathcal{OBB}$ factor of $\mathbf{M}$. The rightmost matrix in the decomposition of $\mathbf{M}$ is $\mathbf{P}$. Note that

$$
\mathbf{PQ} = \begin{bmatrix} 0 & \mathbf{I}_{ek} & 0 & \mathbf{I}_{ek} \\ \mathbf{I}_{ek} & 0 & \mathbf{I}_{ek} & 0 \\ \mathbf{I}_{ek} & 0 & \text{-}\mathbf{I}_{ek} & 0 \\ 0 & \mathbf{I}_{ek} & 0 & \text{-}\mathbf{I}_{ek} \end{bmatrix} = \sqrt{2} \underbrace{\begin{bmatrix} 0 & \mathbf{I}_{ek} & 0 & 0 \\ \mathbf{I}_{ek} & 0 & 0 & 0 \\ 0 & 0 & \mathbf{I}_{ek} & 0 \\ 0 & 0 & 0 & \mathbf{I}_{ek} \end{bmatrix}}_{\widetilde{\mathbf{B}}_{2ek}^{(4ek)}} \underbrace{\begin{bmatrix} \frac{1}{\sqrt{2}}\mathbf{I}_{2ek} & \frac{1}{\sqrt{2}}\mathbf{I}_{2ek} \\ \frac{1}{\sqrt{2}}\mathbf{I}_{2ek} & \text{-}\frac{1}{\sqrt{2}}\mathbf{I}_{2ek} \end{bmatrix}}_{\widehat{\mathbf{B}}_{4ek}^{(4ek)}}.
$$

Just as earlier, the factor of $\sqrt{2}$ can be multiplied through any diagonal matrix. Also, these two orthogonal butterfly factor matrices can be folded into the the rightmost $\mathbf{R}$ matrix (the decomposition of $\mathbf{K}$ above does not use these two, rightmost butterfly factors). Hence, $\sum_{i=1}^{m} \mathbf{A}_i \in (\mathcal{OBB})_{4e}^{mw}$, as desired. $\qquad\square$

### K.3.4 ARITHMETIC CIRCUITS

Just as in Theorem 1, we can use the sparsity result in Lemma K.10 to place matrices with low-depth (linear) arithmetic circuits for matrix vector multiplication in the $\mathcal{OBB}$ hierarchy.

**Corollary K.12.** *Let $\mathbf{M}$ be an $n \times n$ matrix such that matrix-vector multiplication of $\mathbf{M}$ times an arbitrary vector $\mathbf{v}$ can be represented as a be a linear arithmetic circuit $C$ comprised of $s$ gates (including inputs) and having depth $d$. Then, $\mathbf{M} \in (\mathcal{OBB})_{O(\frac{s}{n})}^{O(d)}$.*

*Proof.* We use the construction given in the proof of Theorem 1. Corollaries K.9 and K.4 allow us to recover the same width and expansion factor with the $\mathcal{OBB}$ hierarchy. $\qquad\square$

## L  RELU NETWORK WITH STRUCTURED WEIGHT MATRICES

We show that for any neural network with ReLU nonlinearities and whose weight matrices have arithmetic circuits with few gates, its linear network counterpart (obtained by removing all the ReLU's) also has an arithmetic circuit with not too many more gates. This implies that in trying to find the smallest arithmetic circuit augmented with ReLU gates to represent a ReLU network, one might as well try to find the smallest arithmetic circuits that represent the matrix-vector multiplication of each weight matrix.

**Proposition 2.** *Consider a neural network architecture consisting of $L$ layers with weight matrices $\mathbf{W}_1, \ldots, \mathbf{W}_L \in \mathbb{F}^{n \times n}$ and ReLU nonlinearity in between.*

*Suppose that matrix-vector multiplication of $\mathbf{W}_i$ times an arbitrary vector $\mathbf{v}$ can be represented as a linear arithmetic circuit with $s_i$ gates (including inputs). Then there exists an arithmetic circuit augmented with ReLU gates with $\sum_{i=1}^{L} s_i + Ln$ total gates that computes the output $\mathrm{ReLU}(\mathbf{W}_L(\ldots \mathrm{ReLU}(\mathbf{W}_1 \mathbf{v})))$ of the network for an arbitrary input vector $\mathbf{v}$.*

*Conversely, if there is an arithmetic circuit augmented with ReLU gates with $s$ total gates that computes all the activations of the network $\mathrm{ReLU}(\mathbf{W}_1 \mathbf{v}), \ldots, \mathrm{ReLU}(\mathbf{W}_L \ldots \mathrm{ReLU}(\mathbf{W}_1 \mathbf{v}))$ for an arbitrary input $\mathbf{v}$, then there exists an arithmetic circuit augmented with ReLU gates with $2s + 2Ln$ total gates that computes the activations of the network without ReLU $\mathbf{W}_1 \mathbf{v}, \ldots, \mathbf{W}_L \ldots \mathbf{W}_1 \mathbf{v}$.*

*Proof of Proposition 2.* To compute the output of the network $\mathrm{ReLU}(\mathbf{W}_L(\ldots \mathrm{ReLU}(\mathbf{W}_1 \mathbf{v})))$, we first compute the matrix-vector product $\mathbf{W}_1 \mathbf{v}$ with an arithmetic circuit of $s_1$ gates by assumption, and use $n$ other ReLU gates to compute the pointwise ReLU. Then we repeat the process for layer $2, 3, \ldots, L$, using the arithmetic circuits of $\mathbf{W}_1, \ldots, \mathbf{W}_L$ and $Ln$ additional gates for ReLU. In total we obtain an arithmetic circuit augmented with ReLU gates with $\sum_{i=1}^{L} s_i + Ln$ total gates.

Conversely, to build an arithmetic circuit augmented with ReLU gates to compute $\mathbf{W}_1 \mathbf{v}, \ldots, \mathbf{W}_L \ldots \mathbf{W}_1 \mathbf{v}$, we pass $\mathbf{v}$ and then $-\mathbf{v}$ through the circuit that computes $\mathrm{ReLU}(\mathbf{W}_1 \mathbf{x})$ for an arbitrary $\mathbf{x}$ to get $\mathrm{ReLU}(\mathbf{W}_1 \mathbf{v})$ and $\mathrm{ReLU}(-\mathbf{W}_1 \mathbf{v})$. Noting that $x = \mathrm{ReLU}(x) - \mathrm{ReLU}(-x)$, we can use $n$ additional gates to compute $\mathbf{W}_1 \mathbf{v}$ from $\mathrm{ReLU}(\mathbf{W}_1 \mathbf{v})$ and $\mathrm{ReLU}(-\mathbf{W}_1 \mathbf{v})$.

Repeat the process for layer $2, 3, \ldots, L$ (for example, pass $\mathbf{W}_1 \mathbf{v}$ and $-\mathbf{W}_1 \mathbf{v}$ to the circuit that computes $\mathbf{W}_2 \mathbf{x}$ for an arbitrary $\mathbf{x}$ on layer 2). Overall we need to double the circuits that computes all the activations of the network $\mathrm{ReLU}(\mathbf{W}_1 \mathbf{v}), \ldots, \mathrm{ReLU}(\mathbf{W}_L \ldots \mathrm{ReLU}(\mathbf{W}_1 \mathbf{v}))$, requiring $2s$ gates. We also need $n$ additional gates per layer to compute the negation of the input to that layer (e.g. computing $-\mathbf{v}$ from $\mathbf{v}$), and $n$ additional gates per layer to subtract the output of the ReLU circuit (e.g. computing $\mathbf{W}_1 \mathbf{v}$ from $\mathrm{ReLU}(\mathbf{W}_1 \mathbf{v})$ and $\mathrm{ReLU}(-\mathbf{W}_1 \mathbf{v})$.) Therefore we can construct an arithmetic circuit augmented with ReLU gates with $2s + 2L$ total gates that computes the activations of the network without ReLU $\mathbf{W}_1 \mathbf{v}, \ldots, \mathbf{W}_L \ldots \mathbf{W}_1 \mathbf{v}$.

$\qquad\square$

We now prove an asymptotic bound on the VC dimension of a ReLU network whose weight matrices are kaleidoscope matrices with bounded width and expansion.

**Proposition 3.** *Let $\mathcal{F}$ be the class of ReLU neural networks consisting of $L$ layers, where each layer is a K-matrix with width and expansion bounded by some constant $C$. Suppose that the network has $W$ total parameters. Let $\operatorname{sign}\mathcal{F}$ denote the corresponding classification functions: $\{x \mapsto \operatorname{sign} f(x) : f \in \mathcal{F}\}$. Then this class has VC dimension:*

$$\operatorname{VCdim}(\operatorname{sign}\mathcal{F}) = O(LW \log W).$$

We leverage the result from Thomas et al. (2018) for the case where the entries of the weight matrices interact multiplicatively, but with polynomially bounded degrees. This proof is similar to the VC bound for ReLU networks whose weight matrices are butterfly matrices (Dao et al., 2019).

*Proof.* To use Theorem 3 of Thomas et al. (2018), we simply need to check that the entries of the linear layer, as polynomials of the parameters, has degree at most $c_1 m_l^{c_2}$ for some universal constant $c_1, c_2 > 0$, where $m_l$ is the size of output of the $l$-th layer. If the network weight matrices are K-matrices with bounded width and expansion, each weight matrix is a product of at most $c_3 \log m_l$ sparse factors, for some universal constant $c_3 > 0$. This means that the degree is polynomially bounded, which satisfies the condition of the theorem. Therefore the VC dimension is bounded to be almost linear in the number of parameters:

$$\operatorname{VCdim}(\operatorname{sign}\mathcal{F}) = O(LW \log W).$$

$\square$

# M    ARITHMETIC CIRCUIT PRIMER

We give a quick overview of arithmetic circuits. This is a model of computation that has been studied for numerous computational problems (and is the basic model for *algebraic complexity theory*). For our purposes, we will exclusively focus on arithmetic circuits for the matrix-vector multiplication problem. For a more detailed exposition, the reader is referred to the standard book on this topic (Bürgisser et al., 2013).

**Definition M.1** (Arithmetic Circuits). *An arithmetic circuit that computes $\mathbf{y} = \mathbf{A}\mathbf{x}$ (for $\mathbf{A} \in \mathbb{F}^{m \times n}$) has $n$ input gates (corresponding to $\mathbf{x}[0], \ldots, \mathbf{x}[n-1]$) and $m$ output gates (corresponding to $\mathbf{y}[0], \ldots, \mathbf{y}[m-1]$). All the internal gates correspond to addition, subtraction, multiplication and division[13] over the underlying field $\mathbb{F}$. The circuit is also allowed to use constants from $\mathbb{F}$ for 'free.' The definition of the internal gates can depend on $\mathbf{A}$ (as well as $\mathbf{x}$ of course). In other words, one can 'bake' the knowledge about $\mathbf{A}$ into the circuit.*

*The* size *$s$ of a circuit is $n$ plus the number of addition, multiplication, subtraction and division gates used in the circuit. The depth $d$ of a circuit is the minimum number of layers such that all gates in a given layer take as its input gates from previous layers.[14]*

One drawback of arithmetic circuits (especially for infinite fields e.g. $\mathbb{F} = \mathbb{R}$, which is our preferred choice in this work) is that they assume operations over $\mathbb{F}$ can be performed *exactly*. In particular, it ignores precision issues involved with real arithmetic. Nonetheless, this model turns out to be a very useful model in reasoning about the complexity of doing matrix-vector multiplication for any family of matrices.

Perhaps the strongest argument in support of arithmetic circuits is that a large (if not an overwhelming) majority of matrix-vector multiplication algorithm also imply an arithmetic circuit of size comparable to the runtime of the algorithm (and the depth of the circuit roughly correponds to the time taken to compute it by a parallel algorithm). For example consider the obvious algorithm to compute $\mathbf{A}\mathbf{x}$ (i.e. for each $i \in [m]$, compute $\mathbf{y}[i]$ as the sum $\sum_{i=0}^{n-1} \mathbf{A}[i,j]\mathbf{x}[j]$). It is easy to see that this algorithm implies an arithmetic circuit of size $O(nm)$ and depth $O(\log n)$.[15]

---

[13]Here we assume all the gates have two inputs.

[14]The *input layer* corresponding to the input gates does not contriubte to the depth.

[15]The claim on the depth follow from the fact that each of the sums $\sum_{i=0}^{n-1} \mathbf{A}[i][j]\mathbf{x}[j]$ can be computed in parallel. Further, the sum for each $i \in [m]$ can be done in $\log_2 m$ depth by first computing the partial sums $\mathbf{A}[i][2j']\mathbf{x}[2j'] + \mathbf{A}[i][2j' + 1]\mathbf{x}[2j' + 1]$ for all $j' \in [n/2]$ in parallel and recursively computing pair-wise sums till we are done.

One thing to note about the arithmetic circuit above is that all the multiplications involve at least one input that is a constant from $\mathbb{F}$ (recall that we can assume that the entries of $\mathbf{A}$ are constants that can be used to build the circuit). This leads to the following important sub-class of arithmetic circuits:

**Definition M.2** (Linear Arithmetic Circuits). *An arithmetic circuit is called a* linear *arithmetic circuit if it only uses addition, subtraction and multiplication. Further, every multiplcation has a fixed constant from $\mathbb{F}$ as at least one of its two inputs. In other words, all gates in the circuit are linear functions of their inputs (i.e. of the form $ax + by$ for fixed constants $a, b \in \mathbb{F}$).*

Intuitively for the matrix-vector multiplication, it makes sense to consider linear arithmetic circuits since the final function we want to compute $\mathbf{Ax}$ is indeed a linear function of its inputs. For inifinite fields (e.g. $\mathbb{F} = \mathbb{R}$ or $\mathbb{F} = \mathbb{C}$), it turns out that this is essentially without loss of generality:

**Theorem 4** ((Bürgisser et al., 2013)). *Let $\mathbb{F}$ be an infinite field. Any (general) arithmetic circuit to compute $\mathbf{Ax}$ over $\mathbb{F}$ of size $s$ and depth $d$ can be converted into a* linear *arithmetic circuit of size $O(s)$ and depth $O(d)$.*

The above result implies that for asymptotic considerations, linear arithmetic circuits for matrix-vector multiplication are equivalent to general arithmetic circuits.[16]

One important property of linear arithmetic circuits of depth $d$, which we will use in our arguments, is that such a circuit can be equivalently represented as product of $d$ sparse matrices (see the proof of Theorem 1 for the precise derivation[17]).

As mentioned earlier, a vast majority of efficient matrix vector multiplication algorithms are equivalent to small (both in size and depth) linear arithmetic circuit. For example the FFT can be thought of as an efficient arithmetic circuit to compute the Discrete Fourier Transform (indeed when one converts the linear arithmetic circuit for FFT into a matrix decomposition,[18] then each matrix in the decomposition is a butterfly factor, with each block matrix in each factor being the same). For an illustration of this consider the DFT with $n = 4$ as illustrated in Figure 11.

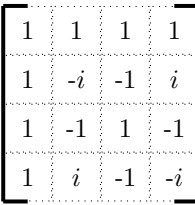

Figure 11: DFT of order $4$.

Figure 12 represent the arithmetic circuit corresponding to FFT with $n = 4$.

---

[16]This follows from the fact that by definition any linear arithmetic circuit is also an arithmetic circuit; the other direction follows from Theorem 4.

[17]To the best of our knowledge, this connection was explicitly made by De Sa et al. (2018) though the connection seems to be folklore.

[18]Using the conversion mentioned in the paragraph above.

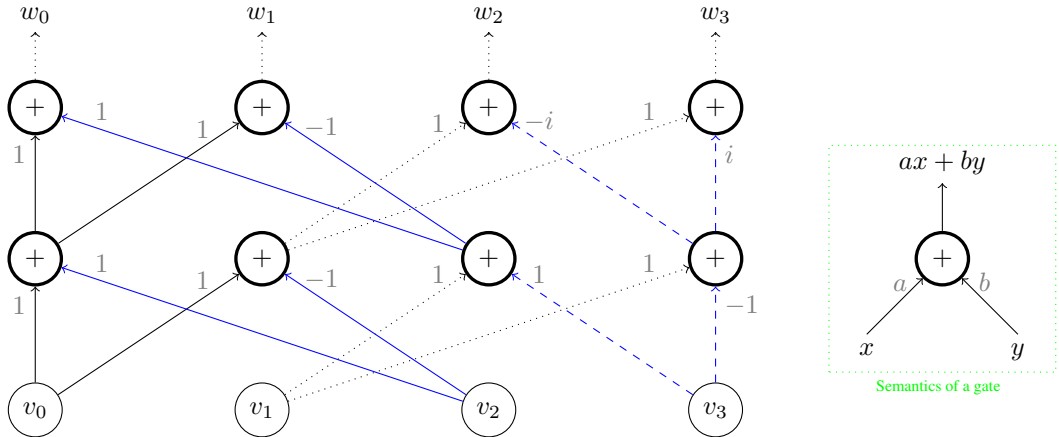

Figure 12: Arithmetic circuit for $4$-DFT from Figure 11.

Finally, Figure 13 is representation of the arithmetic circuit of Figure 12 as a product of a butterfly matrix and (the bit-reversal) permutation. We note that our generic arithmetic circuit to decomposition into $\mathcal{BB}^*$ is not as tight as in Figure 13.

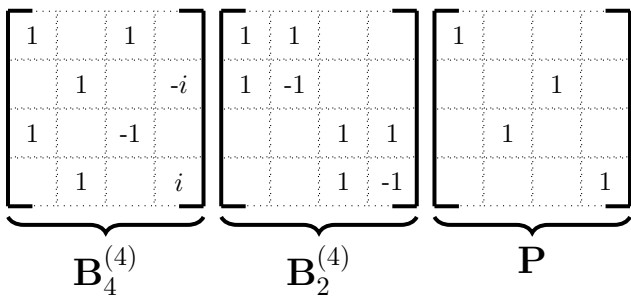

Figure 13: Decomposition of DFT of Figure 11 via the arithmetic circuit of Figure 12.

One reason for the vast majority of existing efficient matrix vector algorithms leading to (linear) arithmetic circuits is that they generally are divide and conquer algorithms that use polynomial operations such as polynomial multiplication or evaluation (both of which themselves are divide and conquer algorithms that use FFT as a blackbox) or polynomial addition. Each of these pieces are well known to have small (depth and size) linear arithmetic circuits (since FFT has these properties). Finally, the divide and conquer structure of the algorithms leads to the circuit being of low depth. See the book of Pan (Pan, 2001) for a more elaborate description of this connection.

In fact, the recent work of De Sa et al. (De Sa et al., 2018) makes this fact explicit and presents the most general known structure on matrices that imply near-linear size linear arithmetic circuits for the corresponding matrix vector multiplication. Their work combines two separate classes of structures matrices– orthogonal polynomial transforms (Driscoll et al., 1997; Szegö, 1967) as well as matrices with low displacement rank (Kailath et al., 1979; Olshevsky & Shokrollahi, 2000)– and presents a linear class of linear arithmetic circuits to solve their matrix vector multiplication problem. We note that structured matrices with low displacement rank have been used to replace fully connected layers in some neural network architectures (Sainath et al., 2013; Thomas et al., 2018).

