# OpenReview forum: "Kaleidoscope: An Efficient, Learnable Representation For All Structured Linear Maps"
_ICLR.cc/2020/Conference — Accept (Spotlight)_

### Official Review · AnonReviewer3 · 2019-10-24
**Official Blind Review #3**

**Rating:** 6

**Review:**

The authors propose learnable "kaleidoscope matrices" (K-matrices) in place of manually engineered structured and sparse matrices. By capturing "all" structured matrices in a way that can be learned, and without imposing a specific structure or sparsity pattern, these K-matrices can improve on existing systems by
* capturing more structure (that was not handled by the existing manually engineered architecture),
* running faster than dense implementations.

The claim that "all" structured matrices can be represented efficiently is a strong one, and in section 2.3 the authors make it clear what they mean by this. Although the proof is long and beyond the expertise of this reviewer, the basic explanation given in section 2.3 makes their point clear for the non-expert reader.

The balance of the paper empirically tests the claims of learnable structure and efficiency.

On the basis that these experiments essentially bear out the claims of the paper, I selected to accept the paper.

Weaknesses:

1. Regarding the ISWLT translation task result:
With this dataset, it's a bit of a stretch to say there was "only a 1 point drop in BLEU score". That's a significant drop, and in fact the DynamicConv paper goes to significant lengths to make a smaller 0.8 point improvement. There are probably many other ways to trade BLEU score for efficiency, and without showing those other methods (and the point drops they have), it's not clear that K-matrices are a good way to speed up decoding a bit.





**Experience Assessment:**

I do not know much about this area.

**Review Assessment: Checking Correctness Of Derivations And Theory:**

I assessed the sensibility of the derivations and theory.

**Review Assessment: Checking Correctness Of Experiments:**

I assessed the sensibility of the experiments.

**Review Assessment: Thoroughness In Paper Reading:**

I read the paper at least twice and used my best judgement in assessing the paper.

---

> ### Author Response · Authors · 2019-11-12
> **Response to “Official Blind Review #3”**
>
> We thank the reviewer for their helpful feedback on our work.
>
> Regarding the IWSLT translation result, the key claim we aim to validate is that the theoretical efficiency of K-matrices translates to practical speedups on real models as well. We agree that there are other approaches that may offer different model quality vs. inference speed tradeoffs; we simply highlight that K-matrices are one promising method, especially given their important theoretical properties. We have added a performance comparison of K-matrices with other structured replacements such as circulant, Fastfood, ACDC, and Toeplitz-like in Appendix B.4.3, showing that K-matrices yield faster inference with similar BLEU score. We also point out that our DynamicConv model with K-matrices in the decoder attains a comparable BLEU score with the state-of-the-art from two years ago – the Transformer model, which continues to enjoy widespread use today – while having over 60% higher sentence throughput and 30% fewer parameters than this model.
>
> As mentioned in the shared response, we believe that the speed-quality tradeoff of K-matrices could be further improved with more extensively tuned and optimized implementations. Exploring how to continue to improve these structured compression approaches, while retaining the efficiency and theoretical benefits of K-matrices, is an exciting question for future investigation.

---

### Official Review · AnonReviewer2 · 2019-10-24
**Official Blind Review #2**

**Rating:** 8

**Review:**

This paper introduces a structured drop-in replacement for linear layers in a neural network, referred to as Kaleidoscope matrices. The class of such matrices are proven to be highly expressive and includes a very general class of sparse matrices, including convolution, Fastfood, and permutation matrices. Experiments are carried in a variety of settings: (i) can nearly replace a series of hand-designed feature extractor, (ii) can perform better than fixed permutation matrices (though parameter count also increased by 10%), (iii) can learn permutations, and (iv) can help reduce parameter count and increase inference speed with a small performance degradation of 1.0 BLEU on machine translation.

This appears to be a solid contribution in terms of both theory and practical use. As I have not thought much about expressiveness in terms of arithmetic circuits (though I was unable to fully follow or appreciate the derivations, the explanations all seem reasonable), my main comments are regarding experiments. Though there are experiments in different domains, each could benefit from some additional ablations, especially to existing parameterizations of structured matrices such as Fastfood, ACDC, and any of the multiple works on permutation matrices and/or orthogonal matrices. Though Kaleidoscope include these as special cases, it is not clear whether when given the same resources (either memory or computational cost), Kaleidoscope would outperform them. There is also a matter of ease of training compared to existing approximations or relaxations, e.g. Gumbel-Sinkhorn.

Pros:
 - The writing is easy to follow and concise, with contributions and place in the literature clearly stated.
 - The Kaleidoscope matrix seem generally applicable, both proven theoretically and shown empirically (experiments are spread across a wide range of domains).
 - The code includes specific C++ and CUDA kernels for computing K matrices, which will be very useful for adaptation.
 - The reasoning using arithmetic circuits seems interesting, and the Appendix includes a primer.

Cons:
 - For the squeezenet and latent permutation experiments, would be nice if there is a comparison to other parameterizations of permutation matrices, e.g. gumbel-sinkhorn.
 - For the speed processing experiment, did you test what the performance would be if K matrix is replaced by a fully connected layer? This comparison appears in other experiments, but seems to be missing here for some reason. It would lead to better understanding than only comparing to SincNet.
 - The setup for the learning to permute experiment is not as general as it would imply in the main text. The matrices are constrained so that an actual permutation matrix is always sampled, and the permutation is (had to be?) pretrained to reduce total variation for 100 epochs before jointly trained with the classifier. Though this is stated very clearly in the Appendix, I hope the authors can also communicate this clearly in the main text as it appears to be a crucial component of the experimental setup.

Comments:
 - How easy is it to train with K matrices? Did you have to change optimizer hyperparameter compared to existing baselines?
 - There seems to be some blurring between the meaning of structure (used to motivate K matrices in the introduction) and sparsity (used to analyze K matrices). Structure might also include parameter sharing, orthogonality, and maybe other concepts. For instance, while Kaleidoscope matrices might include the subclass of circulant matrices, can they also capture the same properties or "inductive bias" (for lack of better word) as convolutional layers when trained?

**Experience Assessment:**

I have read many papers in this area.

**Review Assessment: Checking Correctness Of Derivations And Theory:**

I assessed the sensibility of the derivations and theory.

**Review Assessment: Checking Correctness Of Experiments:**

I carefully checked the experiments.

**Review Assessment: Thoroughness In Paper Reading:**

I read the paper thoroughly.

---

> ### Author Response · Authors · 2019-11-12
> **Response to “Official Blind Review #2”**
>
> We thank the reviewer for their encouraging feedback and thoughtful comments on our work.
>
> Regarding the permutation learning experiment, in response to the feedback, we have revised the main text to clarify the setup. The core of the experiment is the ability to denoise permuted images using some representation of the permutation set. In order to do this successfully, it is necessary for such a representation to have certain properties such as inducing a distribution over permutations. We have implemented and added a comparison to the Gumbel-Sinkhorn method (Mena et al., 2018), which is a customized representation for permutations with these properties, and requires similar techniques (unsupervised objective, permutation sampling, etc.) in order to learn the latent structure. The ResNet classifier on top can be viewed primarily as a way to evaluate the quality of the learned permutation; both of these representations are capable of learning the right latent structure, with test accuracies of 93.6 (Kaleidoscope) and 92.9 (Gumbel-Sinkhorn) respectively. The highlight of this experiment is that the K-matrix representation also comes with the requisite properties for this learning pipeline, despite not being explicitly designed for permutation learning.
>
> Regarding comparison to a dense matrix for the speech experiment, in Table 5 (Appendix B.1.2), we compare the use of K-matrices in the raw-features speech model with several other classes of matrices, including dense matrices. For instance, we find that, while using a trainable dense matrix slightly outperforms just using the fixed FFT (0.3% drop in test phoneme error rate), using a K-matrix instead of a dense matrix yields a further improvement of 0.8% in the phoneme error rate.
>
> Regarding ease of training and hyperparameter tuning, we would like to re-emphasize that for all experiments, all hyperparameters for training were kept the same as those for training the default model architecture, other than those we explicitly mentioned as being tuned. In particular, we did not modify any hyperparameters (such as number of epochs, optimizer, or learning rate) for the ShuffleNet and DynamicConv experiments. For the TIMIT speech experiment, we tune only the “preprocessing layer” learning rate. This is because the default speech pipeline already uses different learning rates for different portions of the network, so there is no clear choice a priori for the learning rate of the “preprocessing layer” (note that most methods, including K-matrices, do not seem to be overly sensitive to the choice of this learning rate). Thus, in these experiments, K-matrices can be used as a drop-in replacement for linear layers without significant tuning effort.
>
> Regarding structure and sparsity: We use “structure” in the context of structured matrices to mean matrices with a fast (subquadratic) multiplication algorithm. Structured matrices have a sparse factorization with total NNZ on the order of the number of operations required in the multiplication. This connection was known in the algebraic complexity community, and formalized by De Sa et al. (2018).
>
> Regarding the inductive bias encoded by K-matrices: the building block of K-matrices is a butterfly matrix, which encodes the recursive divide-and-conquer structure of many fast algorithms such as the FFT. Analyzing the precise effects of the inductive bias imposed by K-matrices is an exciting question for future work.

---

### Official Review · AnonReviewer4 · 2019-11-01
**Official Blind Review #4**

**Rating:** 8

**Review:**

Summary
The authors introduce kaleidoscope matrices (K-matrices) and propose to use them as a substitute for structured matrices arising in ML applications (e.g. circulant matrix used for the convolution operation). The authors prove that K-matrices are expressive enough to capture any structured matrix with near-optimal space and matvec time complexity. The authors demonstrate that learnable K-matrices achieve similar metrics compared to hand-crafted features on speech processing and computer vision tasks, can learn from permuted images, achieve performance close to a CNN trained on unpermuted images and demonstrate the improvement of inference speed of a transformer-based architecture for a machine translation task.


Review
The overall quality of the paper is high. The main contribution of the paper is the introduction of a family of matrices called kaleidoscope matrices (or K-matrices) which can be represented as a product of block-diagonal matrices of a special structure. Because of the special structure, the family allows near-optimal time matvec operations with near-optimal space complexity for structured matrices which are commonly used in deep architectures.

The proposed approach is novel. It gives a new characterization of sparse matrices with optimal space complexity up to a logarithmic term. Moreover, the proposed characterization is able to learn any structured matrix and matvec time complexity of the K-matrix representation is near-optimal matvec time complexity of the structured matrix. Even though in the worst-case complexity is not optimal, the authors argue that for matrices that are commonly used in machine learning architectures (e.g. circulant matrix in a convolution layer) the characterization is optimal. This results in a new differentiable layer based on a K-matrix that can be trained with the rest of an architecture using standard stochastic gradient methods. However, it is worth noting that the reviewer is not an expert in the field, and it is hard for him to compare the proposed approach with previous work.

The paper is generally easy to follow. Even though the introduction of K-matrices requires a lot of definitions, they are presented clearly and Figure 1 helps to understand the concept of K-matrices. The experimental pipeline is also clear.

Given the special structure of the family, the reviewer might guess that having K-matrices can slow down the training, i.e. it might require more epochs to achieve the reported results compared to baselines. Providing training plots might increase the quality of the paper.

The experimental results are convincing. First, the authors show that K-matrices can be used instead of a handcrafted MFSC featurization in an LSTM-based architecture on the TIMIT speech recognition benchmark with only a 0.4% loss of phoneme error rate. Then, the authors evaluate K-matrices on ImageNet dataset. In order to do so, they compare a lightweight ShuffleNet architecture which uses a handcrafted permutation layer to the same architecture but with a learnable K-matrix instead of the permutation layer. The authors demonstrate the 5% improvement of accuracy over the ShuffleNet with 0.46M parameters with only 0.05M additional parameters of the K-matrix and the 1.2% improvement of accuracy over the ShuffleNet with 2.5M parameters with only 0.2M additional parameters of the K-matrix. Next, the authors show that K-matrices can be used to train permutations in image classification domains. In order to demonstrate so, they take the Permuted CIFAR-10 dataset and ResNet-18 architecture, insert a trainable K-matrix at the beginning of the architecture and compare against ResNet-18 with an inserted FC-layer (attempting to learn the permutation as well) and ResNet-18 trained on the original, unpermuted CIFAR-10 dataset. With K-matrix, the authors achieve a 7.9% accuracy improvement over FC+ResNet-18 and only a 2.4% accuracy drop compared to ResNet-18 trained on the original CIFAR-10. Finally, the authors demonstrate that K-matrices can be used instead of the decoder’s linear layers in a Transformer-based architecture on the IWSLT-14 German-English translation benchmark which allows obtaining 30% speedup of the inference using a model with 25% fewer parameters with 1.0 drop of BLEU score.

Overall, the analysis and the empirical evaluations suggest that K-matrices can be a practical tool in modern deep architectures with a variety of potential benefits and tradeoffs between a number of parameters, inference speed and accuracy, and ability to learn complex structures (e.g. permutations).


Improvements
1. Even though K-matrices are aimed at structured matrices, it would be curious either to empirically compare K-matrices to linear transformations in fully-connected networks (i.e. dense matrices) or to provide some theoretical analysis.
2. Section 3.3 argues that K-matrices allow to obtain an improvement of inference speed, however, providing the results of convergence speed (e.g. training plots with a number of epochs) will allow a better understanding of the proposed approach and will improve the quality of the paper.

**Experience Assessment:**

I do not know much about this area.

**Review Assessment: Checking Correctness Of Derivations And Theory:**

I assessed the sensibility of the derivations and theory.

**Review Assessment: Checking Correctness Of Experiments:**

I assessed the sensibility of the experiments.

**Review Assessment: Thoroughness In Paper Reading:**

I read the paper at least twice and used my best judgement in assessing the paper.

---

> ### Author Response · Authors · 2019-11-12
> **Response to “Official Blind Review #4”**
>
> We appreciate the reviewer’s positive comments about our work.
>
> Regarding the convergence and speed of training, we would like to stress that all hyperparameters for training were kept the same as those for training the default model architecture, other than those we explicitly mentioned as being tuned (e.g. learning rate for the speech experiment). In particular, for all experiments, the number of epochs is the same for both the baseline approach and the K-matrix approach. Additionally, for the speech preprocessing and ShuffleNet experiments, we compare the total wall-clock training time of our K-matrix approach to that of the baseline approach, in both cases finding that the training time required by our approach is at most 20% longer than that of the baseline approach. In our updated revision, we also include the training time comparison for the DynamicConv model in Appendix B.4.2 (in this case, the modified model with K-matrices actually trains slightly faster than the baseline). We agree with the reviewer that a training plot can help provide a better understanding of how our proposed approach performs, and therefore have included an example plot (for the ShuffleNet experiment) in our updated revision (in Appendix B.2.3).
>
> Regarding empirical comparisons to dense matrices, in Table 5 (Appendix B.1.2), we compare the use of K-matrices in the raw-features speech model with several other classes of matrices, including dense matrices. We find that, while using a trainable dense matrix slightly outperforms just using the fixed FFT (0.3% drop in test phoneme error rate), using a K-matrix instead of a dense matrix yields a further improvement of 0.8% in the phoneme error rate. Another empirical comparison of K-matrices and dense matrices is in Section 3.3, in which we replace the linear layers in the decoder of a DynamicConv model with K-matrices; these linear layers are by default dense (fully-connected) matrices. Theoretically, in Lemma E.3 we show that arbitrary dense matrices are contained in the BB* hierarchy – in particular, that any n x n matrix is in (BB*)^{2n-2}, which implies that its K-matrix representation requires at most (4n log n)*(2n-2) = O(n^2 log n) parameters and thus is tight up to a logarithmic factor in n.

---

### Public Comment · ~Yaroslav_Bulatov1 · 2019-11-08
**Kronecker-factored maps?**

Can this represent Kronecker-factored matrices efficiently?

---

> ### Author Response · Authors · 2019-11-12
> **Kaleidoscope and Kronecker-factored matrices**
>
> Since Kronecker-factored matrices have an efficient representation, they are automatically captured by a K-matrix with the correct number of parameters up to logarithmic factors.
> There is actually a tighter bound that can be made in the case of the Kronecker products specifically, relating the K-matrix width of the product A ⊗ B to the K-matrix widths of the constituents A and B. We have included this argument as Lemma H.7 in the updated draft.

---

### Author Response · Authors · 2019-11-12
**Shared response to reviewers**

We thank all the reviewers for their thoughtful feedback. We address general comments and questions from the reviewers here, and then answer specific questions in individual responses. We have also uploaded a revised draft improving clarity in response to the reviewers’ suggestions and feedback.

[Ease of training K-matrices]: As K-matrices are fully differentiable (thanks to the fixed sparsity pattern, Section 2.2), they can be trained jointly with the rest of the model using standard learning algorithms (such as SGD, as used in the paper). For all of the experiments, we use the same number of epochs (and other applicable hyperparameters) for K-matrices as for the baselines. Even though each K-matrix is a product of multiple (sparse) matrices, K-matrices take about the same number of training steps to converge. One reason is that they can be easily initialized or constrained to be orthogonal (Section 2.4), thus avoiding vanishing or exploding gradients.

[Role of speed experiment (IWSLT translation task, Section 3.3)]: Even though our implementation is not yet highly optimized, this experiment serves as a proof of concept showing that K-matrices can lead to speedup in practical applications. By contrast, lack of fast implementations has limited the applicability of many other large classes of structured matrices that are efficient in theory, such as Toeplitz-like (Sindhwani et al., 2015) or low-displacement rank (Thomas et al., 2018).

[Additional comparisons with other baselines]: We thank the reviewers for suggesting other baselines to compare against to gain further insights into the applicability of our method. For the permutation learning experiment, we have added comparison to the Gumbel-Sinkhorn method (Mena et al., 2018), a specialized method to learn permutations, and this yields similar performance to K-matrices. We have also compared K-matrices to circulant, Fastfood, ACDC, and Toeplitz-like in the DynamicConv translation experiment (Appendix B.4.3); we find that K-matrices outperform these matrix classes. We do not expect a “free lunch” however: for any particular task, there may be a specialized matrix class that will achieve the best performance on that task when subjected to some resource constraints (i.e. speed and memory). However, K-matrices are more general as they can efficiently capture any structured matrices (up to some additional logarithmic factors in space and runtime), thus avoiding the need for hand-picking a specialized matrix class for every task.

K-matrices are thus expressive (Section 2.3), and efficient both in theory (Section 2.3) and practice (Section 3.3), and their learnability allows them to replace hand-crafted transformations (Section 3.1) and capture challenging latent structures (Section 3.2). We are excited about future work on further hardware-optimized implementations to fully realize the memory and speed benefits of structured matrices.

---

### Public Comment · ~Abhyuday_Jagannatha4 · 2021-04-12
**Retract the paper from ICLR**

Dear ICLR,
this paper did not deserve to be nominated as spotlight, as this can goes only to the papers which are usable for general public. Please have a look at the codebase, this is terribly coded and this is not usable for general public. thank you

---

> ### Author Response · Authors · 2021-04-12
> **Repo and instructions**
>
> Hi Abhyuday,
>
> Just to make sure you're using the right repo, the code is here: https://github.com/HazyResearch/butterfly
> Have you tried the instructions in the repo? Happy to answer questions if you run into problems. You can contact us by email or by creating new Github issues.
>
> Tri

---

> > ### Public Comment · ~Abhyuday_Jagannatha4 · 2021-04-12
> > **repose**
> >
> > Dear Tri
> > Yes, I am using the same codebase, there are enoumerous files without any comments, please take time to organize the codebase.  thank you

---

### Decision · Program_Chairs · 2019-12-19

**Decision:**

Accept (Spotlight)

**Comment:**

The paper generalizes several existing results for structured linear transformations in the form of K-matrices. This is an excellent paper and all reviewers confirmed that.